# From Fragile to Certified: Wasserstein Audits of Group Fairness Under Distribution Shift

## Abstract

Group-fairness metrics (e.g., equalized odds) can vary sharply across resamples and are especially brittle under distribution shift, undermining reliable audits. We propose a Wasserstein distributionally robust framework that certifies worst-case group fairness over a ball of plausible test distributions centered at the empirical law. Our formulation unifies common group fairness notions via a generic conditional-probability functional and defines $\varepsilon$-Wasserstein Distributional Fairness ($\varepsilon$-WDF) as the audit target. Leveraging strong duality, we derive tractable reformulations and an efficient estimator (DRUNE) for $\varepsilon$-WDF. We prove feasibility and consistency and establish finite-sample certification guarantees for auditing fairness, along with quantitative bounds under smoothness and margin conditions. Across standard benchmarks and classifiers, $\varepsilon$-WDF delivers stable fairness assessments under distribution shift, providing a principled basis for auditing and certifying group fairness beyond observational data.

## 1 Introduction

Group–fairness metrics such as statistical parity and equalized odds are widely used to assess algorithmic equity, yet they are highly sensitive to small perturbations in the training data Besse et al. (2018); Barrainkua et al. (2023); Cooper et al. (2024) (Fig. 1). Even mild changes in dataset composition or train–test splits can cause large swings in measured fairness Friedler et al. (2019); Du & Wu (2021), eroding trust in reported guarantees Ji et al. (2020). Because distributions drift in practice, fairness measured on a single empirical sample is unreliable.

To obtain trustworthy assessments, distributionally robust optimization (DRO) evaluates *worst-case fairness* over a set of plausible distributions (e.g., a Wasserstein ball), rather than only the observed data. This guards against distribution shift and promotes models whose fairness and accuracy remain stable when test data diverge from the training set Rahimian & Mehrotra (2022); Lin et al. (2022); Montesuma et al. (2025).

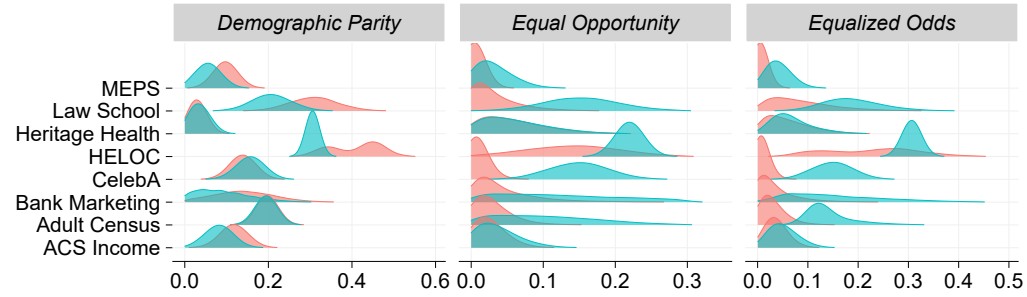

**Figure 1: Sensitivity of group fairness. Red** (Sample–Train–Measure): repeatedly subsample 1,000 points (10,000 reps), retrain, recompute fairness. **Blue** (Fixed-Model–Sample–Measure): train once per dataset, then repeatedly resample 1,000 points to recompute fairness. Large variability across datasets reveals fragility to sampling and measurement instability.

Given observational data $\{z_i = (x_i, a_i, y_i)\}_{i=1}^N$ with features $x_i \in \mathcal{X}$, sensitive attribute $a_i \in \mathcal{A}$, label $y_i \in \{0,1\}$, and a parametric binary classifier $h_\theta : \mathcal{X} \to \{0,1\}$, let $\mathbb{P}^N$ denote the empirical distribution and $\mathbb{P}$ the population distribution. A fairness–disparity functional $\mathcal{F}(\mathbb{P}, \theta)$ measures deviation from a chosen criterion (e.g., demographic parity, equalized odds) under $\mathbb{P}$; for tolerance $\varepsilon \geqslant 0$, we say $h_\theta$ is $\varepsilon$-fair on $\mathbb{P}$ if $|\mathcal{F}(\mathbb{P}, \theta)| \leqslant \varepsilon$ (If $\mathcal{F}$ is vector-valued, use $\|\cdot\|_\infty$). In finite samples, $\mathcal{F}(\mathbb{P}^N, \theta)$ can vary markedly with the particular observations included (Fig. 1), undermining the reliability of fairness assessments. The challenge intensifies under a distribution shift, where fairness judged on $\mathbb{P}^N$ may not reflect the population distribution, so we must certify fairness from the empirical law alone. To mitigate this sample dependence, we seek classifiers whose fairness holds not only on $\mathbb{P}^N$ but uniformly over an ambiguity set of plausible test distributions.

When designing an ambiguity set for DRO, two choices are paramount: (i) the *nominal* distribution and a realism-preserving uncertainty set around it; and (ii) *computational tractability*, i.e., whether optimization over that set admits efficient reformulations and algorithms. A principled way to encode *nearby* distributions is to use metric balls in probability space. While $f$-divergence balls are popular for analytic convenience, they ignore the geometry of the sample space and can fail under support mismatch. To respect geometry and remain meaningful with disjoint supports, we adopt optimal transport and measure distributional proximity with the Wasserstein distance Villani et al. (2009) of distributions $\mathbb{P}, \mathbb{Q}$ on $\mathcal{Z}$ and $q \in [1, \infty)$ with ground cost $c : \mathcal{Z} \times \mathcal{Z} \to \mathbb{R}_{\geqslant 0}$:

$$W_q(\mathbb{P}, \mathbb{Q}) := \inf_{\pi \in \mathcal{P}(\mathcal{Z} \times \mathcal{Z}): \ [\pi]_1 = \mathbb{P}, \ [\pi]_2 = \mathbb{Q}} \left( \mathbb{E}_{(z,z') \sim \pi} \left[ c(z, z')^q \right] \right)^{1/q},$$

where $\mathcal{P}(\mathcal{Z} \times \mathcal{Z})$ is set of all probability distributions on $\mathcal{Z} \times \mathcal{Z}$, $[\pi]_1, [\pi]_2$ are marginal distribution on the first and second coordinate. In real applications, the data-generating distribution drifts in ways that are hard to characterize. To guard against such shifts, we treat the nominal law $\mathbb{P}^*$ (in case distribution shift $\mathbb{P}^* \neq \mathbb{P}$) as any distribution within a Wasserstein distance $\delta$ of the population law $\mathbb{P}$ and define the ambiguity set $\mathcal{B}_\delta(\mathbb{P}) := \{\mathbb{Q} : W_q(\mathbb{P}, \mathbb{Q}) \leqslant \delta\}$, and posit $\mathbb{P}^* \in \mathcal{B}_\delta(\mathbb{P})$.

To handle distributional uncertainty in empirical fairness evaluation $\mathcal{F}(\mathbb{P}^*, \theta)$, we adopt a worst-case quantity of $\varepsilon$-fairness (formalized as $\varepsilon$-*Wasserstein Distributional Fairness* or $\varepsilon$-WDF in §3):

$$\sup_{\mathbb{Q} \in \mathcal{B}_\delta(\mathbb{P})} \left| \mathcal{F}(\mathbb{Q}, \theta) \right| \leqslant \varepsilon, \tag{1}$$

This certifies that the worst-case fairness disparity within a geometrically plausible neighborhood of $\mathbb{P}$ does not exceed $\varepsilon$. Enforcing Eq. 1 during learning is challenging: the constraint quantifies over an infinite-dimensional set of distributions, necessitating dual or surrogate reformulations for tractability. Moreover, standard DRO analyses typically assume Lipschitz or smooth objectives, whereas common group-fairness metrics are indicator-based and discontinuous, so off-the-shelf bounds do not apply. A further difficulty is observability: we cannot access the population ball $\mathcal{B}_\delta(\mathbb{P})$ and only have its empirical proxy $\mathcal{B}_\delta(\mathbb{P}^N)$; thus, we must certify the fairness of the nominal law $\mathbb{P}^*$ from samples, via finite-sample guarantees that relate $\mathcal{B}_\delta(\mathbb{P}^N)$.

In the out-of-sample problem, we only observe the empirical law $\mathbb{P}^N$, so the computable certificate is $\sup_{\mathbb{Q} \in \mathcal{B}_\delta(\mathbb{P}^N)} \left| \mathcal{F}(\mathbb{Q}, \theta) \right|$. The central question is how to calibrate $\delta$ (as a function of $N$) so that this empirical worst-case upper-bounds the population's worst-case $\mathcal{F}(\mathbb{P}, \theta)$ (with high probability), thereby certifying fairness for the population law.

In this work, we tackle these issues with a general framework not tied to a single fairness notion. It covers disparities expressed as differences of conditional probabilities, $\mathbb{P}(h_\theta(\mathbf{X}) = y \mid g_1(\mathbf{A}, \mathbf{Y}) = 0; g_2(\mathbf{A}, \mathbf{Y}) \geqslant 0)$, under trusted labels and sensitive attributes. For this class, we characterize the DRO worst-case, obtain an explicit regularizer with an efficient algorithm, and upper and lower bounds. In the out-of-sample case, we establish finite-sample certification. Our main contributions are:

- **Definition and guarantees.** Introduce $\varepsilon$-*Wasserstein distributional fairness* (Def. 1) and prove feasibility (Prop. 1) and consistency (Prop. 2) of robust fair learning problem (Eq. 6).

- **Tractable reformulation.** Derive a computable formulation of $\varepsilon$-WDF and the associated DRO regularizers (Thm. 1, Thm. 2), and present an efficient algorithm to compute $\varepsilon$-WDF (Alg. 1).
- **Finite-sample certification.** To mitigate out-of-sample problem, Provide finite-sample guarantees for auditing fairness(Thm. 4, Prop. 5).
- **Quantitative bounds.** Under smoothness of the decision boundary and data density, establish upper and lower bounds on $\varepsilon$-WDF (Prop. 6, Thm. 5, Thm. 6).

Additional theoretical results appear in the appendix.

## 1.1 Related Work

Several recent works use DRO to enhance fairness beyond the training set, either by optimizing fairness metrics over plausible distributions or by integrating optimal transport into fair learning. DRO has been applied to classification with fairness constraints, such as in support-vector classifiers and logistic regression using Wasserstein ambiguity sets and equal-opportunity constraints Wang et al. (2024b; 2021); Taskesen et al. (2020). Recent approaches also enforce fairness across perturbed datasets Ferry et al. (2023), extend worst-case group fairness Yang et al. (2023); Casas et al. (2024); Hu & Chen (2024); Miroshnikov et al. (2022), and explore alternative uncertainty sets Baharlouei & Razaviyayn (2023); Zhang et al. (2024); Rezaei et al. (2021); Zhi et al. (2025).

Closest to our setting, Chen et al. (2022) studies fairness transferability under structured shifts (e.g., covariate or label) with shared support, whereas our Wasserstein framework directly certifies exact worst-case fairness without these assumptions, enabling more general empirical-to-population transfers.

Furthermore, while Laakom et al. (2025) addresses fairness overfitting by deriving generalization bounds for nominal fairness, our framework provides a complementary solution by establishing finite-sample certification specifically for robust fairness under distribution shifts. A complementary line mitigates bias and noise via sample selection or reweighting, often with minimax optimization over $f$-divergence sets Du & Wu (2021); Roh et al. (2021); Wang et al. (2024a); Abernethy et al. (2020); Xiong et al. (2024); Hashimoto et al. (2018); Xiong et al. (2025); Jung et al. (2023). Other methods promote fairness by minimizing the Wasserstein distance between outputs across sensitive groups Jiang et al. (2020); Silvia et al. (2020); Chzhen et al. (2020), or by projecting to the closest group-independent distribution under the Wasserstein metric Si et al. (2021); Taskesen et al. (2021); Xue et al. (2020); Lin et al. (2024).

## 2 Background and Foundations

**Data Model.** Let $\mathbf{Z} = (\mathbf{X}, \mathbf{A}, \mathbf{Y})$ be a random vector on $(\mathcal{X}, \mathcal{A}, \mathcal{Y})$ with joint distribution $\mathbb{P}$. We assume feature space $\mathcal{X} \subset \mathbb{R}^d$, binary labels $\mathcal{Y} \in \{0,1\}$ and discrete sensitive attribute $\mathcal{A} \in \{1, \ldots, k\}$. The classifier $h_\theta : \mathcal{X} \to \mathcal{Y}$ is deterministic, trained without using $\mathbf{A}$, and has parameter $\theta \in \Theta \subset \mathbb{R}^K$.

**Fairness Notions.** Many group-fairness metrics (e.g., equalized odds) are defined as the difference between a classifier's conditional expectations over specific, disjoint subsets of $\mathcal{A} \times \mathcal{Y}$. Formally, let $\left\{S_0^i\right\}_{i \in \mathcal{I}}$ and $\left\{S_1^i\right\}_{i \in \mathcal{I}}$ be disjoint subsets of $\mathcal{A} \times \mathcal{Y}$ with positive measure, indexed by a finite set of $\mathcal{I}$ of size $m$. A classifier $h_\theta$ satisfies the $\varepsilon$-fairness if it meets all $m$ constraints:

$$\left| \mathbb{P}\left(h_\theta(\mathbf{X}) \mid S_0^i\right) - \mathbb{P}\left(h_\theta(\mathbf{X}) \mid S_1^i\right) \right| \leqslant \varepsilon \;\; \text{or} \;\; \left| \mathop{\mathbb{E}}_{z \sim \mathbb{P}}\left[ h_\theta(x) \left( \frac{\mathbb{1}_{S_0^i}(a,y)}{\mathbb{E}_\mathbb{P}[\mathbb{1}_{S_0^i}]} - \frac{\mathbb{1}_{S_1^i}(a,y)}{\mathbb{E}_\mathbb{P}[\mathbb{1}_{S_1^i}]} \right) \right] \right| \leqslant \varepsilon, \; \forall i \in \mathcal{I},$$

where $\mathbb{1}_S$ denotes the indicator of set $S$, and $\varepsilon$ is a tolerance for deviations from perfect fairness. To compactly encode $m$ fairness constraints, introduce the random vector $\mathbf{U}(\mathbf{A}, \mathbf{Y}) \in \{0,1\}^{2m}$ with:

$$\mathbf{U}(a,y) = (\mathbb{1}_{S_0^1}(a,y), ..., \mathbb{1}_{S_0^m}(a,y), \mathbb{1}_{S_1^1}(a,y), ..., \mathbb{1}_{S_1^m}(a,y))$$

We can then view the fairness constraints in terms of the value $h_\theta(\mathbf{X})$, the vector $\mathbf{U}$, and $\mathbb{E}[\mathbf{U}]$. Specifically, define a function $\varphi : \mathbb{R}^{2m} \times \mathbb{R}^{2m} \to \mathbb{R}^m$ by:

$$\varphi_i(U, \mu) = \frac{U_i}{\mu_i} - \frac{U_{i+m}}{\mu_{i+m}}, \quad \text{where} \mu = E[U], \quad \forall i \in [m]. \tag{2}$$

Then all constraints collapse into the generic notion of group fairness $\mathcal{F}(\mathbb{P}, \theta)$ Si et al. (2021); Kim et al. (2022):

$$\mathcal{F}(\mathbb{P}, \theta) := \mathbb{E}_{\mathbb{P}}\big[h_\theta(\mathbf{X})\varphi(\mathbf{U}, \mathbb{E}_{\mathbb{P}}[\mathbf{U}])\big]. \tag{3}$$

So $h_\theta$ is $\varepsilon$-fair if it meets all $m$ constraints, $\|\mathcal{F}(\mathbb{P}, \theta)\|_\infty \leqslant \varepsilon$ where $\|x\|_\infty = \max_{1 \leqslant i \leqslant m}|x_i|$.

**Example 1 (Equalized Odds).** *Let us consider the sensitive attribute is binary (e.g., gender). A classifier satisfies equalized odds if its true positive and false positive rates agree across $\mathbf{A} \in \{0, 1\}$:*

$$\big|\mathbb{P}(h_\theta(\mathbf{X}) = 1 \mid \mathbf{Y} = 1, \mathbf{A} = 0) - \mathbb{P}(h_\theta(\mathbf{X}) = 1 \mid \mathbf{Y} = 1, \mathbf{A} = 1)\big| \leqslant \varepsilon,$$

$$\big|\mathbb{P}(h_\theta(\mathbf{X}) = 1 \mid \mathbf{Y} = 0, \mathbf{A} = 0) - \mathbb{P}(h_\theta(\mathbf{X}) = 1 \mid \mathbf{Y} = 0, \mathbf{A} = 1)\big| \leqslant \varepsilon.$$

*Define $S_a^1 = \{z : \mathbf{Y} = 0, \mathbf{A} = a\}$ and $S_a^2 = \{z : \mathbf{Y} = 1, \mathbf{A} = a\}$ for $a \in \{0, 1\}$. Then*

$$\Big|\mathbb{E}_{\mathbb{P}}\big[h_\theta(\mathbf{X})\big(\tfrac{\mathbb{1}_{S_0^1}(a,y)}{\mathbb{P}(S_0^1)} - \tfrac{\mathbb{1}_{S_1^1}(a,y)}{\mathbb{P}(S_1^1)}\big)\big]\Big| \leqslant \varepsilon \quad and \quad \Big|\mathbb{E}_{\mathbb{P}}\big[h_\theta(\mathbf{X})\big(\tfrac{\mathbb{1}_{S_0^2}(a,y)}{\mathbb{P}(S_0^2)} - \tfrac{\mathbb{1}_{S_1^2}(a,y)}{\mathbb{P}(S_1^2)}\big)\big]\Big| \leqslant \varepsilon.$$

*Let $\mathbf{U}(a, y) = (\mathbb{1}_{S_0^1}(a,y), \mathbb{1}_{S_0^2}(a,y), \mathbb{1}_{S_1^1}(a,y), \mathbb{1}_{S_1^2}(a,y))$. By Eq. 2,*

$$\varphi(\mathbf{U}, \mathbb{E}_{\mathbb{P}}[\mathbf{U}]) = \Big(\tfrac{\mathbb{1}_{S_0^1}(a,y)}{\mathbb{E}_{\mathbb{P}}[\mathbb{1}_{S_0^1}]} - \tfrac{\mathbb{1}_{S_1^1}(a,y)}{\mathbb{E}_{\mathbb{P}}[\mathbb{1}_{S_1^1}]}, \tfrac{\mathbb{1}_{S_0^2}(a,y)}{\mathbb{E}_{\mathbb{P}}[\mathbb{1}_{S_0^2}]} - \tfrac{\mathbb{1}_{S_1^2}(a,y)}{\mathbb{E}_{\mathbb{P}}[\mathbb{1}_{S_1^2}]}\Big).$$

*Hence equalized odds is $\|\mathbb{E}_{\mathbb{P}}[h_\theta(\mathbf{X})\varphi(\mathbf{U}, \mathbb{E}_{\mathbb{P}}[\mathbf{U}])]\|_\infty \leqslant \varepsilon$ (for another example, see Example 2).*

**Strong Duality Theorem.** The DRO framework is particularly powerful when we can efficiently characterize the worst-case scenario. Given a function $\psi : \mathcal{Z} \to \mathbb{R}$, its worst-case expectation over an ambiguity set is defined as $\sup_{\mathbb{Q} \in \mathcal{B}_\delta(\mathbb{P})} \mathbb{E}_{x \sim \mathbb{Q}}[\psi(x)]$, where this quantity depends on the ambiguity radius $\delta$ and the reference probability distribution $\mathbb{P}$. A central tool for evaluating worst-case is the *strong duality Theorem* Gao et al. (2017); Mohajerin Esfahani & Kuhn (2018b); Blanchet & Murthy (2019). This theorem transforms the original hard optimization problem into a tractable, finite-dimensional one. Specifically, for any $q \in [1, \infty]$, it states:

$$\sup_{\mathbb{Q} \in \mathcal{B}_\delta(\mathbb{P})} \mathbb{E}_{z \sim \mathbb{Q}}[\psi(z)] = \begin{cases} \inf_{\lambda \geqslant 0}\{\lambda\delta^q + \mathbb{E}_{z \sim \mathbb{P}}[\psi_\lambda(z)]\} & 1 \leqslant q < \infty, \\ \mathbb{E}_{z \sim \mathbb{P}}\Big[\sup_{z' : c(z, z') \leqslant \delta} \psi(z')\Big] & q = \infty, \end{cases} \tag{4}$$

where $\psi_\lambda(z) := \sup_{z' \in \mathcal{Z}}\{\psi(z') - \lambda c^q(z, z')\}$.

**Remark 1 (Robust Optimization).** *When we take $q = \infty$, the Wasserstein ball $\mathcal{B}_\delta(\mathbb{P})$ enforces that every outcome $z$ can be perturbed by at most a distance $\delta$. Consequently, the DRO objective $\sup_{\mathbb{Q} \in \mathcal{B}_\delta(\mathbb{P})} \mathbb{E}_{\mathbb{Q}}[\psi(z)]$ collapses to the classic robust-optimization form $\mathbb{E}_{\mathbb{P}}\Big[\sup_{c(z, z') \leqslant \delta} \psi(z')\Big]$.*

## 3 Distributionally Robust Unfairness Quantification

In fairness-aware classifier learning, the training procedure is modified to promote equitable predictions with respect to protected attributes by incorporating fairness constraints into the optimization objective. The resulting training task is formulated as the following constrained optimization problem:

$$\inf_{\theta \in \Theta} \mathbb{E}_{\mathbb{P}^N}[\ell(h_\theta(\mathbf{X}), \mathbf{Y})] \quad \text{s.t.} \quad \|\mathbb{E}_{\mathbb{P}^N}[h_\theta(\mathbf{X})\varphi(\mathbf{U}, \mathbb{E}_{\mathbb{P}^N}[\mathbf{U}])]\|_\infty \leqslant \varepsilon \tag{5}$$

Here, $\ell : \mathbb{R} \times \mathcal{Y} \to \mathbb{R}$ is the loss function measuring prediction error. However, traditional fairness-aware learning assumes the training distribution perfectly represents the test environment, which is often violated due to sampling bias, covariate shift, or adversarial perturbations. To address this issue, a distributionally robust fair optimization problem is formulated as:

$$\inf_{\theta \in \Theta} \left\{ \sup_{\mathbb{Q} \in \mathcal{B}_\delta(\mathbb{P}^N)} \mathbb{E}_{\mathbb{Q}}[\ell(h_\theta(\mathbf{X}), \mathbf{Y})] \right\} \quad \text{s.t.} \quad \sup_{\mathbb{Q} \in \mathcal{B}_\delta(\mathbb{P}^N)} \left\{ \|\mathbb{E}_{\mathbb{Q}}[h_\theta(\mathbf{X})\varphi(\mathbf{U}, \mathbb{E}_{\mathbb{Q}}[\mathbf{U}])]\|_\infty \right\} \leqslant \varepsilon. \tag{6}$$

This formulation guarantees that the model $h_\theta$ minimizes the worst-case fairness violation over all plausible distributions, thereby certifying fairness under shifts within a Wasserstein ball around $\mathbb{P}^N$.

**Definition 1** ($\varepsilon$-**Wasserstein Distributional Fairness**). *A classifier $h_\theta$ is called $\varepsilon$-Wasserstein distributionally fair ($\varepsilon$-WDF) with respect to some fairness notion that is quantified by Eq. 3 if*

$$\sup_{\mathbb{Q} \in \mathcal{B}_\delta(\mathbb{P}^N)} \left\{ \| \mathbb{E}_{\mathbb{Q}}[h_\theta(\mathbf{X})\varphi(\mathbf{U}, \mathbb{E}_{\mathbb{Q}}[\mathbf{U}])] \|_\infty \right\} \leqslant \varepsilon. \tag{7}$$

Before presenting our main result, we begin by outlining the necessary assumptions.

**Assumption.**

(i) **Classifier:** *The family $\{h_\theta\}_{\theta \in \Theta}$ is insensitive to $\mathbf{A}$ and given by smooth score function $g_\theta$:*

$$h_\theta(x) = \mathbb{I}(g_\theta(x) \geqslant 0), \, g_\theta \in C(\mathcal{X}) \text{ with neural network head}, \, \Theta = \{\theta \in \mathbb{R}^K : \|\theta\| \leqslant R\}.$$

(ii) **Gradient Lower Bound:** $\exists \delta_0 > 0$ *such that* $\inf_{\substack{\theta \in \Theta \\ x \in \mathcal{X}: |g_\theta(x)| \leqslant \delta_0}} \|\nabla_x g_\theta(x)\|_{q*} > 0.$

(iii) **Bounded Density:** *Let $\mathcal{L}_\theta = \{x : g_\theta(x) = 0\}$ and $d(x, \mathcal{L}_\theta)$ distance $x$ to $\mathcal{L}_\theta$ then:*

$$\limsup_{\delta \downarrow 0} \sup_{\theta: \mathcal{L}_\theta \neq \varnothing} \frac{\mathbb{P}\big(0 \leqslant d(\mathbf{X}, \mathcal{L}_\theta) < \delta\big)}{\delta} < \infty.$$

(iv) **Cost Function:** *Let $d$ be a metric on $\mathcal{X} \times \mathcal{X}$. Then, the metric $c$ on $\mathcal{Z} \times \mathcal{Z}$ is defined as:*

$$c\big((x, a, y), (x', a', y')\big) = d(x, x') + \infty \mathbb{I}(a \neq a') + \infty \mathbb{I}(y \neq y').$$

Here $q*$ are conjugate exponents ($1/q* + 1/q = 1$). These assumptions are standard and mild in algorithmic fairness. (i) is standard and covers many classifier families, including linear/GLM, SVM, kernel, and neural networks with continuous activations. (ii) The uniform gradient lower bound ensures the decision boundary remains non-degenerate, aiding robustness and sensitivity analyses. (iii) The bounded-density condition prevents the distribution from concentrating excessive mass in an arbitrarily thin boundary layer. (iv) The cost metric assigns infinite cost to changes in the sensitive attribute or label—reflecting absolute trust in their values, as in previous works Taskesen et al. (2020); Wang et al. (2024b); Si et al. (2021).

**Remark 2.** *Our method applies with or without the sensitive attribute in the classifier. Excluding $\mathbf{A}$ is* not *fairness through unawareness; it reflects legal/policy limits (e.g., GDPR special-category data, U.S. Title VII), so we analyze the $\mathbf{A}$-excluded ($\mathbf{A}$-blind) setting.*

The applicability of problem 6 rests on two key properties: (i) *Feasibility*—for any tolerance level $\varepsilon$, a non-trivial robust classifier exists; and (ii) *Consistency*—as the perturbation budget vanishes ($\delta \to 0$), the robust minimizer converges to the solution of the classical fairness problem. The following two propositions formalize these properties.

**Proposition 1** (**Feasibility**). *By Assumption (i), for any $\varepsilon \in \mathbb{R}_+$, there exists almost sure (with probability 1) a non-trivial classifier ($h_\theta(x) \not\equiv constant$) that is feasible for the problem 6.*

**Proposition 2** (**Consistency**). *Let $\ell$ be a loss satisfying, for every $\theta \in \Theta$, the map $x \mapsto \ell\big(h_\theta(x), y\big)$ is uniformly $L$-Lipschitz with respect to the cost $d$ (e.g. Hinge loss). If there exists some $\theta_0 \in \Theta$ such that $\|\mathcal{F}(\mathbb{P}^N, \theta_0)\|_\infty < \varepsilon$, then any optimal solution $\theta_\delta^*$ of the robust problem 6 converges to the minimizer $\theta^*$ of the classical problem 5 as $\delta \to 0$.*

To characterize the form of $\varepsilon$-WDF, we begin by examining how our assumptions define the ambiguity set. The following proposition demonstrates the precise impact of these assumptions on its structure.

**Proposition 3** (**Shape of Ambiguity Set**). *Let $\mathbb{P} \in \mathcal{P}(\mathcal{Z})$ be a nominal distribution, and Assumption (iv) holds. Then the Wasserstein ambiguity set can be written as:*

$$\mathcal{B}_\delta(\mathbb{P}) = \left\{ \mathbb{Q} \in \mathcal{P}(\mathcal{Z}) : \mathbb{Q}_{\mathbf{A}, \mathbf{Y}} = \mathbb{P}_{\mathbf{A}, \mathbf{Y}} \quad \text{and} \quad \sum_{(a,y) \in \mathcal{A} \times \mathcal{Y}} \mathbb{P}_{\mathbf{A}, \mathbf{Y}}(a, y) W_q^q\big(\mathbb{Q}_{a,y}, \mathbb{P}_{a,y}\big) \leqslant \delta^q \right\},$$

*where $\mathbb{P}_{\mathbf{A}, \mathbf{Y}}$ and $\mathbb{Q}_{\mathbf{A}, \mathbf{Y}}$ are the marginals on $\mathcal{A} \times \mathcal{Y}$ under $\mathbb{P}$ and $\mathbb{Q}$, respectively, $\mathbb{P}_{a,y}$ and $\mathbb{Q}_{a,y}$ denote the conditional laws of $X$ given $(A = a, Y = y)$, and $W_q\big(\mathbb{Q}_{a,y}, \mathbb{P}_{a,y}\big)$ is the $q$-Wasserstein distance between these conditionals, measured with cost $d$.*

Proposition 3 implies that for any $\mathbb{Q}$ satisfying $\mathbf{W}_q(\mathbb{Q}, \mathbb{P}) \leqslant \delta$, the $(\mathbf{A}, \mathbf{Y})$-marginal distribution matches $\mathbb{P}$. Consequently, $\mathbb{E}_{\mathbb{Q}}[\mathbf{U}] = \mathbb{E}_{\mathbb{P}}[\mathbf{U}]$ remains constant. This allows us to simplify $\mathbb{E}_{\mathbb{Q}}\big[h_\theta(\mathbf{X})\,\varphi(\mathbf{U}, \mathbb{E}_{\mathbb{Q}}[\mathbf{U}])\big]$ into a function dependent solely on $(\mathbf{X}, \mathbf{U})$. Since $\mathbf{U}$ is fully determined by $(\mathbf{A}, \mathbf{Y})$, we can express the fairness notion as a **score fairness function** $f : \mathcal{Z} \to \mathbb{R}^m$, defined by:

$$f(z) := h_\theta(x)\,\varphi\big(\mathbf{U}(a, y), \mathbb{E}_{\mathbb{P}}[\mathbf{U}]\big). \tag{8}$$

To derive the $\varepsilon$-WDF constraint Eq. 7, we introduce for each $i \in [m]$ two upward and downward *Wasserstein regularizers*:

$$\mathcal{S}_{\delta,q}^i(\mathbb{P}, \theta) := \sup_{\mathbb{Q} \in \mathcal{B}_\delta(\mathbb{P})} \mathbb{E}_{\mathbb{Q}}\left[f_i(\mathbf{Z})\right] - \mathbb{E}_{\mathbb{P}}\left[f_i(\mathbf{Z})\right], \quad \mathcal{I}_{\delta,q}^i(\mathbb{P}, \theta) := \mathbb{E}_{\mathbb{P}}\left[f_i(\mathbf{Z})\right] - \inf_{\mathbb{Q} \in \mathcal{B}_\delta(\mathbb{P})} \mathbb{E}_{\mathbb{Q}}\left[f_i(\mathbf{Z})\right].$$

These quantify, respectively, the maximum upward and downward deviations of the fairness score relative to the nominal distribution over all $\mathbb{Q}$ in the Wasserstein ball. Let us define $\mathcal{S}_{\delta,q}(\mathbb{P}, \theta) = (\mathcal{S}_{\delta,q}^i(\mathbb{P}, \theta))_{i=1}^m$ and, similarly, $\mathcal{I}_{\delta,q}(\mathbb{P}, \theta)$, and denote the non-robust fairness measure by $\mathcal{F}(\mathbb{P}, \theta) = \mathbb{E}_{\mathbb{P}}[f(\mathbf{Z})]$. Under the assumptions of the following proposition, the classifier $h_\theta$ satisfies $\varepsilon$-WDF.

**Proposition 4** ($\varepsilon$-**WDF Condition**). *Let $\preceq$ denote component-wise comparison. The classifier $h_\theta$ satisfies the $\varepsilon$-WDF condition if and only if*

$$\mathcal{S}_{\delta,q}(\mathbb{P}, \theta) + \mathcal{F}(\mathbb{P}, \theta) \preceq \varepsilon \quad \text{and} \quad \mathcal{I}_{\delta,q}(\mathbb{P}, \theta) - \mathcal{F}(\mathbb{P}, \theta) \preceq \varepsilon \tag{9}$$

Proposition 4 states that for each $i$, we need to have $\mathcal{S}_{\delta,q}^i(\mathbb{P}, \theta) + \mathcal{F}_i(\mathbb{P}, \theta) \leqslant \varepsilon$ and $\mathcal{I}_{\delta,q}^i(\mathbb{P}, \theta) - \mathcal{F}_i(\mathbb{P}, \theta) \leqslant \varepsilon$. Henceforth, for simplicity, we assume that the number of fairness constraints in Eq. 2 is equal to 1, and we have only two disjoint sets, $S_0$ and $S_1$, and the score fairness function:

$$f(z) = h_\theta(x)\left(\tfrac{1}{p_0}\mathbb{1}_{S_0}(a, y) - \tfrac{1}{p_1}\mathbb{1}_{S_1}(a, y)\right) \tag{10}$$

where $p_0 = \mathbb{P}(S_0)$ and $p_1 = \mathbb{P}(S_1)$. Before presenting the next results, we need to establish notation.

The classifier $h_\theta(x)$ divides the feature space $\mathcal{X}$ into two subspaces: $\mathcal{X}^- := \{x \in \mathcal{X} : h_\theta(x) = 0\}$ and $\mathcal{X}^+ := \{x \in \mathcal{X} : h_\theta(x) = 1\}$ (denoted by $\pm$ to avoid confusion with $S_0$ and $S_1$). The distance from a point $x \in \mathcal{X}$ to these subspaces is defined as $d_-(x) := \inf_{x' \in \mathcal{X}^-} d(x', x)$ and $d_+(x) := \inf_{x' \in \mathcal{X}^+} d(x', x)$. Let $\mathbb{P}_0(.) := \mathbb{P}(. \mid S_0)$ and $\mathbb{P}_1(.) := \mathbb{P}(. \mid S_1)$ represent the conditional distributions given $S_0$ and $S_1$. For $s \in (0, \infty)$ and $i \in \{0, 1\}$, the conditional probability distribution of the distance to the decision boundary for each level of sensitive attributes is given by:

$$G_i^-(s) = \mathbb{P}_i(d_-(x) \leqslant s \mid d_-(x) > 0); \quad G_i^+(s) = \mathbb{P}_i(d_+(x) \leqslant s \mid d_+(x) > 0), \quad i \in \{0, 1\}.$$

The following theorem presents the first result on the fairness regularizer in the $\varepsilon$-WDF setting.

**Theorem 1** ($\varepsilon$-**WDF Regularizer:** $q = \infty$). *Given that Assumptions (i), (iii), and (iv) hold, and the fairness score function is defined as in Eq. 10, the corresponding regularizer for $q = \infty$ is given by:*

$$\mathcal{S}_{\delta,\infty}(\mathbb{P}, \theta) = \mathbb{P}_0(\mathcal{X}^-)G_0^+(\delta) + \mathbb{P}_1(\mathcal{X}^+)G_1^-(\delta); \quad \mathcal{I}_{\delta,\infty}(\mathbb{P}, \theta) = \mathbb{P}_0(\mathcal{X}^+)G_0^-(\delta) + \mathbb{P}_1(\mathcal{X}^-)G_1^+(\delta) \tag{11}$$

By Thm. 1, when $q = \infty$ worst-case perturbations move any point by at most $\delta$, so violations are governed by the probability mass within a $\delta$-neighborhood of the decision boundary. We thus simplify (11) by upper-bounding these probabilities with the measure of this $\delta$-margin band in the following.

**Corollary 1** (**Simplified $\varepsilon$-WDF Condition**). *Let $\mathrm{dist}(x, S) = \inf_{x' \in S} d(x, x')$ (distance $d$ from Assumption (iv)). Under Assumptions of Thm. 1, $h_\theta$ satisfies $\varepsilon$-WDF if:*

$$\frac{1}{\min\left(\mathbb{P}(S_0), \mathbb{P}(S_1)\right)}\,\mathbb{P}\left(\mathrm{dist}(\mathbf{X}, \mathcal{L}_\theta) < \delta\right) + |\mathcal{F}(\mathbb{P}, \theta)| \leqslant \varepsilon. \tag{12}$$

Corollary 1 demonstrates that when the minority constitutes a small percentage of the population, achieving $\varepsilon$-WDF becomes significantly more challenging. To conclude this section, we present the regularizers for $q \neq \infty$.

**Theorem 2** ($\varepsilon$-**WDF Regularizer:** $q \neq \infty$). *With Theorem 1 assumptions, for $q \in [1, \infty)$ we have:*

$$\mathcal{S}_{\delta,q} = \inf_{\lambda \geqslant 0} \left\{ \lambda \delta^q + \mathbb{P}_0(\mathcal{X}^-) \int_0^{s_0} (1 - p_0 \lambda s^q) \, \mathrm{d}G_0^+(s) + \mathbb{P}_1(\mathcal{X}^+) \int_0^{s_1} (1 - p_1 \lambda s^q) \, \mathrm{d}G_1^-(s) \right\} \quad (13)$$

$$\mathcal{I}_{\delta,q} = \sup_{\lambda \geqslant 0} \left\{ -\lambda \delta^q + \mathbb{P}_0(\mathcal{X}^+) \int_0^{s_0} (1 - p_0 \lambda s^q) \, \mathrm{d}G_0^-(s) + \mathbb{P}_1(\mathcal{X}^-) \int_0^{s_1} (1 - p_1 \lambda s^q) \, \mathrm{d}G_1^+(s) \right\} \quad (14)$$

*where $s_0 = (p_0 \lambda)^{-1/q}, s_1 = (p_1 \lambda)^{-1/q}$.*

# 4 Finite-Sample Estimation of Fairness Regularizer

In this section, our goal is to estimate the upward/downward regularizers $\mathcal{S}_{\delta,q}(\mathbb{P}^N, \theta)$ and $\mathcal{I}_{\delta,q}(\mathbb{P}^N, \theta)$ using $N$ observations. We begin by presenting an efficient algorithm for estimating the fairness regularizer.

**Theorem 3** (**Fairness Regularizer Linear Programs**). *Let the assumptions of Theorem 1 hold, $\hat{p}_0 = \mathbb{P}^N(S_0), \hat{p}_1 = \mathbb{P}^N(S_1)$ and the coefficients $(\omega_i, d_i)$ and $\widehat{G}^+, \widehat{G}^-$ be defined as:*

$$(\omega_i, d_i) = \begin{cases} (\hat{p}_0^{-1}, d_+(x_i)) & \text{if } z_i \in \mathcal{X}^- \times S_0, \\ (\hat{p}_1^{-1}, d_-(x_i)) & \text{if } z_i \in \mathcal{X}^+ \times S_1, \\ (0, +\infty) & \text{otherwise} \end{cases} \quad \begin{cases} \widehat{G}^+(\delta) = \hat{p}_0^{-1} \frac{1}{N} \# \{z_i \in \mathcal{X}^- \times S_0 : d_+(x_i) \leqslant \delta\} \\ \widehat{G}^-(\delta) = \hat{p}_1^{-1} \frac{1}{N} \# \{z_i \in \mathcal{X}^+ \times S_1 : d_-(x_i) \leqslant \delta\} \end{cases}$$

*Then, the unfairness score is given by the following linear program:*

$$\mathcal{S}_{\delta,q}(\mathbb{P}^N, \theta) = \begin{cases} \max_{\xi \in [0,1]^N} \left\{ \dfrac{1}{N} \sum_{i \in [N]} \omega_i \xi_i : \dfrac{1}{N} \sum_{i \in [N]} d_i^q \xi_i \leqslant \delta^q \right\} & q \in [1, \infty) \\ \widehat{G}^+(\delta) + \widehat{G}^-(\delta) & q = \infty. \end{cases} \quad (15)$$

*To derive $\mathcal{I}_{\delta,q}(\mathbb{P}^N, \theta)$, swap the indices 0 and 1 in the coefficients and expressions given above.*

Theorem 3 indicates that evaluating the quantity $\mathcal{S}_{\delta,q}(\mathbb{P}^N, \theta)$ is equivalent to solving a continuous knapsack problem Papadimitriou & Steiglitz (1998) in $N$ variables. This optimization problem admits a greedy solution that runs in $O(N \log N)$ time. The main challenge, however, lies in computing the distance from a point to the classifier's decision boundary under the $\ell_q$ norm. To compute the projection $x^*$ of an arbitrary point $x$ onto the boundary $\mathcal{L}_\theta$, one must solve the system of equations:

$$\begin{cases} g_\theta(y) = 0, \\ G_q(x - y) \times \nabla g_\theta(y) = 0 \end{cases} \iff F(y, \lambda) = \begin{pmatrix} G_q(x - y) + \lambda \nabla g_\theta(y) \\ g_\theta(y) \end{pmatrix} = 0, \quad (y, \lambda) \in \mathbb{R}^d \times \mathbb{R}$$

where $G_q(v) := (|v_1|^{q-2} v_1, \ldots, |v_n|^{q-2} v_n)^\top$. For a small number of closest-point queries, Newton-like projection methods Saye (2014) are effective. When $N$ is large, the Fast Sweeping method Wong & Leung (2016), which has linear complexity in the grid size ($O(N_{\mathrm{grid}})$), becomes more efficient. Alternatively, one may solve the static Eikonal PDE $\|\nabla \psi(\mathbf{x})\|_{q*} = 1, \quad \psi|_{\phi=0} = 0$.

The Newton-KKT scheme thus scales linearly with the number of points, has the same $O(d^3)$ per-point algebraic cost as the Euclidean solver, and retains rapid quadratic convergence-making it attractive for scenarios requiring only a handful of closest-point computations. By integrating the Newton-KKT method for distance computation with the greedy knapsack algorithm for worst-case selection, we achieve an efficient Algorithm 1 for computing the fairness regularizer. An alternative version of the DRUNE algorithm that incorporates the Fast Sweeping method appears in Algorithm 2.

In very high-dimensional feature spaces, however, the per-point $O(d^3)$ cost of the Newton–KKT projection may become prohibitive, even though it enjoys fast quadratic convergence. To handle this regime, Section 5 derives first-order, margin-based bounds (Proposition 6; Theorems 5–6) that provide closed-form upper and lower estimates of $\mathcal{S}_{\delta,q}(\mathbb{P}, \theta)$ and $\mathcal{I}_{\delta,q}(\mathbb{P}, \theta)$, which scale as $O(\delta^q)$ under mild smoothness assumptions on the conditional distributions. These analytic bounds can be evaluated without solving any projection problem in $\mathbb{R}^d$ and thus serve as a lightweight surrogate constraint when DRUNE is used inside

large-scale training or auditing pipelines, preserving the robustness guarantees of $\varepsilon$-WDF while avoiding repeated $O(d^3)$ distance computations.

---

**Algorithm 1** Distributionally Robust Unfairness Estimator (DRUNE)

---

**Require:** $\{(x_i, a_i, y_i)\}_{i=1}^N$, $g_\theta$, $\delta > 0$, tolerances $\varepsilon_y, \varepsilon_g$, $K_{\max}$, $\{\omega_i\}$, $q > 1$, init. $(y^{(0)}, \lambda^{(0)})$
**Ensure:** $\{\xi_i\} \subset [0,1]$ solving $\max \frac{1}{N} \sum \omega_i \xi_i$ s.t. $\frac{1}{N} \sum d_i^q \xi_i \leqslant \delta^q$

1: **Stage 1:** Compute $d_i = \text{dist}_q(x_i, \mathcal{L}_\theta)$ via Newton–KKT
2: **for** $i = 1, \ldots, N$ **do**
3:    Initialize $k \leftarrow 0$, $(y_i, \lambda_i) \leftarrow (y_i^{(0)}, \lambda_i^{(0)})$
4:    **while** $k < K_{\max}$ **and** $(\|\delta y\| \geqslant \varepsilon_y \ \vee \ |r_g| \geqslant \varepsilon_g)$ **do**
5:       $v \leftarrow x_i - y_i$, $r_y \leftarrow G_q(v) + \lambda_i \nabla g_\theta(y_i)$, $r_g \leftarrow g_\theta(y_i)$
6:       $W_q \leftarrow \text{diag}((q-1)|v_j|^{q-2})$, $J \leftarrow \begin{bmatrix} -W_q + \lambda_i \nabla^2 g_\theta & \nabla g_\theta \\ \nabla g_\theta^\top & 0 \end{bmatrix}$
7:       Solve $J[\delta y; \delta \lambda] = -[r_y; r_g]$
8:       Update $y_i \mathrel{+}= \delta y$, $\lambda_i \mathrel{+}= \delta \lambda$, $k \mathrel{+}= 1$
9:    **end while**
10:   Set $d_i \leftarrow \|x_i - y_i\|_q$
11: **end for**
12: **Stage 2:** Greedy fractional knapsack on items with cost $c_i = d_i^q$, value $\omega_i$
13: $C \leftarrow N\delta^q$, $\xi_i \leftarrow 0$, $r_i \leftarrow \omega_i / c_i$, $\{(k)\} \leftarrow$ sort desc. $r$
14: **for** $k = 1, \ldots, N$ **while** $C > 0$ **do**
15:   **if** $c_{(k)} \leqslant C$ **then**
16:      $\xi_{(k)} \leftarrow 1$, $C \leftarrow C - c_{(k)}$
17:   **else**
18:      $\xi_{(k)} \leftarrow C/c_{(k)}$, $C \leftarrow 0$
19:   **end if**
20: **end for**
21: **return** $\{\xi_i\}$, $\frac{1}{N} \sum_i \omega_i \xi_i$

---

In practice, fairness audits and training rely on finite samples. We must therefore ensure that the empirical Wasserstein-robust fairness we compute is not a sampling artifact but a valid certificate for the unknown deployment distribution. Building on universal generalization results for $\varepsilon$-WDF (e.g., Le & Malick (2024)), the next theorem provides a finite-sample guarantee: with high probability over the draw of the data, the worst-case fairness estimated from the sample upper-bounds the true worst-case disparity under shifts within an $\varepsilon$-Wasserstein ball. Before stating it, we define the distance-to-boundary expectations constant $\rho_0$ under the true probability as follows:

$$\rho_0 := \inf_{\theta \in \Theta} \{ \mathbb{E}_{x \sim \mathbb{P}_0} [d_+^q(x)] + \mathbb{E}_{x \sim \mathbb{P}_1} [d_-^q(x)] \} \tag{16}$$

**Theorem 4 (Finite Sample Guarantee for $\varepsilon$-WDF under Distribution Shift).** *Given that Assumptions (i)- (iv) hold, and the fairness score function is defined as in Eq. 10. Suppose $\rho_0 > 0$. Then there exists a constants $\alpha$ and $\beta$ depending on accuracy level $\sigma$, the dimension $K$ and diameter $D$ of the parameter space, such that whenever $N > \max(\frac{16(\alpha+\beta)^2}{\rho_0^2}, \frac{\alpha^2}{\delta^2})$, we have, with probability at least $1 - \sigma$, the uniform lower bound:*

$$\sup_{\mathbb{Q} \in \mathcal{B}_\delta(\mathbb{P}^N)} \mathbb{E}_{z \sim \mathbb{Q}} [f(z)] \geqslant \mathbb{E}_{z \sim \mathbb{P}} [f(z)] \quad \text{for all } \theta \in \Theta.$$

Before using $\varepsilon$-WDF in audits, generalization alone (Thm.4) is not enough, so we must also calibrate how conservative the empirical worst-case estimate is. The next proposition quantifies the *excess fairness* of $\varepsilon$-WDF—how much larger the empirical worst-case disparity can be than its population counterpart—and links this gap to sample size and the Wasserstein radius, yielding a practical calibration rule.

**Proposition 5 (Excess Fairness for $\varepsilon$-WDF).** *Under the assumptions of Theorem 4, let $\alpha$ be as defined there, and let $\rho_0 > 0$ and $\delta < \rho_0/4$. If $N > \max\left( \frac{16\alpha^2}{\rho_0^2}, \frac{\alpha^2}{(\rho_0/4 - \delta)^2} \right)$, then with probability at least $1 - \sigma$,*

$$\sup_{\mathbb{Q} \in \mathcal{B}_\delta(\mathbb{P}^N)} \mathbb{E}_{z \sim \mathbb{Q}} [f(z)] \leqslant \sup_{\mathbb{Q} \in \mathcal{B}_{\delta + \alpha/\sqrt{N}}(\mathbb{P})} \mathbb{E}_{z \sim \mathbb{Q}} [f(z)] \quad \text{for all } \theta \in \Theta.$$

*Equivalently, take $\delta_N = \delta + \alpha/\sqrt{N}$ to upper-bound the population worst-case by the empirical one.*

## 5 First-Order Estimation of Fairness Regularizer

In Section 3, we observed that the effectiveness of the fairness regularizer hinges critically on the function $G_i^{\pm}$. In this section, we ask: if we impose assumptions on the support and derivatives of $G_i^{\pm}$, can we derive sharper bounds? Before proceeding, we introduce the necessary definitions.

The worst-case behavior depends on the distance between $\text{supp}(\mathbb{P}) \cap \mathcal{X}^{\pm}$ and the boundary of $\mathcal{L}_\theta$. More precisely, we define the margin. $s_i^{\pm} = \inf\{s > 0 : G_i^{\pm}(s) > 0\}$, $i \in \{0,1\}$, which represents the minimal distance between $\text{supp}(\mathbb{P}) \cap \mathcal{X}^{\pm}$ and the boundary of $\mathcal{L}_\theta$. Under Assumption (iii), the derivative of $G_i^{\pm}$ is well-defined for $s_0 \in (0,\infty)$:

$$g_i^{\pm}(s_0) := \frac{1}{\mathbb{P}_i(\mathcal{X}^{\pm})} \lim_{s \downarrow s_0} \frac{\mathbb{P}_i(s_0 \leqslant d_{\mp}(\mathbf{X}) \leqslant s)}{s - s_0}, \quad i \in \{0,1\}$$

Since Theorem 1 gives a closed-form for the fairness regularizer at $q = \infty$, we focus on $q \in [1,\infty)$. The following proposition shows that, under a positive margin, the regularizer scales as $O(\delta^q)$.

**Proposition 6 (Positive Margin).** *Let $\lambda^*$ be the solution of the optimization problem 13. With Assumptions (i)- (iv) and for $q \in [1,\infty)$, if there exists $s_0^+, s_1^- > 0$ then we have:*

$$\lambda^* \delta^q \leqslant \mathcal{S}_{\delta,q}(\mathbb{P},\theta) \leqslant \frac{\delta^q}{\min(p_0 s_0^{+q}, p_1 s_1^{-q})}, \qquad \lambda^* \delta^q \leqslant \mathcal{I}_{\delta,q}(\mathbb{P},\theta) \leqslant \frac{\delta^q}{\min(p_1 s_0^{-q}, p_0 s_1^{+q})},$$

The lower bound in Proposition 6 depends on $\lambda^*$, so estimating $\lambda^*$ requires additional assumptions.

**Assumption.** There exists a constant $\upsilon > 0$ such that for each $i \in \{0,1\}$, the functions $G_i^{\pm}$ are differentiable on $s \in [0,\upsilon]$ with $G_i^{\pm}(s) > 0$ and their derivatives $g_i^{\pm}$ satisfy the $L_i$-Lipschitz condition:

$$\left|g_i^{\pm}(s_1) - g_i^{\pm}(s_2)\right| \leqslant L_i |s_1 - s_2|, \forall s_1, s_2 \in [0,\upsilon] \tag{$v$}$$

Any probability distribution $\mathbb{P}$ whose density lies in $C^{0,1}(\mathbb{R}^d)$ that has both continuity and a global Lipschitz-like property like a Gaussian distribution satisfies Assumption $v$. Under this assumption, we derive a lower bound for the fairness regularizer. The analogous expression for $\mathcal{I}_{\delta,q}(\mathbb{P},\theta)$ follows by swapping the index $i$ and is therefore omitted.

**Theorem 5 (Positive Margin and Lipschitz).** *With assumptions of proposition 6 and (v), there exists a positive constant $\delta_0$ that dependent on $(\mathbb{P},q)$ such that for any $\delta < \delta_0$:*

$$\mathcal{S}_{\delta,q}(\mathbb{P},\theta) \geqslant \frac{\delta^q}{\min(p_0 s_0^{+q}, p_1 s_1^{-q})} - \frac{2q\delta^{2q}}{\min(p_0 s_0^{+2q+1} g_0^{-}(s_0^+)\mathbb{P}_0(\mathcal{X}^{-}), p_1 s_1^{-2q+1} g_1^{+}(s_1^-)\mathbb{P}_1(\mathcal{X}^{+}))}.$$

With positive margins, the boundary is buffered, so small Wasserstein shifts can only touch a thin shell near it—making the worst-case unfairness grow like $\delta^q$ with only a tiny $\delta^{2q}$ correction from boundary-density slopes. By contrast, when margins vanish, the buffer disappears and even infinitesimal shifts move mass across the boundary, yielding a slower $\delta^{\frac{q}{q+1}}$ growth; Theorem 6 formalizes this with a two-term lower bound.

**Theorem 6 (Zero Margin and Lipschitz).** *Let $q \in [1,\infty)$. Suppose $s_0^+, s_1^- = 0$, and Assumptions (i)-(v) hold. There exists constants $\delta_0, C$ depending on $(\mathbb{P},q)$ such that for any $\delta < \delta_0$,*

$$\mathcal{S}_{\delta,q}(\mathbb{P},\theta) \geqslant (q+1)^{\frac{1}{q+1}} \left(\mathbb{P}_0(\mathcal{X}^{-}) g_0^{+}(0) p_0^{-\frac{1}{q}} + \mathbb{P}_1(\mathcal{X}^{+}) g_1^{-}(0) p_1^{-\frac{1}{q}}\right)^{\frac{q}{q+1}} \delta^{\frac{q}{q+1}} - C\delta^{\frac{2q}{q+1}}$$

*where $C = \zeta \left(\mathbb{P}_1(\mathcal{X}^{+}) L_0 p_0^{-\frac{2}{q}} + \mathbb{P}_0(\mathcal{X}^{-}) L_1 p_1^{-\frac{2}{q}}\right) \left(\mathbb{P}_0(\mathcal{X}^{-}) g_0^{+}(0) p_0^{-\frac{1}{q}} + \mathbb{P}_1(\mathcal{X}^{+}) g_1^{-}(0) p_1^{-\frac{1}{q}}\right)^{\frac{-2}{q+1}}$ and $\zeta = 2^{\frac{2-q}{q}} \frac{q}{(q+2)} (q+1)^{\frac{2}{q+1}}$.*

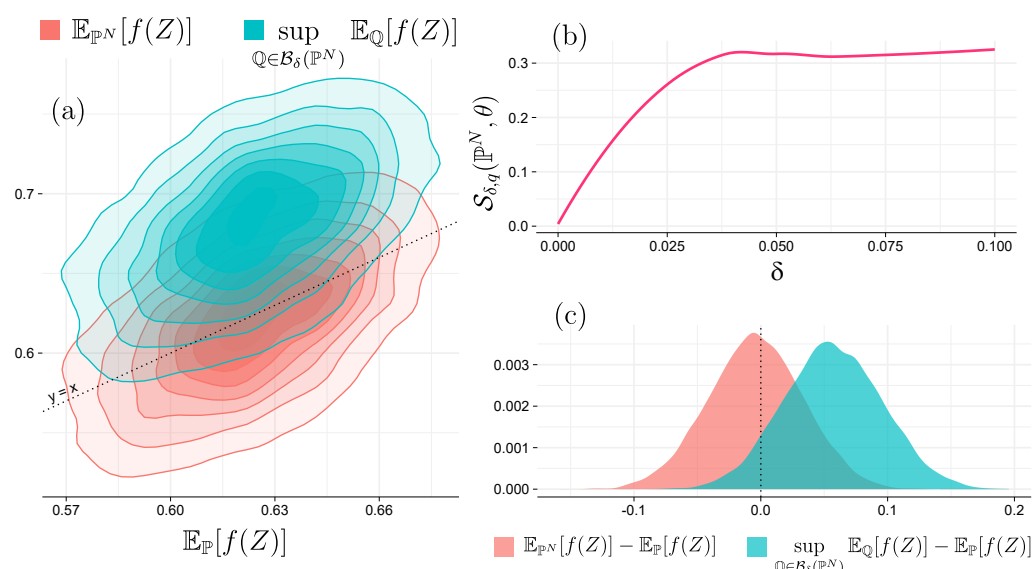

**Figure 2:** (a) Density plot comparing empirical and worst-case fairness estimates $(\hat{f}_\delta)$ against true fairness values across 10,000 SVM models ($\delta = 0.01$, $q = 2$). (b) Fairness regularizer $\mathcal{S}_{\delta,q}$ approaching zero as uncertainty parameter $\delta$ decreases. (c) Direct visualization of the gap between worst-case fairness and true fairness values.

## 6 NUMERICAL STUDIES

We empirically evaluate our framework on eight real-world datasets and four classifier families (details in Appx. C, Tables 1-2). Our primary objective is to assess the out-of-sample sensitivity of fairness metrics to distributional shifts and model choices. To demonstrate the widespread fragility of common fairness notions, we use the following benchmarks: Adult (U.S. Census income prediction) Asuncion & Newman (1996), ACS Income (American Community Survey) U.S. Census Bureau (2023), Bank Marketing Moro et al. (2014), Heritage Health (insurance claims) Prize (2014), MEPS (Medical Expenditure Panel Survey) Agency for Healthcare Research and Quality (AHRQ) (2024), HELOC (home equity line of credit applications) Mae (2023), CelebA (celebrity face attributes) Liu et al. (2015), and Law School Admissions Law School Admission Council (2002). Binary sensitive-attribute and label definitions for each dataset appear in Appx. C (Table 1).

We encode each dataset with a binary sensitive attribute (e.g., gender, race, age group) and a binary target, train diverse classifiers (logistic regression; linear/nonlinear SVM; MLP), and assess group fairness via Demographic Parity, Equal Opportunity, and Equalized Odds (hyperparameters and settings in Appx. C, Tables 2–3).

**Experiment 1: sampling fragility.** Each trial uses subsamples of size 1,000 and is repeated 10,000 times. Scenario 1: we draw 1,000-point subsamples, fit a classifier on each, and compute fairness metrics (red band in Fig. 1). Scenario 2: we train a single classifier once, then repeatedly sample 1,000 points and recompute the metrics (blue band in Fig. 1). Fairness measures are highly sensitive to the input sample, with large variability on datasets such as HELOC. Complete results are in Fig. 4 (Scenario 1) and Fig. 5 (Scenario 2); numeric summaries appear in Appx. C.

**Experiment 2: empirical vs. worst-case vs. true.** On HELOC, we repeat the following 10,000 times: draw 1,000 samples, train an SVM, set $\delta = 0.01$ and $q = 2$, then compute (i) empirical fairness $\mathbb{E}_{\mathbb{P}^N}[f(Z)]$, (ii) true fairness $\mathbb{E}_{\mathbb{P}}[f(Z)]$ (operationalized by evaluating under $\mathbb{P}$ on the full dataset), and (iii) worst-case fairness $\sup_{\mathbb{Q} \in \mathcal{B}_\delta(\mathbb{P}^N)} \mathbb{E}_{\mathbb{Q}}[f(Z)]$ via the DRUNE Algorithm 1. Fig. 2(a) plots true fairness (x-axis) against empirical and worst-case estimates (y-axis); consistent with our theoretical guarantees, worst-case fairness typically exceeds true fairness with high probability. Fig. 2(c) visualizes the gap as *worst-case − true*. Fig. 2(b) shows $\mathcal{S}_{\delta,q}(\mathbb{P}^N, \theta) \to 0$ as $\delta \to 0$.

## 7 DISCUSSION

We introduced $\varepsilon$-WDF, which certifies worst-case group fairness over a Wasserstein ball centered at the empirical distribution $\mathbb{P}^N$. When a classifier satisfies the $\varepsilon$-WDF constraint on $\mathbb{P}^N$, our theory shows that certificate transfers to the true distribution $\mathbb{P}$ up to a small radius inflation $\delta \mapsto \delta + \alpha/\sqrt{N}$ (Thm. 4; Prop. 5), and the worst-case bound dominates the non-robust fairness measured at $\mathbb{P}$.

Our goal was not to design a new fair-learning algorithm, but to quantify a robust fairness constraint that can be plugged into existing pipelines. In practice, our DRUNE estimator (Alg. 1) computes the $\varepsilon$-WDF regularizer efficiently and can be used for audits or as a constraint during training.

Although our theoretical framework is presented for binary classifiers, it is flexible and can be extended to multi-class settings. While some research addresses the challenge of non-continuity in fairness notions using relaxation techniques such as softmax, we avoid these approaches because they alter the original definition of fairness. Finally, the theoretical estimation in Section 5 suggests that improving the finite-sample rate is possible, which we leave as a direction for future work.

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

## A  THEORETICAL SUPPLEMENT

This section provides supplementary results, illustrative examples, and extended explanations that could not be incorporated into the main text due to space limitations.

### A.1  GENERIC NOTION OF FAIRNESS

The general group fairness formulation in Eq. 3 encompasses a wide range of fairness metrics by appropriately specifying the sets $S_0^i, S_1^i$ and the corresponding transformation $\varphi(\cdot, \cdot)$. To illustrate the flexibility and generality of this framework, we present two concrete examples—demographic parity and equalized odds—and show how each can be expressed as a special case of Eq. 3 with suitable choices of sets and mappings.

**Example 2** (**Demographic Parity**). *A classifier satisfies demographic parity if its positive prediction rate is equal across all sensitive groups* $\mathbf{A} \in \{1, \ldots, k\}$:

$$\big|\mathbb{P}(h_\theta(\mathbf{X}) = 1 \mid \mathbf{A} = a) - \mathbb{P}(h_\theta(\mathbf{X}) = 1 \mid \mathbf{A} = b)\big| \leqslant \varepsilon \quad \text{for all } a, b \in \{1, \ldots, k\}.$$

*Define*

$$S_a = \{z \in \mathcal{Z} : \mathbf{A} = a\}, \quad a = 1, \ldots, k.$$

*Then, each pairwise constraint can be written as*

$$\Big\|\mathbb{E}_{z \sim \mathbb{P}}\Big[h_\theta(x)\Big(\tfrac{\mathbb{1}_{S_a}(a,y)}{\mathbb{P}(S_a)} - \tfrac{\mathbb{1}_{S_b}(a,y)}{\mathbb{P}(S_b)}\Big)\Big]\Big\|_\infty \leqslant \varepsilon, \quad a, b \in \{1, \ldots, k\}.$$

*Let*

$$\mathbf{U}(a,y) = (\mathbb{1}_{S_1}(a,y), \mathbb{1}_{S_2}(a,y), \ldots, \mathbb{1}_{S_k}(a,y)) \in \mathbb{R}^k.$$

*By Eq. 2, choose the* $k(k-1)/2$-*dimensional vector*

$$\varphi(\mathbf{U}, \mathbb{E}_\mathbb{P}[\mathbf{U}]) = \Big(\tfrac{\mathbb{1}_{S_i}(a,y)}{\mathbb{E}_\mathbb{P}[\mathbb{1}_{S_i}]} - \tfrac{\mathbb{1}_{S_j}(a,y)}{\mathbb{E}_\mathbb{P}[\mathbb{1}_{S_j}]}\Big)_{i,j \in [k]: \ i < j}.$$

*Hence, demographic parity is equivalent*

$$\big\|\mathbb{E}_\mathbb{P}\big[h_\theta(\mathbf{X})\,\varphi(\mathbf{U}, \mathbb{E}_\mathbb{P}[\mathbf{U}])\big]\big\|_\infty \leqslant \varepsilon.$$

### A.2  DUAL FORMULATION OF WASSERSTEIN DISTRIBUTIONAL FAIRNESS.

To obtain a tractable formulation of $\varepsilon$-WDF, it is necessary to adapt the strong duality theorem to the specific cost function described in Assumption (iv). The following proposition provides the explicit formulation of strong duality tailored to our setting.

**Proposition 7** (**Strong Duality Theorem**). *Let $\psi$ be upper semi-continuous $\psi : \mathcal{Z} \to \mathbb{R}$ and assumption (iv) satisfies, then*

$$\sup_{\mathbb{Q} \in \mathcal{B}_\delta(\mathbb{P})} \Big\{\mathbb{E}_{z \sim \mathbb{Q}}[\psi(z)]\Big\} = \begin{cases} \inf_{\lambda \geqslant 0}\Big\{\lambda \delta^q + \mathbb{E}_{z \sim \mathbb{P}}[\sup_{x' \in \mathcal{X}} \psi(x',a,y) - \lambda d^q(x,x')]\Big\} & q \in [1, \infty), \\[2mm] \mathbb{E}_{z \sim \mathbb{P}}\Big[\sup_{x': d(x,x') \leqslant \delta} f(x',a,y)\Big] & q = \infty. \end{cases}$$

In DRO, the notion of the worst-case distribution is fundamental, as it identifies the most adverse distribution within a prescribed ambiguity set—often defined by a divergence or Wasserstein distance—from the empirical data. Optimizing over this worst-case distribution ensures that the solution is robust to distributional uncertainty and potential data shifts. Importantly, the structure of the worst-case distribution often admits a closed-form or tractable representation, which facilitates both theoretical analysis and efficient computation. The following proposition characterizes the explicit form of the worst-case distribution in our setting.

**Proposition 8** (**Worst-Case Distribution**). *Suppose the assumption (iv) satisfies and $\psi$ is upper semi-continuous on $\mathcal{Z}$ and satisfies:*

$$\inf\Big\{\lambda \geqslant 0 : \mathbb{E}_{z \sim \mathbb{P}}\Big[\sup_{x' \in \mathcal{X}}\{\psi(x',a,y) - \lambda d^q(x',x)\}\Big] < \infty\Big\} < \infty. \tag{17}$$

*If $\lambda_*$ is the minimum solution of proposition 7 then, a worst-case distribution $\mathbb{P}^*$ exists, given by:*

*i. For $q = \infty$, there is a $\mathbb{P}$-measurable map $T^* : \mathcal{Z} \to \mathcal{Z}$ such that*

$$T^*(x,a,y) \in \Big\{(\tilde{x},a,y) : \tilde{x} \in \arg\max_{x' \in \mathcal{X}}\{\psi(x',a,y) : d(x',x) \leqslant \delta\}\Big\} \quad \mathbb{P}\text{-a.e.}$$

*Then the worst-case distribution is obtained by $\mathbb{P}^* = T^*_\# \mathbb{P}$.*

*ii. For $q \in [1, \infty)$ and $\lambda^* = 0$, there is a $\mathbb{P}$-measurable map $T^*$ satisfying*

$$T^*(x, a, y) \in \left\{ (\tilde{x}, a, y) : \tilde{x} \in \arg\min_{x' \in \mathcal{X}} \left\{ d(x, x') : x' \in \arg\max_{\tilde{x} \in \mathcal{X}} \psi(\tilde{x}, a, y) \right\} \right\}, \quad \mathbb{P}\text{-}a.e.$$

*In this case worst-case distribution is $\mathbb{P}^* = T^*_{\#}\mathbb{P}$.*

*iii. For $q \in [1, \infty)$ and $\lambda^* > 0$, there are $\mathbb{P}$-measurable maps $T^*$ and $T^-$ such that*

$$T^*(x, a, y) \in \left\{ (\tilde{x}, a, y) : \tilde{x} \in \arg\max_{x' \in \mathcal{X}} \left\{ d(x, x') : x' \in \arg\max_{\tilde{x} \in \mathcal{X}} \psi(\tilde{x}, a, y) - \lambda^* d^q(\tilde{x}, x) \right\} \right\},$$

$$T^-(x, a, y) \in \left\{ (\tilde{x}, a, y) : \tilde{x} \in \arg\min_{x' \in \mathcal{X}} \left\{ d(x, x') : x' \in \arg\max_{\tilde{x} \in \mathcal{X}} \psi(\tilde{x}, a, y) - \lambda^* d^q(\tilde{x}, x) \right\} \right\}.$$

*Define $t^*$ as the largest number in $[0, 1]$ such that:*

$$\delta^q = t^* \mathop{\mathbb{E}}_{z \sim \mathbb{P}} [d^q(T^*(x), x)] + (1 - t^*) \mathop{\mathbb{E}}_{z \sim \mathbb{P}} [d^q(T^-(x), x)].$$

*Then, $\mathbb{P}^* = t^* T^*_{\#}\mathbb{P} + (1 - t^*) T^-_{\#}\mathbb{P}$ is a worst-case distribution.*

Now we are ready to apply the proposition 8 to the formulation of fairness 3. Let $\lambda^*$ be the solution of optimization problems in Theorem 2. To describe the worst-case distribution, let us define the boundary and region sets for each $i \in \{0, 1\}$ (see Fig. 3 for geometric intuition):

$$\mathcal{R}_i^+ := \begin{cases} x \in \mathcal{X}^+ : 0 < d_-(x) \leqslant (p_i \lambda^*)^{\frac{-1}{q}} & q \in [1, \infty), \\ x \in \mathcal{X}^+ : 0 < d_-(x) \leqslant \delta & q = \infty. \end{cases}$$

$$\mathcal{R}_i^- := \begin{cases} x \in \mathcal{X}^- : 0 < d_+(x) \leqslant (p_i \lambda^*)^{\frac{-1}{q}} & q \in [1, \infty), \\ x \in \mathcal{X}^- : 0 < d_+(x) \leqslant \delta & q = \infty. \end{cases}$$

$$\partial_i^+ := \begin{cases} x \in \mathcal{X}^+ : d_-(x) = (p_i \lambda^*)^{\frac{-1}{q}}, & q \in [1, \infty), \\ \varnothing, & q = \infty. \end{cases}$$

$$\partial_i^- := \begin{cases} x \in \mathcal{X}^- : d_+(x) = (p_i \lambda^*)^{\frac{-1}{q}}, & q \in [1, \infty), \\ \varnothing, & q = \infty. \end{cases}$$

In the cases $\lambda^* = 0$, we can set $(p_i \lambda^*)^{\frac{-1}{q}} = \infty$ in above formulation. Let us define two set-valued maps $\mathcal{T}^*, \mathcal{T}^- : \mathcal{Z} \to \mathcal{Z}$ as:

$$\mathcal{T}^*(x, a, y) = \begin{cases} (\mathcal{T}_0^*(x), a, y) & (a, y) \in S_0 \\ (\mathcal{T}_1^*(x), a, y) & (a, y) \in S_1 \end{cases}; \quad \mathcal{T}^-(x, a, y) = \begin{cases} (\mathcal{T}_0^-(x), a, y) & (a, y) \in S_0 \\ (\mathcal{T}_1^-(x), a, y) & (a, y) \in S_1 \end{cases}$$

where:

$$\mathcal{T}_0^*(x) = \begin{cases} x, & x \in \mathcal{X} \backslash \mathcal{R}_0^-, \\ \arg\min_{x' \in \mathcal{X}^+} d(x, x'), & x \in \mathcal{R}_0^- \backslash \partial_0^-, \\ x \cup \arg\min_{x' \in \mathcal{X}^+} d(x, x'), & x \in \partial_0^-, \end{cases}$$

$$\mathcal{T}_1^*(x) = \begin{cases} x, & x \in \mathcal{X} \backslash \mathcal{R}_1^+, \\ \arg\min_{x' \in \mathcal{X}^-} d(x, x'), & x \in \mathcal{R}_1^+ \backslash \partial_1^+, \\ x \cup \arg\min_{x' \in \mathcal{X}^-} d(x, x'), & x \in \partial_1^+, \end{cases}$$

$$\mathcal{T}_0^-(x) = \begin{cases} x, & x \in \mathcal{X} \backslash \mathcal{R}_0^- \cup \partial_0^-, \\ \arg\min_{x' \in \mathcal{X}^+} d(x, x'), & x \in \mathcal{R}_0^- \backslash \partial_0^-. \end{cases}$$

$$\mathcal{T}_1^-(x) = \begin{cases} x, & x \in \mathcal{X} \backslash \mathcal{R}_1^+ \cup \partial_1^+, \\ \arg\min_{x' \in \mathcal{X}^-} d(x, x'), & x \in \mathcal{R}_1^+ \backslash \partial_1^+. \end{cases}$$

Then it follows from Proposition 8, there exist $\mathbb{P}$-measurable transport maps $T^*, T^- : \mathcal{Z} \to \mathcal{Z}$ that are measurable selections of $\mathcal{T}^*$ and $\mathcal{T}^-$, respectively.

**Theorem 7 (Worst-Case Distribution).** *Given that Assumptions (i) and (iv) hold, and the fairness score function is defined as in Eq. 10, then:*

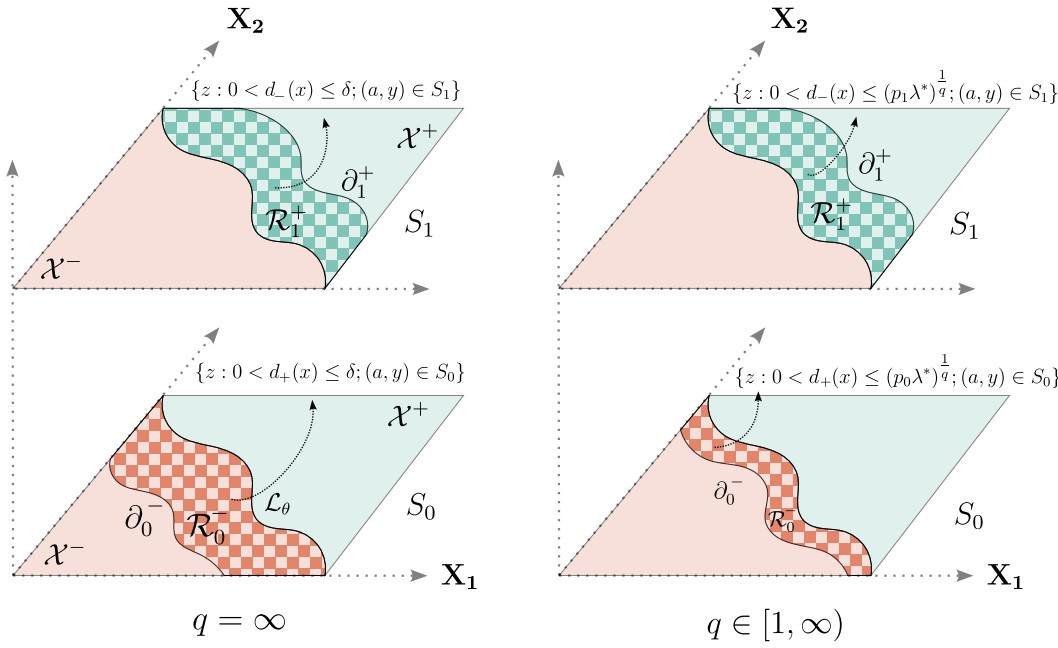

**Figure 3:** Illustration of the boundary and region sets $\mathcal{R}_i^+$ and $\mathcal{R}_i^-$ defined in Eq. 3, corresponding to the worst-case distribution described in Proposition 8. The shaded regions indicate the sets of points within the distance threshold, while the boundaries $\partial_i^+$ and $\partial_i^-$ (for $q \in [1,\infty]$) are shown as level sets of the distance functions.

(i) *When $q = \infty$ and when $q \in [1,\infty)$ with a dual optimizer $\lambda^* = 0$, let $T^*$ be a measurable selection of $\mathcal{T}^*$. Then $\mathbb{P}^* := T_{\#}^* \mathbb{P}$ is a worst-case distribution with probability*

$$\mathbb{P}^*(\mathcal{X}^- \mid S_0) = \mathbb{P}(\mathcal{X}^- \backslash \mathcal{R}_0^- \mid S_0); \quad \mathbb{P}^*(\mathcal{X}^+ \mid S_1) = \mathbb{P}(\mathcal{X}^+ \backslash \mathcal{R}_1^+ \mid S_1)$$

(ii) *When $q \in [1,\infty)$ and all dual optimizers $\lambda^* > 0$, any worst-case transport plan $\pi^* \in \mathcal{P}(\mathcal{Z} \times \mathcal{Z})$ satisfies:*

$$\delta^q = \mathop{\mathbb{E}}_{(z,z') \sim \pi^*} \left[ d^q(z, z') \right]$$

*and if $\mathcal{Z}^* = \mathcal{R}_0^+ \times S_0 \bigcup \mathcal{R}_1^- \times S_1$ then:*

$$\{(z, \mathcal{T}^-(z)) : z \in \mathcal{Z}^*\} \subseteq \mathrm{supp}(\pi^*) \subseteq \{(z, \mathcal{T}^*(z)) : z \in \mathcal{Z}^*\}.$$

*Moreover, there exist $t^* \in [0,1]$ and measurable selections $T^*$ of $\mathcal{T}^*$ and $T^-$ of $\mathcal{T}^-$ such that*

$$\mathbb{P}^* := t^* T_{\#}^* \mathbb{P} + (1-t^*) T_{\#}^- \mathbb{P}$$

*is a worst-case distribution with probability*

$$\mathbb{P}^*(\mathcal{X}^- \mid S_0) = \mathbb{P}(\mathcal{X}^- \backslash \mathcal{R}_0^- \mid S_0) + (1-t^*)\mathbb{P}(\partial_0^- \mid S_0)$$
$$\mathbb{P}^*(\mathcal{X}^+ \mid S_1) = \mathbb{P}(\mathcal{X}^+ \backslash \mathcal{R}_1^+ \mid S_1) + (1-t^*)\mathbb{P}(\partial_1^+ \mid S_1)$$

By applying the Theorem 7 we can calculate the fairness regularizers $\mathcal{S}_{\delta,q}^i(\mathbb{P}, \theta)$ and $\mathcal{I}_{\delta,q}^i(\mathbb{P}, \theta)$.

**Proposition 9.** *With assumption of Theorem 7, there exists $t^* \in [0,1]$ such that:*

$$\mathcal{S}_{\delta,q}(\mathbb{P}, \theta) = \mathbb{P}_0(\mathcal{R}_0^- \backslash \partial_0^-) + (1-t^*)\mathbb{P}_0(\partial_0^-) + \mathbb{P}_1(\mathcal{R}_1^+ \backslash \partial_1^+) + (1-t^*)\mathbb{P}_1(\partial_1^+)$$
$$\mathcal{I}_{\delta,q}(\mathbb{P}, \theta) = \mathbb{P}_1(\mathcal{R}_1^- \backslash \partial_1^-) + (1-t^*)\mathbb{P}_1(\partial_1^-) + \mathbb{P}_0(\mathcal{R}_0^+ \backslash \partial_0^+) + (1-t^*)\mathbb{P}_0(\partial_0^+)$$

Proposition 9 is more general than Theorem 1. In this proposition, we do not require Assumption (iii); therefore, the probability distribution $\mathbb{P}$ may be concentrated on the margins.

To build intuition for the definitions above and to illustrate how distances to the decision boundary, as well as their conditional distributions, can be computed in practice, we present two representative examples. These examples—one for a linear classifier and one for a non-linear kernel classifier—demonstrate how the relevant quantities, such as $d_-(x)$, $d_+(x)$, and the conditional CDFs $G_i^-(s)$ and $G_i^+(s)$, can be explicitly derived or efficiently approximated in common settings.

**Example 3** (**Linear Classifier**). *In the $\ell_q$ feature-space cost, consider the linear SVM, $h_\theta(x) = \mathbb{I}(w^\top x + b \geqslant 0)$, where $\|w\|_{q*} > 0$ and $q, q*$ are conjugate exponents ($1/q* + 1/q = 1$). The distances to the decision boundary are*

$$d_-(x) = \frac{1}{\|w\|_{q*}} \mathbb{I}\left(w^\top x + b > 0\right) \left|w^\top x + b\right|, \quad d_+(x) = \frac{1}{\|w\|_{q*}} \mathbb{I}\left(w^\top x + b < 0\right) \left|w^\top x + b\right|$$

*If we have explicit formulation fo conditional distribution, $\mathbb{P}_0 \sim \mathcal{N}(\mu_0, \Sigma_0)$, then*

$$G_0^-(s) = \mathbb{P}\left(0 < \frac{w^\top \mathbf{X} + b}{\|w\|_{q*}} < s\right) = \varphi\left(\frac{s - \frac{w^\top \mu_0 + b}{\|w\|_{q*}}}{\sqrt{\frac{w^\top \Sigma_0 w}{\|w\|_{q*}^2}}}\right) - \varphi\left(-\frac{\frac{w^\top \mu_0 + b}{\|w\|_{q*}}}{\sqrt{\frac{w^\top \Sigma_0 w}{\|w\|_{q*}^2}}}\right)$$

*where $\varphi(\cdot)$ is the CDF of the standard normal distribution. Similarly, we can calculate another $G_i^\pm(s)$ by the same derivation.*

**Example 4** (**RBF Kernel Classifier**). *In the $\ell_2$ feature-space cost, consider an RBF-kernel SVM with decision function*

$$g_\theta(x) = \sum_{i=1}^N \alpha_i\, y_i \exp\left(-\gamma \|x - x_i\|_2^2\right) + b, \qquad h_\theta(x) = \mathbb{I}\left(g_\theta(x) \geqslant 0\right).$$

*The exact distance from $x$ to the nonlinear boundary $\mathcal{L}_\theta = \{x : g_\theta(x) = 0\}$ is intractable, but a first-order approximation follows from a local linearisation of $g_\theta$:*

$$d_-(x) \approx \frac{\mathbb{I}(g_\theta(x) > 0)\, |g_\theta(x)|}{\|\nabla_x g_\theta(x)\|_2}, \qquad d_+(x) \approx \frac{\mathbb{I}(g_\theta(x) < 0)\, |g_\theta(x)|}{\|\nabla_x g_\theta(x)\|_2},$$

*where the gradient has the closed form*

$$\nabla_x g_\theta(x) = -2\gamma \sum_{i=1}^N \alpha_i\, y_i \exp\left(-\gamma \|x - x_i\|_2^2\right)(x - x_i).$$

*Because both $g_\theta(x)$ and $\nabla_x g_\theta(x)$ are explicit, the distance estimate is available in closed form.*

A central issue in the dual formulation is to determine whether the optimal dual variable $\lambda*$ vanishes. The next proposition pinpoints the conditions under which $\lambda*$ is strictly positive.

**Proposition 10** (**Optimal Dual Solution Behavior**). *Let $\delta_\mathcal{S}$ and $\delta_\mathcal{I}$ be the constants:*

$$\delta_\mathcal{S} := \left(p_0 \mathbb{E}_{\mathbb{P}_0}\left[(1 - h_\theta(x)) d_+^q(x)\right] + p_1 \mathbb{E}_{\mathbb{P}_1}\left[h_\theta(x) d_-^q(x)\right]\right)^{\frac{1}{q}},$$

$$\delta_\mathcal{I} := \left(p_0 \mathbb{E}_{\mathbb{P}_0}\left[h_\theta(x) d_-^q(x)\right] + p_1 \mathbb{E}_{\mathbb{P}_1}\left[(1 - h_\theta(x)) d_+^q(x)\right]\right)^{\frac{1}{q}}.$$

*Consider the optimization problem equation 13 with associated dual variable $\lambda$, then*

- *If $\delta \geqslant \delta_\mathcal{S}$, the optimal dual solution is $\lambda* = 0$.*
- *If $\delta < \delta_\mathcal{S}$, the optimal dual solution satisfies $\lambda* > 0$.*

*An entirely analogous statement holds for $\delta_\mathcal{I}$ in problem equation 14.*

### A.3 Reformulation of Wasserstein Distributional Fairness

The $\varepsilon$-WDF objective admits equivalent formulations via various conjugate representations. The next proposition gives its characterization through the concave conjugate.

**Theorem 8** ($\varepsilon$-**WDF as Concave Conjugate**). *Let $\Psi_\mathcal{S}$ and $\Psi_\mathcal{I}$ denote the functions defined below:*

$$\Psi_\mathcal{S}(t) := \mathbb{E}_{x \sim \mathbb{P}}\left[\mathbb{1}_{\mathcal{X}^-}(x) \min(d_+^q(x), p_0^{-1} t) + \mathbb{1}_{\mathcal{X}^+}(x) \min(d_-^q(x), p_1^{-1} t)\right]$$

$$\Psi_\mathcal{I}(t) := \mathbb{E}_{x \sim \mathbb{P}}\left[p_0^{-1} \mathbb{1}_{\mathcal{X}^-}(x) \min(p_0 d_-^q(x), t) + p_1^{-1} \mathbb{1}_{\mathcal{X}^+}(x) \min(p_1 d_+^q(x), t)\right]$$

*For any function $\Psi(t)$, define its concave conjugate by $\Psi*(s) := \inf_{t>0}\{ts - \Psi(t)\}$. Then $h_\theta$ satisfies $\varepsilon$-WDF if and only if:*

$$\Psi_\mathcal{S}^*(1 - \varepsilon) \geqslant \delta^q \quad \text{and} \quad \Psi_\mathcal{I}^*(1 - \varepsilon) \geqslant \delta^q \tag{18}$$

### A.4 Finite Sample Guarantee for Wasserstein Distributional Fairness.

The concentration theorem in DRO provides probabilistic guarantees that the true data-generating distribution lies within a Wasserstein ambiguity set constructed from empirical data. The Proposition highlights the trade-off between robustness (via $\delta$) and sample complexity, particularly in high-dimensional settings.

**Proposition 11** (**Concentration of Empirical Measures**). *Let $\mathbb{P} \in \mathcal{P}(\mathcal{Z})$ be compactly supported and satisfy Assumption (iv), and define the product measure $\mathbb{P}^{\otimes} = \mathbb{P} \otimes \mathbb{P} \otimes \dots$ on $\mathcal{Z}^N$. Then for any $N \geqslant 1$ and confidence level $1 - \varepsilon$ with $\varepsilon \in (0,1)$, there exists $\delta = \delta(N, \varepsilon)$ such that if:*

$$\delta(N, \varepsilon) \lesssim \left( N \ln \left( C \varepsilon^{-1} \right) \right)^{-\frac{1}{\max\{d, 2q\}}} \implies \mathbb{P}^{\otimes} \left( \mathbb{P} \in \mathcal{B}_p(\mathbb{P}^N, \delta) \right) \geqslant 1 - \varepsilon, \tag{19}$$

*where $C$ is a constant depending only on $\mathbb{P}$ and the metric dimension $d$.*

---

**Algorithm 2** DRUNE Algorithm with Sweeping Method

---

**Require:** $\{(x_i, a_i, y_i)\}_{i=1}^N$, $g_\theta$, $\delta > 0$, tolerances $\varepsilon_\phi$, $K_{\max}$, $\{\omega_i\}$, $q \geqslant 1$

**Ensure:** $\{\xi_i\} \subset [0,1]$ solving $\displaystyle\max_{\xi \in [0,1]^N} \frac{1}{N} \sum_{i=1}^N \omega_i \xi_i \text{s.t.} \frac{1}{N} \sum_{i=1}^N d_i^q \xi_i \leqslant \delta^q$

1: **Stage 1:** Fast Sweeping distance to the constraint set $\mathcal{L}_\theta := \{x \mid g_\theta(x) = 0\}$
2:        *Solve $|\nabla \phi(x)| = 1$ with boundary $\phi(x) = 0$ on $\mathcal{L}_\theta$*
3: Construct a Cartesian grid $\mathcal{G} \subset \mathbb{R}^d$ with spacing $h$
4: Initialize $\phi(x) \leftarrow 0$ for $x \in \mathcal{L}_\theta$ ($g_\theta(x) = 0$); otherwise $\phi(x) \leftarrow \infty$
5: **for** $k = 1, \dots, K_{\max}$ **do**
6:      *Eight sweeping orders in 2-D (or $2^d$ in d-D)*
7:    **for** each sweep direction $s = 1, \dots, 2^d$ **do**
8:      **for** grid point $x \in \mathcal{G}$ in order $s$ **do**
9:        Compute tentative value $\tilde{\phi}(x)$ by the upwind discretizations of $|\nabla \phi| = 1$
10:        $\phi(x) \leftarrow \min\left(\phi(x), \tilde{\phi}(x)\right)$
11:      **end for**
12:    **end for**
13:    **if** $\displaystyle\max_{x \in \mathcal{G}} \left| \phi^{(k)}(x) - \phi^{(k-1)}(x) \right| < \varepsilon_\phi$ **then**
14:      **break**
15:    **end if**
16: **end for**
17: **for** $i = 1, \dots, N$ **do**
18:    $d_i \leftarrow \left| \phi(x_i) \right|$     *// (for general $q$ one may apply $\|x_i - y\|_q$ post-correction)*
19: **end for**
20: **Stage 2:** Greedy fractional knapsack on items with cost $c_i = d_i^q$, value $\omega_i$
21: $C \leftarrow N\delta^q$, $\xi_i \leftarrow 0$, $r_i \leftarrow \omega_i / c_i$, $\{(k)\} \leftarrow$ sort desc. $r$
22: **for** $k = 1, \dots, N$ **while** $C > 0$ **do**
23:    **if** $c_{(k)} \leqslant C$ **then**
24:      $\xi_{(k)} \leftarrow 1$, $C \leftarrow C - c_{(k)}$
25:    **else**
26:      $\xi_{(k)} \leftarrow C / c_{(k)}$, $C \leftarrow 0$
27:    **end if**
28: **end for**
29: **return** $\{\xi_i\}$, $\frac{1}{N} \sum_{i=1}^N \omega_i \xi_i$

---

To establish finite-sample guarantees for $\varepsilon$-WDF, we adopt two key theorems from Le et al. Le & Malick (2024). Below, we present their assumptions and main results exactly as stated, as these form the foundation for the proof of our Theorem 4. For clarity, we also briefly summarize the assumptions underlying these theorems.

**Assumption 1.**

*1. $(\mathcal{X}, \|.\|_q)$ is compact.*
*2. $d$ is jointly continuous with respect to $\|.\|_q$, non-negative, and*

$$d(x, \zeta) = 0 \quad \text{if and only if} \quad x = \zeta.$$

3. *Every $f \in \mathcal{F}$ is continuous and $(\mathcal{F}, \|\cdot\|_\infty)$ is compact. Furthermore, if $N(t, \mathcal{X}, \|\cdot\|_\infty)$ denotes the $t$-packing number of $\mathcal{F}$, then Dudley's entropy of $\mathcal{F}$ is defined by*

$$\mathcal{I}_\mathcal{F} := \int_0^\infty \sqrt{\log N(t, \mathcal{X}, \|\cdot\|_\infty)} dt,$$

*is finite.*

The following constant, referred to as the critical radius $\rho_{\mathrm{crit}}$, is also introduced.

$$\rho_{\mathrm{crit}} := \inf_{f \in \mathcal{F}} \mathbb{E}_{\xi \sim P}\Big[\min\big\{c(\xi, \zeta) \colon \zeta \in \arg\max_{\zeta \in \mathcal{Z}} f(\zeta)\big\}\Big].$$

**Theorem 9** (**Generalization Guarantee for Wasserstein Robust Models** Le & Malick (2024)). *If Assumption 1 holds and $\rho_0 > 0$, then there exists $\lambda_{\mathrm{low}} > 0$ such that when $N > \frac{16(\alpha+\beta)^2}{\rho_{\mathrm{crit}}^2}$ and $\delta > \frac{\alpha}{\sqrt{n}}$, We have with probability at least $1 - \sigma$:*

$$R_{\delta, \mathbb{P}^N}(f) \geqslant \mathbb{E}_{x \sim \mathbb{P}}\big[f(x)\big] \quad \text{for all } f \in \mathcal{F},$$

*where $\alpha$ and $\beta$ are the two constants*

$$\alpha = 48\Big(1 + \|\mathcal{F}\|_\infty + \tfrac{1}{\lambda_{\mathrm{low}}}\Big)\Big(I_\mathcal{F} + \tfrac{2\|\mathcal{F}\|_\infty}{\lambda_{\mathrm{low}}}\sqrt{2\log\tfrac{4}{\sigma}}\Big), \quad \beta = \tfrac{96 I_\mathcal{F}}{\lambda_{\mathrm{low}}} + 48\tfrac{\|\mathcal{F}\|_\infty}{\lambda_{\mathrm{low}}}\sqrt{2\log\tfrac{4}{\sigma}}.$$

**Proposition 12** (**Excess Risk for Wasserstein Robust Models** Le & Malick (2024)). *Let $\alpha$ be given by Theorem 9 Under Assumption 1, if $\rho_{\mathrm{crit}} > 0$,*

$$N > \frac{16\alpha^2}{\rho_{\mathrm{crit}}^2} \quad and \quad \delta \leqslant \frac{\rho_{\mathrm{crit}}}{4} - \frac{\alpha}{\sqrt{n}},$$

*then with probability at least $1 - \delta$,*

$$R_{\delta, \mathbb{P}^N}(f) \leqslant R_{\delta + \alpha/\sqrt{N}, \mathbb{P}}(f) \qquad \text{for all } f \in \mathcal{F}.$$

*In particular, if $c = d(\cdot, \cdot)^p$ with $p \in [1, \infty)$ and every $f \in \mathcal{F}$ is $\mathrm{Lip}_\mathcal{F}$–Lipschitz, then*

$$R_{\delta, \mathbb{P}^N}(f) \leqslant \mathbb{E}_{z \sim \mathbb{P}}\big[f(z)\big] + \mathrm{Lip}_\mathcal{F}\left(\delta + \frac{\alpha}{\sqrt{N}}\right)^{1/p}.$$

We conclude this section with Algorithm 2, which blends a Fast-Sweeping level-set solver with a fractional knapsack routine to produce the optimal fractional activation vector $\xi \in [0, 1]^N$ under an $\ell_q$ budget constraint.

## B  PROOF

**Proof of Proposition 1.**  First, we need to prove the following lemma:

**Lemma 1** (**Compact Approximation of Support**). *Let $\{x_i\}_{i=1}^N$ be a set of observations in a Polish space $\mathcal{X}$ with proper metric, and consider the ambiguity set $\mathcal{B}_\delta(\mathbb{P}^N)$ centered at the empirical distribution $\mathbb{P}^N$ with radius $\delta > 0$. Then, for any $\varepsilon > 0$, there exists a compact set $K_\epsilon \subseteq \mathcal{X}$, such that for all measures $\mathbb{Q} \in \mathcal{B}_\delta(\mathbb{P}^N)$, we have $\mathbb{Q}(X \in K_\epsilon) > 1 - \varepsilon$.*

**Proof of Lemma 1.**  The empirical distribution $\mathbb{P}^N$ assigns probability mass $\frac{1}{N}$ to each observation $x_i$. Let $S = \{x_1, x_2, \ldots, x_N\}$ denote the support of $\mathbb{P}^N$, which is a finite set and thus compact due to its finiteness in the metric space $\mathcal{X}$. Let $r > 0$ be a radius to be determined later, and define the closed $r$-neighborhood of $S$ as

$$K_r = \bigcup_{i=1}^N \overline{B}(x_i, r),$$

where $\overline{B}(x_i, r) = \{x \in \mathcal{X} : d(x, x_i) \leqslant r\}$ is the closed ball of radius $r$ centered at $x_i$. Since $S$ is finite and each $\overline{B}(x_i, r)$ is closed, their finite union $K_r$ is closed. Additionally, each ball is bounded (diameter at most $2r$), and the finite union of bounded sets is bounded, so in a Polish space with a proper metric, where closed and bounded subsets are compact, $K_r$ is compact.

Our goal is to choose $r > 0$ such that, for all $\mathbb{Q} \in \mathcal{B}_\delta(\mathbb{P}^N)$, $\mathbb{Q}(X \in K_{2r}) > 1 - \varepsilon$. Since for $\mathcal{B}_\delta(\mathbb{P}^N)$ we have simple below equation:

$$\mathcal{B}_\delta(\mathbb{P}^N) = \big\{\mathbb{Q} : W_{q,d}(\mathbb{Q}, \mathbb{P}^N) \leqslant \delta\big\} = \big\{\mathbb{Q} : W_{1,d^q}(\mathbb{Q}, \mathbb{P}^N) \leqslant \delta^q\big\},$$

where $W_{q,d}$ means Wasserstein distance with power $q$ and distance $d$. It result to find the properties of $\mathbb{Q}$ we only need to check problem for $q = 1$ and $d'(x_1, x_2) = d^q(x_1, x_2)$, So for simplicity, we can take $q = 1$, which is standard for applying Kantorovich–Rubinstein duality Villani et al. (2009) which states: The Kantorovich–Rubinstein duality states that this distance can equivalently be expressed as

$$W_1(\mathbb{P}, \mathbb{Q}) = \sup_{\|f\|_L \leq 1} \left\{ \int_X f(x) \ d\mathbb{P}(x) - \int_X f(x) \ d\mathbb{Q}(x) \right\},$$

where the supremum is taken over all functions $f : X \to \mathbb{R}$ with Lipschitz constant not exceeding 1. Therefore, for any $\mathbb{Q} \in \mathcal{B}_\delta(\mathbb{P}^N)$ and any non-negative, Lipschitz continuous function $f : \mathcal{X} \to \mathbb{R}$ with Lipschitz constant $L_f$, the Kantorovich–Rubinstein duality implies

$$\left| \int f \ d\mathbb{Q} - \int f \ d\mathbb{P}^N \right| \leq L_f \delta^q.$$

Let us define the function $f_r : \mathcal{X} \to [0,1]$ as

$$f_r(x) = \begin{cases} 1, & \text{if } x \in K_r, \\ 1 - \dfrac{1}{r}\text{dist}^q(x, K_r), & \text{if } x \notin K_r \text{ and } \text{dist}(x, K_r) \leq r, \\ 0, & \text{if } \text{dist}(x, K_r) > r, \end{cases}$$

where $\text{dist}(x, K_r) = \inf_{y \in K_r} d(x, y)$. The function $f_r$ is Lipschitz continuous with Lipschitz constant $L_{f_r} = \frac{1}{r}$, and serves as a non-negative, bounded approximation to the indicator of $K_r$.

Compute the expectation of $f_r$ under $\mathbb{P}^N$:

$$\int f_r d\mathbb{P}^N = \frac{1}{N} \sum_{i=1}^{N} f_r(x_i) = 1,$$

since each $x_i \in S \subseteq K_r$ by construction, so $f_r(x_i) = 1$ for all $i$. Using the inequality from Kantorovich–Rubinstein duality, we have

$$\int f_r d\mathbb{Q} \geq \int f_r d\mathbb{P}^N - L_{f_r} \delta^q = 1 - \frac{\delta^q}{r}.$$

Since $f_r(x) \leq \mathbb{I}_{K_{2r}}(x)$ for all $x$, where $\mathbb{I}_{K_{2r}}$ is the indicator function of $K_{2r}$, it follows that

$$\mathbb{Q}(\mathbf{X} \in K_{2r}) = \int \mathbb{I}_{K_{2r}} d\mathbb{Q} \geq \int f_r d\mathbb{Q} \geq 1 - \frac{\delta^q}{r}.$$

To ensure that $\mathbb{Q}(\mathbf{X} \in K_{2r}) > 1 - \varepsilon$, choose $r$ such that

$$\frac{\delta^q}{r} < \varepsilon \quad \implies \quad r > \frac{\delta^q}{\varepsilon}.$$

Then set $K_\epsilon = K_{2r}$ depends on $\delta$ , and we have $\mathbb{Q}(\mathbf{X} \in K_\epsilon) > 1 - \varepsilon$. Since $K_\epsilon$ is compact, this establishes the existence of a compact set satisfying the required condition, completing the proof. □

For each $\varepsilon > 0$, by Lemma 1, there exists a compact set $K_\epsilon$ such that for all $\mathbb{Q} \in \mathcal{B}_\delta(\mathbb{P}^N)$, we have $\mathbb{Q}(\mathbf{X} \in K_\epsilon) > 1 - \varepsilon$. We show that there exists $\theta$ such that for it we have $g_\theta(x) > 0, \forall x \in K_\epsilon$. By assumption, $g_\theta$ has a neural network header, so we can write the

$$g_\theta(x) = \rho\big(\theta_1^\top \phi_{\theta_2}(x) + \theta_0\big), \qquad \theta = (\theta_0, \theta_1, \theta_2),$$

Where $\rho$ is a continuous link function with domain in $\mathbb{R}$, and $\phi_{\theta_2}$ is a feature extractor, such as a kernel map, or a neural network with parameters $\theta_2$. By assumption, $\rho$ is a continuous function with respect to $x$ and $\theta$. Then the inverse image $\rho^{-1}((0, \infty))$ is an open set (suppose $\rho$ has positive in its domain). So there exists an open interval $(\alpha, \beta) \subset \rho^{-1}((0, \infty)) \subset \mathbb{R}^+$. Fix some $\theta_2$ such that $\phi_{\theta_2}(K_\epsilon) \subset \phi_{\theta_2}(\mathcal{X})$. Since $\phi_{\theta_2}$ is continuous function then $\phi_{\theta_2}(K_\epsilon)$ is compact, and bounded; therefore, we can find parameters $\theta_0$ and $\theta_1$ such that $\theta_1 \phi_{\theta_2}(K_\epsilon) + \theta_0 \subset (\alpha, \beta)$ and $\theta_1 \phi_{\theta_2}(\mathcal{X}) + \theta_0 \not\subset (\alpha, \beta)$. It means for all $x \in K_\epsilon$, there exist non-trivial parameters $\theta$ such that for all $x \sim \mathbb{Q} \in \mathcal{B}_\delta(\mathbb{P}^N)$, we have $h_\theta(x) = 1$ with high probability $1 - \varepsilon$ and there exists $x \in \mathcal{X} \backslash K_\epsilon$ such that $h_\theta(x) = 0$. By the definition of the generic notion of fairness, it satisfies the group fairness. Since for each $\epsilon$ the equation has a solution, the equation has a solution almost surely. □

**Proof of Proposition 2.** To prove the proposition, it is sufficient to show that, as the Wasserstein radius $\delta \downarrow 0$, the distributionally–robust fair-learning problem

$$(\mathrm{DRO})_\delta := \min_{\theta \in \Theta} F_\delta(\theta) \quad \text{s.t.} \quad G_\delta(\theta) \leqslant \varepsilon,$$

where

$$F_\delta(\theta) := \sup_{\mathbb{Q} \in \mathcal{B}_\delta(\mathbb{P}^N)} \mathbb{E}_{\mathbb{Q}}\left[\ell\left(h_\theta(\mathbf{X}), \mathbf{Y}\right)\right], \quad G_\delta(\theta) := \sup_{\mathbb{Q} \in \mathcal{B}_\delta(\mathbb{P}^N)} \left\| \mathbb{E}_{\mathbb{Q}}\left[h_\theta(\mathbf{X})\varphi\left(\mathbf{U}, \mathbb{E}_{\mathbb{Q}}\left[\mathbf{U}\right]\right)\right] \right\|_\infty,$$

converges (value and minimizers) to the nominal fair-constrained problem $(\mathrm{NR}) = (\mathrm{DRO})_0$. We need to prove the two lemmas below before discussing assertions.

**Lemma 2.** *By assumption (i), we have:*

$$\lim_{\delta \to 0} G_\delta(\theta) = G_0(\theta)$$

*Proof.* By assumption the classifier $h_\theta$ for each $\theta \in \Theta$, is upper-semicontinuous so the function $h_\theta(\cdot)\varphi\left(\mathbf{U}, \mathbb{E}_{\mathbb{Q}}\left[\mathbf{U}\right]\right)$ also upper-semicontinuous and that for $q < \infty$ the following growth condition holds:

$$\exists x_0 \in \mathcal{X} \quad \text{such that} \quad \sup_{\theta \in \Theta} \limsup_{d(x, x_0) \to \infty} \frac{\left(h_\theta(x)\varphi\left(\mathbf{U}, \mathbb{E}_{\mathbb{Q}}\left[\mathbf{U}\right]\right) - h_\theta(x_0)\varphi\left(\mathbf{U}, \mathbb{E}_{\mathbb{Q}}\left[\mathbf{U}\right]\right)\right)^+}{d(x, x_0)^q} < \infty,$$

Then by applying the proposition 1 of Gao et al. (2024) we can write

$$\lim_{\delta \to 0} \sup_{\mathbb{Q} \in \mathcal{B}_\delta(\mathbb{P}^N)} \mathbb{E}_{\mathbb{Q}}\left[h_\theta(\mathbf{X})\varphi\left(\mathbf{U}, \mathbb{E}_{\mathbb{Q}}\left[\mathbf{U}\right]\right)\right] - \mathbb{E}_{\mathbb{P}^N}\left[h_\theta(\mathbf{X})\varphi\left(\mathbf{U}, \mathbb{E}_{\mathbb{P}^N}\left[\mathbf{U}\right]\right)\right] = 0$$

$$\implies \lim_{\delta \to 0} G_\delta(\theta) = G_0(\theta) \tag{A}$$

$\square$

**Lemma 3.** *By assumptions (i) and (iii), the function $G_0(\theta)$ is continuous.*

*Proof.* Since the $G_0(\theta) = \mathbb{E}_{x \sim \mathbb{P}_0}\left[h_\theta(x)\right] - \mathbb{E}_{x \sim \mathbb{P}_1}\left[h_\theta(x)\right]$, then if we prove for arbitrary $\mathbb{P}$ By the assumption, it suffices to show $F(\theta) = \mathbb{E}_{x \sim \mathbb{P}}\left[h_\theta(x)\right]$ is continuous then the assertion is satisfied. Fix $\theta \in \Theta$ and let $\{\theta_n\}_{n \in \mathbf{N}} \subset \Theta$ with $\theta_n \to \theta$. Smoothness of $g$ implies $g_{\theta_n}(x) \to g_\theta(x)$ for every $x \in \mathbf{R}^d$. Define

$$\Delta_n(x) = \mathbb{1}_{\{g_{\theta_n}(x) \geqslant 0\}} - \mathbb{1}_{\{g_\theta(x) \geqslant 0\}}.$$

If $g_\theta(x) \neq 0$, the sign of $g_{\theta_n}(x)$ eventually matches the sign of $g_\theta(x)$, hence $\Delta_n(x) \to 0$. The exceptional set $A_\theta := \{x : g_\theta(x) = 0\}$ has probability 0 by Assumption (iii).

Because $|\Delta_n(x)| \leqslant 1$ for all $(x, n)$ and $p$ is integrable, The dominated convergence theorem yields

$$\left|F(\theta_n) - F(\theta)\right| = \left|\int_{\mathbf{R}^d} \Delta_n(x)\, p(x)\, dx\right| \longrightarrow 0.$$

Thus $F(\theta_n) \to F(\theta)$, proving continuity of $F$ on $\Theta$. $\square$

By assumption, we know that the loss $(x, y) \mapsto \ell(h_\theta(x), y)$ is $L$ Lipschitz in $z$ and $\theta$. For example, we have score-based loss $\ell(g_\theta(X), Y)$, such as Hinge loss, which is Lipschitz. Since the Lipschitz property is preserved by the average, the $F_\delta(\theta)$ has Lipschitz and continuous too. By Kantorovich–Rubinstein duality Villani et al. (2009) yields, for every $\theta \in \Theta$,

$$\left|F_\delta(\theta) - F_0(\theta)\right| \leqslant L\, W_1(\mathbb{Q}, \mathbb{P}^N) \leqslant L\, W_q(\mathbb{Q}, \mathbb{P}^N) \leqslant L\, \delta. \tag{B}$$

By assumption, the bounds equation B are *uniform* in $\theta$. The mapping $\delta \mapsto G_\delta(\theta)$ is non-decreasing, whence the feasible sets satisfy $\mathcal{S}(\delta) \supseteq \mathcal{S}(\delta')$ for $\delta < \delta'$ and the optimal values $v(\delta) := \inf_{\theta \in \mathcal{S}(\delta)} F_\delta(\theta)$ form a non-increasing sequence.

By assumption there exist strictly feasible $\theta_0 \in \Theta$ with $G_0(\theta_0) < \varepsilon$. Let $\rho = \varepsilon - G_0(\theta_0) > 0$. By Lemma 2, there exist $\delta_0$ such that for $\delta < \delta_0$, we have $G_\delta(\theta_0) - G_0(\theta_0) < \rho$, therefore we have $G_\delta(\theta_0)$ satisfies the fairness constraints and therefore $\mathcal{S}(\delta)$ is non-empty.

we show $v(\delta) \downarrow v(0)$ as $\delta \downarrow 0$. By proof by contradiction suppose there exist sequence $\{\delta_k\}_{k=1}^\infty$ such that $\delta_k \to 0$ and for it there exist $\tau > 0$ such that for it $v(\delta_k) \geqslant v(0) + \tau$ for all $k$. Let $\theta^*$ be the solution of $v(0)$. We assert without loss of generality that we can suppose for every small enough $\rho > 0$, there exists $\hat{\theta} \in B_\rho(\theta^*)$ such that for it we have $G_0(\hat{\theta}) < \varepsilon$. If $G_0(\theta^*) < \varepsilon$,

by continuity of $G_0$ by Lemma 3, there exist $\rho_0$ such that for $\rho < \rho_0$ for all $\hat{\theta} \in B_\rho(\theta^*)$, we have $G_0(\hat{\theta}) < \varepsilon$.

So suppose that $G_0(\theta^*) = \varepsilon$. Since $\mathbb{P}$ has a bounded density and $g_\theta$ is smooth with non-degenerate zeros, the classifier mapping $\theta \mapsto h_\theta$ cannot be locally constant: whenever $\theta_1 \neq \theta_2$, one has $\|h_{\theta_1} - h_{\theta_2}\|_\infty > 0$. It follows that $G_0$ itself is not locally constant at $\theta^*$. By the preceding argument, it suffices to show that $\theta^*$ cannot be a local maximum of $G_0$. Since $G_0$ is nowhere locally constant and is differentiable except at a countable set of points, we can perturb $\varepsilon$ by an arbitrarily small amount to ensure that no local extremum of $G_0$ lies exactly on the level set $G_0(\theta) = \varepsilon$. In practice, such an infinitesimal adjustment of $\varepsilon$ is always permitted.

Therefore for small enough $\rho$, there exists $\hat{\theta}$ such that $G_0(\hat{\theta}) < \varepsilon$. By continuity of $F_0$, we can select $\rho$ such that for it we have $\left| F_0(\hat{\theta}) - F_0(\theta^*) \right| < \tau/2$.

Such as $\theta_0$, there exist $\delta_0$ that if $\delta_k < \delta_0$, we have $G_\delta(\hat{\theta}) < \varepsilon$, so we can write:

$$F_{\delta_k}(\hat{\theta}) \geqslant F_0(\theta^*) + \tau > F_0(\hat{\theta}) + \tau/2 \implies \left| F_{\delta_k}(\hat{\theta}) - F_0(\hat{\theta}) \right| > \tau/2 \implies L\delta_k > \tau/2$$

So the last inequality is not valid for small $\delta_k$; consequently, by contradiction, we show $v(\delta) \downarrow v(0)$.

Let $\theta_\delta \in \arg\min(\text{DRO})_\delta$ and pick any sequence $\delta_k \downarrow 0$ for which $\theta_{\delta_k} \to \theta^*$ (compactness of $\Theta$). by continuity of $G_\delta$ at 0, together with $G_{\delta_k}(\theta_{\delta_k}) \leqslant \varepsilon$, gives $G_0(\theta^*) \leqslant \varepsilon$, i.e. $\theta^*$ is feasible for (NR). Using equation B and the value convergence,

$$F_0(\theta^*) = \lim_{k \to \infty} F_{\delta_k}(\theta_{\delta_k}) = \lim_{k \to \infty} v(\delta_k) = v(0),$$

so $\theta^*$ is *optimal* for (NR). Hence, every accumulation point of DRO minimizers lies in $\arg\min(\text{NR})$, proving set convergence. $\qquad\square$

**Proof of Proposition 3.** By assumption (iv) the cost function $c$ is defined as:

$$c\big((x,a,y),\ (x',a',y')\big) = d(x,x') + \infty \cdot \mathbb{I}(a \neq a') + \infty \cdot \mathbb{I}(y \neq y'),$$

The cost function imposes a constraint that if the actions $a$ and $a'$ are not equal or $y$ and $y'$ are not, the cost becomes infinite. This implies that in the Wasserstein distance computation between distributions $\mathbb{Q}$ and $\mathbb{P}$, the marginal distributions over actions $\mathbf{A}$ and labels $\mathbf{Y}$ must match exactly, i.e., $\mathbb{Q}_{\mathbf{A},\mathbf{Y}} = \mathbb{P}_{\mathbf{A},\mathbf{Y}}$.

Let $\mathbb{P}$ be a nominal probability distribution and consider the Wasserstein ambiguity set:

$$\mathcal{B}_\delta(\mathbb{P}) = \left\{ \mathbb{Q} \in \mathcal{P}(\mathcal{X} \times \mathcal{A} \times \mathcal{Y}) \mid W_q(\mathbb{P},\mathbb{Q}) \leqslant \delta \right\}.$$

By the Kantorovich–Rubinstein duality Villani et al., 2009, Theorem 1.14, the $q$-Wasserstein distance between two probability distributions $\mathbb{P}$ and $\mathbb{Q}$ is given by:

$$W_q^q(\mathbb{P},\mathbb{Q}) = \sup_{\|f\|_{\text{Lip}} \leqslant 1} \left( \int_{\mathcal{X} \times \mathcal{A} \times \mathcal{Y}} f(x,a,y) \, d\mathbb{P} - \int_{\mathcal{X} \times \mathcal{A} \times \mathcal{Y}} f(x,a,y) \, d\mathbb{Q} \right),$$

where $f$ is a 1-Lipschitz function respect to the cost function $d^q$.

Now, applying this dual form of the Wasserstein distance to the distributions $\mathbb{Q}_{a,y}$ and $\mathbb{P}_{a,y}$, we have:

$$\sup_{\|f\|_{\text{Lip}} \leqslant 1} \left( \int_{\mathcal{X} \times \mathcal{A} \times \mathcal{Y}} f(x,a,y) \, d\mathbb{P} - \int_{\mathcal{X} \times \mathcal{A} \times \mathcal{Y}} f(x,a,y) \, d\mathbb{Q} \right) =$$

$$\sup_{\|f\|_{\text{Lip}} \leqslant 1} \left( \int_{\mathcal{A} \times \mathcal{Y}} \int_{\mathcal{X}} f(x,a,y) \, d\mathbb{P}_{a,y}(x) \, d\mathbb{P}_{\mathbf{A},\mathbf{Y}}(a,y) - \int_{\mathcal{A} \times \mathcal{Y}} \int_{\mathcal{X}} f(x,a,y) \, d\mathbb{Q}_{a,y}(x) \, d\mathbb{Q}_{\mathbf{A},\mathbf{Y}}(a,y) \right) =$$

$$\sup_{\|f\|_{\text{Lip}} \leqslant 1} \left( \sum_{(a,y) \in \mathcal{A} \times \mathcal{Y}} \mathbb{P}_{\mathbf{A},\mathbf{Y}}(a,y) \left( \int_{\mathcal{X}} f(x,a,y) \, d\mathbb{P}_{a,y}(x) - \int_{\mathcal{X}} f(x,a,y) \, d\mathbb{Q}_{a,y}(x) \right) \right) =$$

$$\sum_{(a,y) \in \mathcal{A} \times \mathcal{Y}} \mathbb{P}_{\mathbf{A},\mathbf{Y}}(a,y) \left( \sup_{\|f_{a,y}\|_{\text{Lip}} \leqslant 1} \left( \int_{\mathcal{X}} f_{a,y}(x) \, d\mathbb{P}_{a,y}(x) - \int_{\mathcal{X}} f_{a,y}(x) \, d\mathbb{Q}_{a,y}(x) \right) \right) =$$

$$\sum_{(a,y) \in \mathcal{A} \times \mathcal{Y}} \mathbb{P}_{\mathbf{A},\mathbf{Y}}(a,y) W_q^q(\mathbb{Q}_{a,y}, \mathbb{P}_{a,y})$$

where $f_{a,y}(x) = f(x, a, y)$. Since the total Wasserstein distance is bounded by $\delta$, summing over all $(a, y) \in \mathcal{A} \times \mathcal{Y}$, the ambiguity set $\mathcal{B}_\delta(\mathbb{P})$ restricts the Wasserstein distances as:

$$\sum_{(a,y)\in\mathcal{A}\times\mathcal{Y}} \mathbb{P}_{\mathbf{A},\mathbf{Y}}(a,y) W_q^q(\mathbb{Q}_{a,y}, \mathbb{P}_{a,y}) \leqslant \delta^q$$

where $W_q(\mathbb{Q}_{a,y}, \mathbb{P}_{a,y})$ is the $q$-Wasserstein distance between these conditional distributions computed with the cost $d$. $\qquad\square$

**Proof of Proposition 4.** By the definition 1, $h_\theta$ satisfies the $\varepsilon$-WDF property, if

$$\sup_{\mathbb{Q}\in\mathcal{B}_\delta(\mathbb{P}^N)} \left\{ \|\mathbb{E}_\mathbb{Q}[h_\theta(\mathbf{X})\varphi(\mathbf{U}, \mathbb{E}_\mathbb{Q}[\mathbf{U}])]\|_\infty \right\} \leqslant \varepsilon \iff \sup_{\mathbb{Q}\in\mathcal{B}_\delta(\mathbb{P}^N)} |\mathbb{E}_\mathbb{Q}[h_\theta(\mathbf{X})\varphi_i(\mathbf{A}, \mathbf{Y})]| \leqslant \varepsilon, \forall i$$

$$\iff \sup_{\mathbb{Q}\in\mathcal{B}_\delta(\mathbb{P}^N)} \mathbb{E}_\mathbb{Q}[h_\theta(\mathbf{X})\varphi_i(\mathbf{A}, \mathbf{Y})] \leqslant \varepsilon \ \wedge \ \inf_{\mathbb{Q}\in\mathcal{B}_\delta(\mathbb{P}^N)} \mathbb{E}_\mathbb{Q}[h_\theta(\mathbf{X})\varphi_i(\mathbf{A}, \mathbf{Y})] \geqslant -\varepsilon, \ \forall i$$

$$\iff \mathcal{S}_{\delta,q}^i(\mathbb{P}, \theta) + \mathcal{F}(\mathbb{P}, \theta) \leqslant \varepsilon \ \wedge \ \mathcal{I}_{\delta,q}^i(\mathbb{P}, \theta) - \mathcal{F}(\mathbb{P}, \theta) \leqslant \varepsilon, \ \forall i$$

$$\iff \max(\mathcal{S}_{\delta,q}(\mathbb{P}, \theta)) + \mathcal{F}(\mathbb{P}, \theta) < \varepsilon \ \wedge \ \max(\mathcal{I}_{\delta,q}(\mathbb{P}, \theta)) - \mathcal{F}(\mathbb{P}, \theta) < \varepsilon$$

The last equation completes the proof. $\qquad\square$

**Proof of Theorem 1.** Based on Proposition 7, we need to compute the mapping worst-case fairness criteria that depends on computing $\psi^*(z) = \sup_{d(x',x)\leqslant\delta} \psi(x', a, y)$ for the function $\psi(z) = h_\theta(x)\left(p_0^{-1}\mathbb{1}_{S_0}(a,y) - p_1^{-1}\mathbb{1}_{S_1}(a,y)\right)$. First, we need to compute the value of $\psi$ under different conditions. It is simply obtained by:

$$\psi(z) = \begin{cases} 0, & (x,a,y) \in \mathcal{X}^- \times S_0, \\ 0, & (x,a,y) \in \mathcal{X}^- \times S_1, \\ p_0^{-1}, & (x,a,y) \in \mathcal{X}^+ \times S_0, \\ -p_1^{-1}, & (x,a,y) \in \mathcal{X}^+ \times S_1. \end{cases} \implies \psi^*(z) = \begin{cases} 0, & (x,a,y) \in \mathcal{X}^- \times S_0 \wedge d_+(x) \geqslant \delta, \\ p_0^{-1}, & (x,a,y) \in \mathcal{X}^- \times S_0 \wedge d_+(x) < \delta, \\ 0, & (x,a,y) \in \mathcal{X}^- \times S_1, \\ p_0^{-1}, & (x,a,y) \in \mathcal{X}^+ \times S_0, \\ -p_1^{-1}, & (x,a,y) \in \mathcal{X}^+ \times S_1 \wedge d_-(x) \geqslant \delta, \\ 0, & (x,a,y) \in \mathcal{X}^+ \times S_1 \wedge d_-(x) < \delta. \end{cases}$$

Therefore by subtracting $\psi^*$ by $\psi$ we have:

$$(\psi^* - \psi)(z) = \begin{cases} p_0^{-1}, & (x,a,y) \in \mathcal{X}^- \times S_0 \wedge d_+(x) < \delta, \\ p_1^{-1}, & (x,a,y) \in \mathcal{X}^+ \times S_1 \wedge d_-(x) < \delta, \\ 0 & \text{otherwise.} \end{cases}$$

Therefore, we have:

$$\mathcal{S}_{\delta,\infty}(\mathbb{P}, \theta) = \sup_{\mathbb{Q}\in\mathcal{B}_\delta(\mathbb{P})} \left\{ \mathbb{E}_{z\sim\mathbb{Q}}[\psi(z)] \right\} - \mathbb{E}_{z\sim\mathbb{P}}[\psi(z)] = \mathbb{E}_{z\sim\mathbb{P}}[(\psi^* - \psi)(z)]$$

$$= p_0^{-1}\mathbb{P}(z : \mathcal{X}^- \times S_0 \wedge d_+(x) < \delta) + p_1^{-1}\mathbb{P}(z : \mathcal{X}^+ \times S_1 \wedge d_-(x) < \delta)$$

$$= \mathbb{P}_0(\mathcal{X}^-)p_0^{-1}\mathbb{P}(S_0 \wedge d_+(x) < \delta \mid d_+(x) > 0) + p_1^{-1}\mathbb{P}_1(\mathcal{X}^+)\mathbb{P}(S_1 \wedge d_-(x) < \delta \mid d_-(x) > 0)$$

$$= \mathbb{P}_0(\mathcal{X}^-)G_0^+(\delta) + \mathbb{P}_1(\mathcal{X}^+)G_1^-(\delta)$$

If we define $\psi_*(z) = \sup_{d(x',x)\leqslant\delta} \psi(x', a, y)$, then we have:

$$(\psi - \psi_*)(z) = \begin{cases} p_0^{-1}, & (x,a,y) \in \mathcal{X}^+ \times S_0 \wedge d_-(x) < \delta, \\ p_1^{-1}, & (x,a,y) \in \mathcal{X}^- \times S_1 \wedge d_+(x) < \delta, \\ 0 & \text{otherwise.} \end{cases}$$

Then we have:

$$\mathcal{I}_{\delta,\infty}(\mathbb{P}, \theta) = \mathbb{E}_{z\sim\mathbb{P}}[(\psi^* - \psi)(z)] = \mathbb{P}_0(\mathcal{X}^+)G_0^-(\delta) + \mathbb{P}_1(\mathcal{X}^-)G_1^+(\delta)$$

The last completes the proof. $\qquad\square$

**Proof of Corollary 1.** The proof is obtained by applying Theorem 1. When we have:

$$\mathbb{P}\big(x : \mathrm{dist}(x, \mathcal{L}_\theta) \leqslant \delta\big)$$

$$= \mathbb{P}(d_+(x) \leqslant \delta \mid d_+(x) > 0)\mathbb{P}(d_+(x) > 0) + \mathbb{P}(d_-(x) \leqslant \delta \mid d_-(x) > 0)\mathbb{P}(d_-(x) > 0)$$

$$= p_0 \mathbb{P}_0(\mathcal{X}^-)G_0^+(\delta) + p_1 \mathbb{P}_1(\mathcal{X}^+)G_1^-(\delta) \geqslant \min(p_0, p_1)\, \mathcal{S}_{\delta,\infty}(\mathbb{P}, \theta)$$

$$\implies \mathcal{S}_{\delta,\infty}(\mathbb{P}, \theta) \leqslant \frac{1}{\min(p_0, p_1)}\mathbb{P}\big(x : \mathrm{dist}(x, \mathcal{L}_\theta) \leqslant \delta\big)$$

Similarly, it can be shown that:

$$\mathcal{I}_{\delta,\infty}(\mathbb{P}, \theta) \leqslant \frac{1}{\min(p_0, p_1)}\mathbb{P}\big(x : \mathrm{dist}(x, \mathcal{L}_\theta)\big) \leqslant \delta)$$

By combining the two results, it is concluded that:

$$\frac{1}{\min(p_0, p_1)}\mathbb{P}\big(x : \mathrm{dist}(x, \mathcal{L}_\theta)\big) \leqslant \delta) \geqslant \max(\mathcal{S}_{\delta,\infty}(\mathbb{P}, \theta), \mathcal{I}_{\delta,\infty}(\mathbb{P}, \theta))$$

Now by applying the proposition 4, we can say $h_\theta$ satisfies the $\varepsilon$-WDF property if:

$$|\mathcal{F}(\mathbb{P}, \theta)| + \frac{1}{\min(p_0, p_1)}\mathbb{P}\big(x : \mathrm{dist}(x, \mathcal{L}_\theta)\big) \leqslant \delta) \leqslant \varepsilon$$

The last equation completes the proof. $\qquad\square$

**Proof of Theorem 2.** We want to compute the worst-case loss quantity. By strong duality formula which has explained in Proposition 7, we have:

$$\sup_{\mathbb{Q} \in \mathcal{B}_\delta(\mathbb{P})} \left\{ \mathbb{E}_{z \sim \mathbb{Q}}[\psi(z)] \right\} = \inf_{\lambda \geqslant 0} \left\{ \lambda \delta^q + \mathbb{E}_{z \sim \mathbb{P}}[\psi_\lambda(x, a, y)] \right\}$$

where $\psi_\lambda(x, a, y) = \sup_{x' \in \mathcal{X}} \psi(x', a, y) - \lambda d^q(x', x)$. We can write

$$\psi_\lambda(z) = \sup_{x' \in \mathcal{X}} h_\theta(x)\left(p_0^{-1}\mathbb{1}_{S_0}(a, y) - p_1^{-1}\mathbb{1}_{S_1}(a, y)\right) - \lambda d^q(x', x) \implies$$

$$\psi_\lambda(z) = \sup_{x' \in \mathcal{X}} \begin{cases} p_0^{-1}\mathbb{1}_{\mathcal{X}^+}(x') - \lambda d^q(x, x') & (x, a, y) \in \mathcal{X}^+ \times S_0 \\ -p_1^{-1}\mathbb{1}_{\mathcal{X}^+}(x') - \lambda d^q(x, x') & (x, a, y) \in \mathcal{X}^+ \times S_1 \\ 0 & \text{otherwise} \end{cases}$$

Since we have

$$\mathcal{S}_{\delta,q}(\mathbb{P}, \theta) = \inf_{\lambda \geqslant 0} \left\{ \lambda \delta^q + \mathbb{E}_{z \sim \mathbb{P}}[(\psi_\lambda - \psi)(z)] \right\} \tag{A}$$

We want to calculate the function $\psi_\lambda - \psi$. We split it into two cases: **Case $(a, y) \in S_0$:**

$$\psi_\lambda(z) = \sup_{x' \in \mathcal{X}} \left\{ p_0^{-1}\mathbb{1}_{\mathcal{X}^+}(x') - \lambda d^q(x, x') \right\} = \begin{cases} p_0^{-1} & x \in \mathcal{X}^+ \\ p_0^{-1} - \lambda d_+^q(x) & x \notin \mathcal{X}^+, x' \in \mathcal{X}^+ \implies \\ 0 & x \notin \mathcal{X}^+, x' \notin \mathcal{X}^+ \end{cases}$$

$$\begin{cases} p_0^{-1} & x \in \mathcal{X}^+ \\ \max(0, p_0^{-1} - \lambda d_+^q(x)) & x \notin \mathcal{X}^+ \end{cases}$$

Therefore for $(a, y) \in S_0$ we have $(\psi_\lambda - \psi)(z) = \max(0, p_0^{-1} - \lambda d_+^q(x))\mathbb{1}_{\mathcal{X}^-}(x)$. **Case $(a, y) \in S_1$:**

$$\sup_{x' \in \mathcal{X}} \left\{ -p_1^{-1}\mathbb{1}_{\mathcal{X}^+}(x') - \lambda d^q(x, x') \right\} = \begin{cases} -p_1^{-1} & x \in \mathcal{X}^+, x' \in \mathcal{X}^+ \\ -\lambda d_-^q(x) & x \in \mathcal{X}^+, x' \notin \mathcal{X}^+ \implies \\ 0 & x \in \mathcal{X}^- \end{cases}$$

$$\begin{cases} \max(-p_1^{-1}, -\lambda d_-^q(x)) & x \in \mathcal{X}^+ \\ 0 & x \notin \mathcal{X}^+ \end{cases}$$

So it results for for $(a, y) \in S_1$ we have $(\psi_\lambda - \psi)(z) = \max(0, p_1^{-1} - \lambda d_-^q(x))\mathbb{1}_{\mathcal{X}^+}(x)$. By collecting both results, we have:

$$(\psi_\lambda - \psi)(z) = \begin{cases} \max(0, p_0^{-1} - \lambda d_+^q(x)), & z \in \mathcal{X}^- \times S_0, \\ \max(0, p_1^{-1} - \lambda d_-^q(x)), & z \in \mathcal{X}^+ \times S_1, \\ 0, & \text{otherwise.} \end{cases}$$

So we can calculate:

$$\psi_\lambda(z) = \begin{cases} p_0^{-1}, & z \in \mathcal{X}^+ \times S_0, \\ -p_1^{-1} + \max(0, p_1^{-1} - \lambda d_-^q(x)), & z \in \mathcal{X}^+ \times S_1, \\ \max(0, p_0^{-1} - \lambda d_+^q(x)), & z \in \mathcal{X}^- \times S_0, \\ 0, & z \in \mathcal{X}^- \times S_1. \end{cases} \tag{B}$$

By strong duality, the worst-case loss equals:

$$\mathcal{S}_{\delta,q}(\mathbb{P}, \theta) = \inf_{\lambda \geqslant 0} \left\{ \lambda \delta^q + \mathbb{E}_{z \sim \mathbb{P}} \left[ (\psi_\lambda - \psi)(z) \right] \right\} =$$

$$\inf_{\lambda \geqslant 0} \left\{ \lambda \delta^q + \mathbb{E}_{z \sim \mathbb{P}} \left[ \max(0, p_0^{-1} - \lambda d_+^q(x)) \mathbb{1}_{\mathcal{X}^- \times S_0}(z) + \max(0, p_1^{-1} - \lambda d_-^q(x)) \mathbb{1}_{\mathcal{X}^+ \times S_1}(z) \right] \right\} =$$

$$\inf_{\lambda \geqslant 0} \left\{ \lambda \delta^q + \mathbb{E}_{x \sim \mathbb{P}_0} \left[ \mathbb{1}_{\mathcal{X}^-}(x) (1 - p_0 \lambda d_+^q(x))^+ \right] + \mathbb{E}_{x \sim \mathbb{P}_1} \left[ \mathbb{1}_{\mathcal{X}^+}(x) (1 - p_1 \lambda d_-^q(x))^+ \right] \right\} =$$

$$\inf_{\lambda \geqslant 0} \left\{ \lambda \delta^q + \mathbb{P}_0(\mathcal{X}^-) \int_0^{(p_0 \lambda)^{-1/q}} (1 - p_0 \lambda s^q) \, \mathrm{d}G_0^-(s) + \mathbb{P}_1(\mathcal{X}^+) \int_0^{(p_1 \lambda)^{-1/q}} (1 - p_1 \lambda s^q) \, \mathrm{d}G_1^+(s) \right\}$$

For Computing $\mathcal{I}_{\delta,q}(\mathbb{P}, \theta)$ the infimum we have:

$$\mathcal{I}_{\delta,\infty}(\mathbb{P}, \theta) = \mathbb{E}_{z \sim \mathbb{P}} [\psi(z)] - \inf_{\mathbb{Q} \in \mathcal{B}_\delta(\mathbb{P})} \left\{ \mathbb{E}_{z \sim \mathbb{Q}} [\psi(z)] \right\} = \mathbb{E}_{z \sim \mathbb{P}} [\psi(z)] + \sup_{\mathbb{Q} \in \mathcal{B}_\delta(\mathbb{P})} \left\{ \mathbb{E}_{z \sim \mathbb{Q}} [-\psi(z)] \right\} =$$

$$\mathbb{E}_{z \sim \mathbb{P}} [\psi(z)] + \inf_{\lambda \geqslant 0} \left\{ \lambda \delta^q + \mathbb{E}_{z \sim \mathbb{P}} [\psi_\lambda^-(z)] \right\}$$

where $\psi_\lambda^-(z)$ is dual conjugate of $-h_\theta(x)[p_1^{-1} \mathbb{1}_{\mathcal{X}^+ \times S_1}(z) - p_0^{-1} \mathbb{1}_{\mathcal{X}^- \times S_0}(z)]$. With similar reasoning as in part one, we have the following:

$$(\psi + \psi_\lambda^-)(z) = \begin{cases} \max(0, p_0^{-1} - \lambda d_-^q(x)), & z \in \mathcal{X}^+ \times S_0, \\ \max(0, p_1^{-1} - \lambda d_+^q(x)), & z \in \mathcal{X}^- \times S_1, \\ 0, & \text{otherwise.} \end{cases}$$

By substituting the above function in the strong duality formula, we have

$$\mathcal{I}_{\delta,q}(\mathbb{P}, \theta) = \inf_{\lambda \geqslant 0} \left\{ \lambda \delta^q + \mathbb{E}_{z \sim \mathbb{P}} \left[ (\psi + \psi_\lambda^-)(z) \right] \right\} =$$

$$\inf_{\lambda \geqslant 0} \left\{ \lambda \delta^q + \mathbb{E}_{x \sim \mathbb{P}_0} \left[ \mathbb{1}_{\mathcal{X}^+}(x) (1 - p_0 \lambda d_-^q(x))^+ \right] + \mathbb{E}_{x \sim \mathbb{P}_1} \left[ \mathbb{1}_{\mathcal{X}^-}(x) (1 - p_1 \lambda d_+^q(x))^+ \right] \right\} =$$

$$\inf_{\lambda \geqslant 0} \left\{ \lambda \delta^q + \mathbb{P}_0(\mathcal{X}^+) \int_0^{(p_0 \lambda)^{-1/q}} (1 - p_0 \lambda s^q) \, \mathrm{d}G_0^+(s) + \mathbb{P}_1(\mathcal{X}^-) \int_0^{(p_1 \lambda)^{-1/q}} (1 - p_1 \lambda s^q) \, \mathrm{d}G_1^-(s) \right\}$$

The last equation completes the proof. $\qquad\square$

**Proof of Theorem 3.** To begin, we establish the case $q \in [1, \infty)$. Central to our analysis is a robust semi-infinite duality theorem, which forms the cornerstone of the subsequent proofs. To this end, assume that $\phi : \mathcal{X} \times \mathcal{A} \times \mathcal{Y} \to \mathbb{R}$ is a Borel measurable loss function, and recall that $p_{ay} = \mathbb{P}^N(\mathbf{A} = a, \mathbf{Y} = y)$ for all $a \in \mathcal{A}$ and $y \in \mathcal{Y}$. So we have:

**Strong Duality Theorem.** If $p_{ay} \in (0, 1)$ for all $a \in \mathcal{A}$ and $y \in \mathcal{Y}$, and if $\delta > 0$, then the following strong semi-infinite duality holds:

$$\sup_{\mathbb{Q} \in \mathcal{B}_\delta(\mathbb{P}^N)} \mathbb{E}_{\mathbb{Q}}[\phi(X, A, Y)] = \inf \quad \lambda \delta^q + \sum_{a \in \mathcal{A}} \sum_{y \in \mathcal{Y}} p_{ay} \mu_{ay} + \frac{1}{N} \sum_{i=1}^N \nu_i$$

$$\text{s.t.} \quad \lambda \in \mathbb{R}_+, \ \mu \in \mathbb{R}^{2 \times 2}, \ \nu \in \mathbb{R}^d$$

$$\lambda d^q \left( (x_i', a_i', y_i'), (x_i, a_i, y_i) \right) + \mu_{a_i y_i} + \nu_i \geqslant \phi(x_i', a_i', y_i')$$

$$\forall (x_i', a_i', y_i') \in \mathcal{X} \times \mathcal{A} \times \mathcal{Y}, \forall i \in [N]. \tag{A}$$

The proof of the above theorem can be found in the references Blanchet & Murthy (2019); Gao et al. (2017); Mohajerin Esfahani & Kuhn (2018a), so we omit it. By applying our cost

assumption, the formulation A converts to:

$$\sup_{\mathbb{Q}\in\mathcal{B}_\delta(\mathbb{P}^N)} \mathbb{E}_{\mathbb{Q}}[\phi(X,A,Y)] = \inf \quad \lambda\delta^q + \frac{1}{N}\sum_{i=1}^N \nu_i$$

$$\text{s.t.} \quad \lambda\in\mathbb{R}_+,\ \nu\in\mathbb{R}^d$$

$$\lambda d^q(x_i',x_i) + \nu_i \geqslant \phi(x_i',a_i,y_i)$$

$$\forall x_i'\in\mathcal{X},\forall i\in[N]. \tag{B}$$

To compute $\mathcal{S}_{\delta,q}(\mathbb{P}^N,\theta)$, we define the equation $\phi$ as follows:

$$\phi(x,a,y) = h_\theta(x)\left(\frac{\mathbb{1}_{S_0}(a,y)}{\mathbb{E}_{\mathbb{P}}[\mathbb{1}_{S_0}]} - \frac{\mathbb{1}_{S_1}(a,y)}{\mathbb{E}_{\mathbb{P}}[\mathbb{1}_{S_1}]}\right) = \frac{1}{p_0}\mathbb{1}_{\mathcal{X}^+\times S_0}(x,a,y) - \frac{1}{p_1}\mathbb{1}_{\mathcal{X}^+\times S_1}(x,a,y)$$

To further simplify Eq. B, we reformulate the constraints on $\nu_i$ using Proposition 2 as follows:

$$\nu_i \geqslant \sup_{x_i'\in\mathcal{X}}\left\{\phi(x_i',a_i,y_i) - \lambda d^q(x_i',x_i)\right\} = \begin{cases} p_0^{-1}, & z\in\mathcal{X}^+\times S_0, \\ -p_1^{-1} + \max(0,p_1^{-1}-\lambda d_-^q(x_i)), & z\in\mathcal{X}^+\times S_1, \\ \max(0,p_0^{-1}-\lambda d_+^q(x_i)), & z\in\mathcal{X}^-\times S_0, \\ 0, & z\in\mathcal{X}^-\times S_1. \end{cases}$$

After putting these constraints in Eq. B, we have:

$$\min \quad \lambda\delta^q + \frac{1}{N}\sum_{i=1}^N \nu_i$$

$$\text{s.t.} \quad \lambda\in\mathbb{R}_+,\ \nu\in\mathbb{R}^d$$

$$\left.\begin{array}{ll} \nu_i \geqslant p_0^{-1} & \text{if } z\in\mathcal{X}^+\times S_0 \\ \nu_i \geqslant -p_1^{-1} & \text{if } z\in\mathcal{X}^+\times S_1 \\ \nu_i + \lambda d_-^q(x_i) \geqslant 0 & \text{if } z\in\mathcal{X}^+\times S_1 \\ \nu_i \geqslant 0 & \text{if } z\in\mathcal{X}^-\times S_0 \\ \nu_i + \lambda d_-^q(x_i) \geqslant p_0^{-1} & \text{if } z\in\mathcal{X}^-\times S_0 \\ \nu_i \geqslant 0 & \text{if } z\in\mathcal{X}^-\times S_1 \end{array}\right\}\forall i\in[N]. \tag{C}$$

By defining the sets $\mathcal{I}_1^+ = \{i\in[N] : z_i\in\mathcal{X}^+\times S_1\}$ and $\mathcal{I}_0^- = \{i\in[N] : z_i\in\mathcal{X}^-\times S_0\}$, and subtracting the $\mathcal{F}(\mathbb{P}^N,\theta)$ from both side we simplified the equation as

$$\mathcal{S}_{\delta,q}(\mathbb{P}^N,\theta) = \min \quad \lambda\delta^q + \frac{1}{N}\sum_{i\in\mathcal{I}_1^+\cup\mathcal{I}_0^-}\nu_i$$

$$\text{s.t.} \quad \lambda\in\mathbb{R}_+,\ \nu\in\mathbb{R}^d$$

$$\left.\begin{array}{l} \nu_i + \lambda d_-^q(x_i) \geqslant p_1^{-1} \\ \nu_i \geqslant 0 \end{array}\right\}\forall i\in\mathcal{I}_1^+$$

$$\left.\begin{array}{l} \nu_i \geqslant 0 \\ \nu_i + \lambda d_-^q(x_i) \geqslant p_0^{-1} \end{array}\right\}\forall i\in\mathcal{I}_0^-.$$

Rewrite every inequality in the form "function $\leqslant 0$" and attach a multiplier. For each $i\in\mathcal{I}_1^+$:

$$\begin{array}{lll} g_{1i}(\lambda,\nu) := p_1^{-1} - \nu_i - \lambda d_-^q(x_i) \leqslant 0 & \longleftrightarrow & \gamma_{1i}\geqslant 0, \\ g_{2i}(\nu) := -\nu_i \leqslant 0 & \longleftrightarrow & \gamma_{2i}\geqslant 0; \\ g_{3i}(\nu) := -\nu_i \leqslant 0 & \longleftrightarrow & \gamma_{3i}\geqslant 0, \\ g_{4i}(\lambda,\nu) := p_0^{-1} - \nu_i - \lambda d_-^q(x_i) \leqslant 0 & \longleftrightarrow & \gamma_{4i}\geqslant 0. \end{array}$$

Define $d_{1i} := d_-(x_i), \forall i\in\mathcal{I}_1^+$ and $d_{0i} := d_+(x_i), \forall i\in\mathcal{I}_0^-$ the Lagrangian is

$$\mathcal{L}(\lambda,\nu,\gamma) = \lambda\delta^q + \frac{1}{N}\sum_i \nu_i + \sum_{i\in\mathcal{I}_1^+}\gamma_{1i}(p_1^{-1}-\nu_i-\lambda d_{0i}^q) + \sum_{i\in\mathcal{I}_1^+}\gamma_{2i}(-\nu_i)$$

$$+ \sum_{i\in\mathcal{I}_0^-}\gamma_{3i}(-\nu_i) + \sum_{i\in\mathcal{I}_0^-}\gamma_{4i}(p_0^{-1}-\nu_i-\lambda d_{1i}^q),$$

where $\gamma = (\gamma_1,\ldots,\gamma_4)\geqslant 0$. Because $\nu$ is unconstrained after dualisation, the finiteness of $\inf_\nu\mathcal{L}$ requires the $\nu_i$-coefficients to vanish, giving

$$\frac{1}{N} - \gamma_{1i} - \gamma_{2i} = 0 \quad (i\in\mathcal{I}_1^+), \qquad \frac{1}{N} - \gamma_{3i} - \gamma_{4i} = 0 \quad (i\in\mathcal{I}_0^-).$$

Hence $0 \leqslant \gamma_{1i}, \gamma_{4i} \leqslant 1/N$. So we can write:

$$\max \quad p_1^{-1} \sum_{i \in \mathcal{I}_1^+} \gamma_{1i} + p_0^{-1} \sum_{i \in \mathcal{I}_0^-} \gamma_{4i}$$

$$\text{s.t.} \quad \gamma_1 \in \mathbb{R}_+^{|\mathcal{I}_1^+|}, \quad \gamma_4 \in \mathbb{R}_+^{|\mathcal{I}_0^-|},$$

$$\delta^q - \sum_{i \in \mathcal{I}_1^+} \gamma_{1i} d_{1i}^q - \sum_{i \in \mathcal{I}_0^-} \gamma_{4i} d_{0i}^q \geqslant 0,$$

$$\gamma_{1i} \leqslant \frac{1}{N} \quad \forall i \in \mathcal{I}_1^+,$$

$$\gamma_{4i} \leqslant \frac{1}{N} \quad \forall i \in \mathcal{I}_0^-.$$

Set the rescaled variables.

$$\xi_i := N\gamma_{1i} \in [0,1](i \in \mathcal{I}_1^+), \qquad \xi_i := N\gamma_{4i} \in [0,1](i \in \mathcal{I}_0^-).$$

Taking the infimum over $\lambda \geqslant 0$ yields the additional feasibility condition

$$\delta^q - \sum_{i \in \mathcal{I}_1^+} \gamma_{1i} d_i^q - \sum_{i \in \mathcal{I}_0^-} \gamma_{4i} d_i^q \geqslant 0 \iff \frac{1}{N} \sum_i \xi_i d_i^q \leqslant \delta^q.$$

So the problem can be simplified as

$$\max_z \quad \frac{1}{Np_1} \sum_{i \in \mathcal{I}_1^+} \xi_i + \frac{1}{Np_0} \sum_{i \in \mathcal{I}_0^-} \xi_i$$
$$\text{s.t.} \quad 0 \leqslant \xi_i \leqslant 1 \qquad \forall i \in \mathcal{I}_1^+ \cup \mathcal{I}_0^-,$$
$$\frac{1}{N} \sum_{i \in \mathcal{I}_1^+ \cup \mathcal{I}_0^-} \xi_i d_i^q(x_i) \leqslant \delta^q.$$

**Case $q = \infty$:** In this case by Theorem 1, we can write:

$$\mathcal{S}_{\delta,\infty}(\mathbb{P}, \theta) = \mathbb{P}_0(\mathcal{X}^-)G_0^+(\delta) + \mathbb{P}_1(\mathcal{X}^+)G_1^-(\delta)$$

If instead of $\mathbb{P}$ we use the $\mathbb{P}^N$, so we have

$$\begin{cases} \widehat{G}^+(\delta) := \mathbb{P}_0^N(\mathcal{X}^-)\widehat{G_0^+}(\delta) = p_0^{-1} \frac{1}{N} \#\{z_i \in \mathcal{X}^- \times S_0 : d_+(x_i) \leqslant \delta\} \\ \widehat{G}^-(\delta) := \mathbb{P}_1^N(\mathcal{X}^+)\widehat{G_1^-}(\delta) = p_1^{-1} \frac{1}{N} \#\{z_i \in \mathcal{X}^+ \times S_1 : d_-(x_i) \leqslant \delta\} \end{cases}$$

So $\mathcal{S}_{\delta,\infty}(\mathbb{P}^N, \theta) = \widehat{G}^+(\delta) + \widehat{G}^-(\delta)$. Therefore, the last equation completes the proof. $\square$

**Proof of Theorem 4.** The complete version of Theorem 4 is presented in the following:

**Theorem.** Given that Assumptions (i)- (iv) hold, and the fairness score function is defined as in Eq. 10. Suppose $\rho_0 > 0$. Then there exists a constant $\lambda_0 > 0$ such that whenever $N > \frac{16(\alpha+\beta)^2}{\rho_0^2}$ and $\delta > \frac{\alpha}{\sqrt{N}}$, We have, with probability at least $1 - \sigma$, the uniform lower bound

$$\sup_{\mathbb{Q} \in \mathcal{B}_\delta(\mathbb{P}^N)} \mathbb{E}_{z \sim \mathbb{Q}}[f(z)] \geqslant \mathbb{E}_{z \sim \mathbb{P}}[f(z)] \quad \text{for all } \theta \in \Theta,$$

Here the constants $\alpha$ and $\beta$ depend on the dimension $K$ and diameter $D$ of the parameter space, and are defined by

$$\alpha := 48\left(2 + \frac{1}{\lambda_0}\right)\left(\frac{L}{\delta} + \frac{2}{\lambda_0}\sqrt{2\log\frac{4}{\sigma}}\right), \quad \beta := \frac{96L}{\delta\lambda_0} + 48\frac{1}{\lambda_0}\sqrt{2\log\frac{4}{\sigma}}, \quad M := \sup_{\theta \in \Theta, x \in \mathcal{X}} \left\|\nabla_\theta \, g_\theta(x)\right\|_{q*},$$

$$c := \inf_{\theta \in \Theta, x \in \mathcal{X} |g_\theta(x)| \leqslant \delta_0} \left\|\nabla_x g_\theta(x)\right\|_{q*}, \quad L := \frac{2\sqrt{\pi}DqM}{c}\max\left(\frac{1}{p_0}, \frac{1}{p_1}\right)^{\frac{q+1}{q}}\sqrt{K}.$$

Hence, $\delta_N$ decays at the dimension-independent rate $O(N^{-\frac{1}{4}})$.

Let $f$ be the fairness score function 10. The generic notion of fairness is not continuous with respect to $x$, so by adding the function $f^\epsilon(z)$:

$$g_\theta^\epsilon(z) = \begin{cases} p_0^{-1}\left(1 - \frac{1}{\epsilon^q}d_+^q(x)\right) & z \in \mathcal{X}^- \times S_0 \wedge d_+(x) \leqslant \epsilon, \\ p_1^{-1}\left(1 - \frac{1}{\epsilon^q}d_-^q(x)\right) & z \in \mathcal{X}^+ \times S_1 \wedge d_-(x) \leqslant \epsilon, \\ 0 & \text{otherwise.} \end{cases} \quad (A)$$

So the function $f_\theta^\epsilon(z) = f(z) + g_\theta^\epsilon(z)$ is continuous.

For family of functions $\mathcal{F}$, and for $\lambda \geqslant 0$, we recall the expression of the maximal radius:

$$\rho_{\max}(\lambda) = \inf_{f \in \mathcal{F}} \mathbb{E}_{z \sim \mathbb{P}} \left[ -\partial_\lambda^+ f_\lambda(z) \right].$$

where $\partial_\lambda^+$ the right-sided derivative (i.e. $\partial_\lambda^+ f(z) = \lim_{h \downarrow 0} \frac{f(z+h\lambda) - f(z)}{h}$) with respect to $\lambda \in \mathbb{R}$ and transport conjugate $f_\lambda(z) = \sup_{z' \in \mathcal{Z}} f(z') - \lambda c^q(z', z)$. Let $f_{\theta, \lambda}^\epsilon$ be the cost-conjugate of $f_\theta^\epsilon$. We need to explore the behavior of the family $\mathcal{F} = \{f_\theta^\epsilon : \theta \in \Theta\}$ and the function $f_\theta^\epsilon$. Before proving the main result, we need some lemmas.

**Lemma 4.** *If* $\lambda < \min(\frac{1}{p_0}, \frac{1}{p_1}) \frac{2}{\epsilon^q}$, *then* $f_{\theta, \lambda}^\epsilon = f_\lambda$.

**Proof of Lemma 4.** For the binary classifier $h_\theta$, the transport conjugate $f_\lambda(z) = \sup_{z' \in \mathcal{Z}} f(z') - \lambda c^q(z', z)$. It can be written:

$$\arg\max_{\mathcal{Z}} \{f(.) - \lambda c^q(., z)\} =$$

$$\begin{cases} \{(x', a, y) \in \mathcal{X}^+ \times S_0 : d(x', x) = d_+(x)\}, & z \in \mathcal{X}^- \times S_0 \wedge d_+(x) \leqslant (p_0 \lambda)^{-\frac{1}{q}}, \\ \{(x', a, y) \in \mathcal{X}^- \times S_1 : d(x', x) = d_-(x)\}, & z \in \mathcal{X}^+ \times S_1 \wedge d_-(x) \leqslant (p_1 \lambda)^{-\frac{1}{q}}, \\ \{z\}, & \text{Otherwise.} \end{cases}$$

Since our goal is to explore the behavior of $f_\theta^\epsilon$ for sufficiently small $\epsilon$ and $\lambda$, it suffices to consider the family $\mathcal{F}^\epsilon$ for the case where $\lambda < \min(\frac{1}{p_0}, \frac{1}{p_1}) \frac{2}{\epsilon^q}$. Specifically, the set of maximizers can be explicitly characterized as follows:

$$\arg\max_{\mathcal{Z}} \{f_\theta^\epsilon(\cdot) - \lambda d^q(\cdot, z)\} =$$

$$\begin{cases} \{z\}, & z \in \mathcal{X}^+ \times S_0, \\ \{(x', a, y) \in \mathcal{X}^+ \times S_0 : d(x', x) = d_+(x)\}, & z \in \mathcal{X}^- \times S_0 \wedge d_+(x) \leqslant \epsilon, \\ \{(x', a, y) \in \mathcal{X}^+ \times S_0 : d(x', x) = d_+(x)\}, & z \in \mathcal{X}^- \times S_0 \wedge \epsilon < d_+(x) \leqslant (p_0 \lambda)^{-\frac{1}{q}}, \\ \{z\}, & z \in \mathcal{X}^- \times S_0 \wedge d_+(x) > (p_0 \lambda)^{-\frac{1}{q}}, \quad\text{(B)} \\ \{z\}, & z \in \mathcal{X}^- \times S_1, \\ \{(x', a, y) \in \mathcal{X}^- \times S_1 : d(x', x) = d_-(x)\}, & z \in \mathcal{X}^+ \times S_1 \wedge d_-(x) \leqslant \epsilon, \\ \{(x', a, y) \in \mathcal{X}^- \times S_1 : d(x', x) = d_-(x)\}, & z \in \mathcal{X}^+ \times S_1 \wedge \epsilon < d_-(x) \leqslant (p_1 \lambda)^{-\frac{1}{q}}, \\ \{z\}, & z \in \mathcal{X}^+ \times S_1 \wedge d_-(x) > (p_1 \lambda)^{-\frac{1}{q}}, \end{cases}$$

Therefore in the case $\lambda < \min(\frac{1}{p_0}, \frac{1}{p_1}) \frac{2}{\epsilon^q}$, we have $f_{\theta, \lambda}^\epsilon(z) = f_\lambda(z)$ and completes the proof. $\square$

**Lemma 5.** *let* $\lambda^*$ *be the solution of problem* $\inf_{\lambda \geqslant 0} \{\lambda \delta^q + \mathbb{E}_{z \sim \mathbb{P}}[f_\lambda(z)]\}$, *then* $\lambda^* \leqslant \max(\frac{1}{p_0}, \frac{1}{p_1}) \frac{2}{\delta^q}$.

**Proof of Lemma 5.** By applying part (iii) of Proposition 7 for fairness score $f$, we can write that

$$\delta^q = \mathbb{E}_{x \sim \mathbb{P}_0}\left[ d_+^{\,q}(x) \mathbb{I}\left(0 < d_+(x) \leqslant (p_0 \lambda^*)^{-\frac{1}{q}}\right)\right] + \mathbb{E}_{x \sim \mathbb{P}_1}\left[ d_-^{\,q}(x) \mathbb{I}\left(0 < d_-(x) \leqslant (p_1 \lambda^*)^{-\frac{1}{q}}\right)\right]$$

$$\leqslant \frac{1}{p_0 \lambda^*} \mathbb{E}_{x \sim \mathbb{P}_0}\left[ \mathbb{I}\left(0 < d_+(x) \leqslant (p_0 \lambda^*)^{-\frac{1}{q}}\right)\right] + \frac{1}{p_1 \lambda^*} \mathbb{E}_{x \sim \mathbb{P}_1}\left[ \mathbb{I}\left(0 < d_-(x) \leqslant (p_1 \lambda^*)^{-\frac{1}{q}}\right)\right]$$

$$\leqslant \max(\frac{1}{p_0}, \frac{1}{p_1}) \frac{1}{\lambda^*} \implies \lambda^* \leqslant \max(\frac{1}{p_0}, \frac{1}{p_1}) \frac{2}{\delta^q}.$$

Where $\mathbb{I}$ is the indicator function. The last equation completes the proof. $\square$

**Lemma 6.** *Let* $\mathcal{F}^\epsilon := \{f_\theta^\epsilon : \theta \in \Theta\}$ *be the family of functions defined in Eq. A, constructed from the original classifier family* $\mathcal{F}$. *Then we have* $\rho_{\max}^\epsilon(\lambda)$ *is right continuous at zero and* $\lim_{\lambda \to 0^+} \rho_{\max}^\epsilon(\lambda) = \rho_0$. *Moreover, there exists a constant* $\lambda_0 > 0$ *such that*

$$\rho_{\max}^\epsilon(\lambda) \geqslant \frac{\rho_0}{4}, \quad \text{for all } \lambda \in [0, 2\lambda_0].$$

*Importantly, if* $\epsilon < \frac{1}{\delta^q}$, *both* $\lambda_0$ *and* $\rho_0$ *are independent of the value of* $\epsilon$.

**Proof of Lemma 6.** To prove the lemma, we have adopted the same strategy as in the proof of Lemma D1 from Le & Malick (2024). Observing the definition of $hf^\epsilon_\theta$, we clearly see that $f^\epsilon_\theta(z) > f(z)$. Since for any $x \in \mathcal{X}$, the function $f^\epsilon_\theta(\cdot) - \lambda c^q(\cdot, z)$ is continuous, we can invoke the envelope theorem (Corollary 1, section 2.8 in Clarke (1990)). Consequently, the right-sided derivative of the function $f^\epsilon_{\theta,\lambda}$ with respect to $\lambda$, is given by:

$$\partial^+_\lambda f^\epsilon_{\theta,\lambda}(z) = -\min\left\{d^q(z',z) : z' \in \arg\max_{\mathcal{Z}}\{f^\epsilon_\theta(\cdot) - \lambda c^q(\cdot,z)\}\right\}.$$

Let define for any compact set $S \subseteq \mathcal{Z}$, the distance to set $c_*(z,S) := \min\{c(z,s) : s \in S\}$. By integrating and subsequently taking the infimum over $\mathcal{F}^\epsilon$, we have:

$$\rho^\epsilon_{\max}(\lambda) = \inf_{\theta \in \Theta} \mathbb{E}_{z \sim \mathbb{P}}\left[c^q_*\left(z, \arg\max_{\mathcal{Z}}\{f^\epsilon_\theta(\cdot) - \lambda c^q(\cdot,z)\}\right)\right]. \tag{C}$$

we define $\rho^\epsilon_0$ as below:

$$\rho^\epsilon_0 = \inf_{\theta \in \Theta} \mathbb{E}_{x \sim \mathbb{P}}\left[\min\{d^q(z,z') : z' \in \arg\max_{\mathcal{Z}} f^\epsilon_\theta(\cdot)\}\right]$$

$$= \inf_{\theta \in \Theta} \mathbb{E}_{x \sim \mathbb{P}}\left[\min\{d^q(z,z') : z' \in \arg\max_{\mathcal{Z}} f(\cdot)\}\right] = \inf_{\theta \in \Theta}\left\{\mathbb{E}_{x \sim \mathbb{P}_0}[d^q_+(x)] + \mathbb{E}_{x \sim \mathbb{P}_1}[d^q_-(x)]\right\}.$$

Thus, by the very construction of $f^\epsilon_\theta$, the critical constant $\rho^\epsilon_0$ does not depend on the choice of $\epsilon$, remaining invariant for all $\epsilon$. So we use $\rho_0$ notation from now on.

To establish the result, it suffices to demonstrate that for any positive sequence $(\lambda_k)_{k\in\mathbb{N}}$ approaching $0$ as $k \to \infty$, the following holds $\liminf_{k\to\infty} \rho^\epsilon_{\max}(\lambda_k) \geq \rho_0$. The functions $\mathbb{E}_{z \sim \mathbb{P}}[f^\epsilon_{\theta,\lambda}(z)]$ are convex with respect to $\lambda$, so their right-hand derivatives $\mathbb{E}_{z \sim \mathbb{P}}[-\partial^+_\lambda f^\epsilon_{\theta,\lambda}(z)]$ are nondecreasing. As a result, $\rho^\epsilon_{\max}$, defined as the infimum over these nondecreasing functions, is also nondecreasing. Hence, for any sequence $\lambda_k \to 0$, we have $\limsup_{k\to\infty} \rho_{\max}(\lambda_k) \leq \rho_{\max}(0)$. Now, suppose for the sake of contradiction that there exists an $\tau > 0$ and a sequence $(\lambda_k)_{k\in\mathbb{N}}$ in $\mathbb{R}_+$ with $\lambda_k \to 0$ as $k \to \infty$, such that $\rho^\epsilon_{\max}(\lambda_k) \leq \rho_0 - \tau$ for all $k \in \mathbb{N}$. From the definition of $\rho^\epsilon_{\max}$ in Eq. C, this implies that for each $k$, there exists an $f^\epsilon_{\theta_k}$ such that:

$$\mathbb{E}_{x \sim \mathbb{P}}\left[c^q_*\left(z, \arg\max_{\mathcal{Z}}\{f^\epsilon_{\theta_k}(\cdot) - \lambda d(\cdot,z)\}\right)\right] \leq \rho_0 - \frac{\tau}{2}.$$

Given the compactness of $\mathcal{F}^\epsilon$ under the $\|\cdot\|_\infty$ norm, we can assume the sequence $(f^\epsilon_{\theta_k})_{k\in\mathbb{N}}$ converges to some $f^\epsilon_\theta \in \mathcal{F}^\epsilon$. Specifically, for $z \in \mathcal{Z}$, the expression $f^\epsilon_{\theta_k} - \lambda_k d^q(\cdot, z)$ converges to $f^\epsilon_\theta$ as $k \to \infty$. Consider an arbitrary $z \in \mathcal{Z}$. The mapping $(\lambda, f^\epsilon_\theta) \mapsto \arg\max_{\mathcal{Z}}\{f^\epsilon_\theta - \lambda c^q(\cdot,z)\}$ is outer semicontinuous with compact values (By Lemma A.2 Le & Malick (2024)), and $d$ is jointly continuous. Thus, the mapping $(\lambda, f^\epsilon_\theta) \mapsto c_*(z, \arg\max_{\mathcal{Z}}\{f^\epsilon_\theta - \lambda c^q(\cdot,z)\})$ Is lower semicontinuous, according to Lemma A.1 Le & Malick (2024). Consequently:

$$\liminf_{k\to\infty} c_*(z, \arg\max_{\mathcal{Z}}\{f^\epsilon_{\theta_k} - \lambda_k d^q(\cdot,z)\}) \geq c_*(z, \arg\max_{\mathcal{Z}} f^\epsilon_\theta(\cdot)).$$

Taking the expectation over $z \sim \mathbb{P}$, we obtain:

$$\mathbb{E}_{z \sim \mathbb{P}}[c^q_*(z, \arg\max_{\mathcal{Z}} f^\epsilon_\theta(\cdot)] \leq \mathbb{E}_{z \sim \mathbb{P}}[\liminf_{k\to\infty} c^q_*(z, \arg\max_{\mathcal{Z}}\{f^\epsilon_{\theta_k} - \lambda_k d^q(\cdot,z)\})]$$

$$\leq \liminf_{k\to\infty} \mathbb{E}_{z \sim \mathbb{P}}[c^q_*(z, \arg\max_{\mathcal{Z}}\{f^\epsilon_{\theta_k} - \lambda_k d^q(\cdot,z)\})]$$

$$\leq \rho_0 - \frac{\epsilon}{2}.$$

However, since: $\rho_0 \leq \mathbb{E}_{z \sim \mathbb{P}}[c^q_*(z, \arg\max_{\mathcal{Z}} f^\epsilon_\theta)]$, this creates a contradiction; therefore, there exist $\lambda^\epsilon_0$ such that we have $\rho^\epsilon_{\max}(\lambda) \geq \frac{\rho_0}{4}$, for all $\lambda \in [0, 2\lambda^\epsilon_0]$.

To complete the proof, we know from Lemma 4, if $\lambda < \min(\frac{1}{p_0}, \frac{1}{p_1})\frac{2}{\epsilon^q}$, then $f^\epsilon_{\theta,\lambda} = f_\lambda$ As clearly evident, the definition of $\arg\max_{\mathcal{Z}}\{f^\epsilon_\theta(\cdot) - \lambda c^q(\cdot,z)\}$ is independent of $\epsilon$. Thus, the quantity $\lambda^\epsilon_0$ also does not depend on $\epsilon$ and remains valid for the entire family $\mathcal{F}^\epsilon$. $\square$

**Lemma 7 (Estimation of Distance).** *The approximation of distance to the decision boundary is expressed as:*

$$d_\theta(x) = \frac{|g_\theta(x)|}{\|\nabla_x g_\theta(x)\|_{q*}} + O(d_\theta(x)^2),$$

*Proof.* Let $x^*$ be the projection of $x$ on the decision boundary $\mathcal{L}_\theta$. Expanding $g_\theta(x^*)$ around projection of $x$ using a Taylor series:

$$g_\theta(x^*) = g_\theta(x) + \nabla_x g_\theta(x) \cdot (x^* - x) + \frac{1}{2}(x^* - x)^T \nabla^2 g_\theta(\xi)(x^* - x),$$

for some $\xi \in \mathbb{R}^d$. Since $g_\theta(x^*) = 0$ and $d_\theta(x) = \|x^* - x\|_q$, Thus the quadratic term is $O(\|x^* - x\|_q^2) = O(d_\theta(x)^2)$. Therefore:

$$0 = g_\theta(x) + \nabla_x g_\theta(x) \cdot (x^* - x) + O(d_\theta(x)^2).$$

Using Hölder's inequality again:

$$|g_\theta(x)| = \|\nabla_x g_\theta(x)\|_{q*} \cdot d_\theta(x) + O(d_\theta(x)^2).$$

Solving for $d_\theta(x)$:

$$d_\theta(x) = \frac{|g_\theta(x)|}{\|\nabla_x g_\theta(x)\|_{q*}} + O(d_\theta(x)^2).$$

$\square$

**Lemma 8 (Lipschitz Coefficient).** *Let $g_\theta(x)$ be $C^1$ in both $x \in \mathcal{X} \subset \mathbf{R}^n$ and $\theta \in \Theta \subset \mathbf{R}^d$ are compact and bounded set. Assume the quantitative regularity bounds*

$$M := \sup_{\theta \in \Theta, x \in \mathcal{X}} \left\|\nabla_\theta g_\theta(x)\right\|_{q*} < \infty, \qquad c := \inf_{\theta \in \Theta, x \in \mathcal{X} \mid f_\theta(x) \mid \leqslant \varepsilon} \left\|\nabla_x g_\theta(x)\right\|_{q*} > 0. \qquad \text{(D)}$$

*Then For all $\theta, \theta' \in \Theta$ and Lipschitz coefficient $L = \max(\frac{1}{p_0}, \frac{1}{p_1})\frac{qM}{c\varepsilon}$, we have:*

$$\|f_\theta^\varepsilon - f_{\theta'}^\varepsilon\|_\infty \leqslant L\|\theta - \theta'\|_q.$$

**Proof of Lemma 8.** By Eq. A, We can write the function

$$f_\theta^\varepsilon(z) = p_0^{-1}\left(1 - \frac{1}{\epsilon^q}d_+^q(x)\right)^+ \mathbb{1}_{S_0}(a, y) - p_1^{-1}\left(1 - \frac{1}{\epsilon^q}d_-^q(x)\right)^+ \mathbb{1}_{S_1}(a, y). \qquad \text{(E)}$$

Since the we just measure the distance in $\epsilon$-distance from boundary $\mathcal{L}_\theta$, by using Lemma 7, we can write:

$$d_{+\theta}(x) := \frac{g_\theta(x)}{\|\nabla_x g_\theta(x)\|_{q*}} + O(\epsilon^2) \quad \left(d_{+\theta}(x) = 0 \iff g_\theta(x) = 0\right).$$

where $q*$ is dual conjugate of $q$, i.e., $\frac{1}{q} + \frac{1}{q*} = 1$. Since the mapping $\vartheta \mapsto g_\vartheta(x)$ is differentiable,

$$g_{\theta_1}(x) - g_{\theta_2}(x) = \nabla_\vartheta g_{\bar\theta}(x) \cdot (\theta_1 - \theta_2) \quad \text{for some } \bar\theta \in [\theta_1, \theta_2].$$

Therefore $|g_{\theta_1}(x) - g_{\theta_2}(x)| \leqslant M\|\theta - \theta'\|_q$. If the $x_2^*$ is projection point of $x$ on decision boundary $\mathcal{L}_{\theta_2}$, the we have:

$$|g_{\theta_1}(x_2^*) - g_{\theta_2}(x_2^*)| = |g_{\theta_1}(x_2^*)| \leqslant M\|\theta - \theta'\|_q$$

Hence, we can calculate the distance $x_2^*$ to the new boundary $\mathcal{L}_{\theta_1}$ with an extra motion of length at most $\frac{M}{c}\|\theta_1 - \theta_2\|$. Thus, by the triangle inequality, we have:

$$d_{\theta_1}(x) \leqslant d_{\theta_2}(x) + \frac{M}{c}\|\theta_1 - \theta_2\|_q.$$

Interchanging $\theta_1$ and $\theta_2$ yields the reverse inequality, so

$$|d_{\theta_1}(x) - d_{\theta_2}(x)| \leqslant \frac{M}{c}\|\theta_1 - \theta_2\|_q \qquad \forall x, \theta_1, \theta_2.$$

Inside the smoothing part, $\rho_\varepsilon(t) := [1 - \varepsilon^{-q}t_+^q]_+$ has slope $\rho_\varepsilon'(t) = -q\varepsilon^{-q}t^{q-1}$, so $|\rho_\varepsilon'| \leqslant q/\varepsilon$. Because $\rho_\varepsilon$ is $(q/\varepsilon)$-Lipschitz and $(*)$ holds,

$$|\rho_\varepsilon(d_{+\theta_1}(x)) - \rho_\varepsilon(d_{+\theta_2}(x))| \leqslant \frac{qM}{c\varepsilon}\|\theta_1 - \theta_2\|_q.$$

So by combining this result in Eq. E, we can write

$$|f_{\theta_1}^\varepsilon(z) - f_{\theta_2}^\varepsilon(z)| \leqslant \frac{qM}{p_0 c\varepsilon}\|\theta_1 - \theta_2\|_q + \frac{qM}{p_1 c\varepsilon}\|\theta_1 - \theta_2\|_q.$$

So, the function $f_\theta^\varepsilon$ is Lipschitz with $L = \max(\frac{1}{p_0}, \frac{1}{p_1})\frac{2qM}{c\varepsilon}$ It completes the proof. $\square$

**Lemma 9** (**Entropy Integral for Lipschitz Classes**). *Let $\mathcal{F} = \{f_\theta : \theta \in \Theta\}, \Theta \subset \mathbf{R}^d$ compact, $D := \mathrm{diam}(\Theta)$. Assume that the parameter map is $L$–Lipschitz in the sup–norm, i.e.*

$$\|f_\theta - f_{\theta'}\|_\infty \leqslant L\|\theta - \theta'\|_2 \quad \forall \theta, \theta' \in \Theta.$$

*Denote by $\mathcal{I}_\mathcal{F} := \int_0^1 \sqrt{\log N(\mathcal{F}, \|\cdot\|_\infty, \delta)}\, d\delta$ Dudley's entropy integral. Then*

$$\mathcal{I}_\mathcal{F} \leqslant \sqrt{\pi}\, DL\sqrt{d}.$$

**Proof of Lemma 9.** First, we bound the covering numbers of the class $\mathcal{F}$. Since the map $\theta \mapsto f_\theta$ is $L$–Lipschitz in the supremum norm, for any $\theta, \theta' \in \Theta$,

$$\|f_\theta - f_{\theta'}\|_\infty \leqslant L\|\theta - \theta'\|_2.$$

Hence an $\varepsilon/L$–cover of $\Theta$ in $\|\cdot\|_2$ induces an $\varepsilon$–cover of $\mathcal{F}$ in $\|\cdot\|_\infty$. Thus

$$N\big(\mathcal{F}, \|\cdot\|_\infty, \varepsilon\big) \leqslant N\big(\Theta, \|\cdot\|_2, \varepsilon/L\big).$$

Since $\Theta \subset \mathbf{R}^d$ is compact of diameter $D$, the standard volumetric estimate gives, for $0 < \varepsilon \leqslant DL$,

$$N\big(\Theta, \|\cdot\|_2, \varepsilon/L\big) \leqslant \Big(\frac{2DL}{\varepsilon}\Big)^d,$$

and therefore

$$\log N\big(\mathcal{F}, \|\cdot\|_\infty, \varepsilon\big) \leqslant d\log\Big(\frac{2DL}{\varepsilon}\Big).$$

Dudley's entropy integral is

$$\mathcal{I}_\mathcal{F} = \int_0^1 \sqrt{\log N\big(\mathcal{F}, \|\cdot\|_\infty, \delta\big)}\, d\delta.$$

Substituting the bound on the covering numbers,

$$\mathcal{I}_\mathcal{F} \leqslant \sqrt{d}\int_0^1 \sqrt{\log\Big(\frac{2DL}{\delta}\Big)}\, d\delta.$$

Set $a := 2DL$ and make the change of variables $t = \log(a/\delta)$, so that $\delta = ae^{-t}$ and $d\delta = -ae^{-t}dt$. The integral becomes

$$\int_0^1 \sqrt{\log\Big(\frac{a}{\delta}\Big)}\, d\delta = a\int_{t=\log a}^\infty \sqrt{t}\, e^{-t}\, dt \leqslant a\int_0^\infty \sqrt{t}\, e^{-t}\, dt = a\,\Gamma\big(\tfrac{3}{2}\big) = a\frac{\sqrt{\pi}}{2}.$$

Hence

$$\mathcal{I}_\mathcal{F} \leqslant \sqrt{d}\frac{\sqrt{\pi}}{2}(2DL) = \sqrt{\pi}\, DL\sqrt{d}.$$

This completes the proof. $\qquad\qquad\square$

First of all it is easy to check that Assumption 1 is valid for family of $\mathcal{F}^\epsilon$, so By applying Theorem 9 (Theorem 3.1 Le & Malick (2024)) on the family of functions $\mathcal{F}^\epsilon$, and using Lemma 6, Lemma 9, Lemma 8, we can find $\rho_0$, $\lambda_0$, $\alpha$, and $\beta$ such that we have with probability at least $1 - \sigma$:

$$R_{\delta, \mathbb{P}^N}(f_\theta^\epsilon) \geqslant \mathbb{E}_{z \sim \mathbb{P}}\big[f_\theta^\epsilon(z)\big] \quad \text{for all } \theta \in \Theta, \tag{F}$$

Here $R_{\delta, \mathbb{P}}(f) := \sup_{\mathbb{Q} \in \mathcal{B}_\delta(\mathbb{P})} \mathbb{E}_{z \sim \mathbb{Q}}\big[f(z)\big]$. By replacing $f_\theta^\epsilon = f + g_\theta^\epsilon$ we can write $\mathbb{E}_{z \sim \mathbb{P}}\big[f_\theta^\epsilon(z)\big] = \mathbb{E}_{z \sim \mathbb{P}}\big[g_\theta^\epsilon(z)\big] + \mathbb{E}_{z \sim \mathbb{P}}\big[f(x)\big]$. By Lemma 5, we know $\lambda^* \leqslant \max(\frac{1}{p_0}, \frac{1}{p_1})\frac{2}{\delta^q}$, so if we set $\epsilon \leqslant \delta\max(\frac{1}{p_0}, \frac{1}{p_1})^{\frac{-1}{q}}$, so by Lemma 4, we can write $f_{\theta, \lambda}^\epsilon = f_\lambda$. By replacing it in the equation

$$R_{\delta, \mathbb{P}^N}(f) \geqslant \mathbb{E}_{z \sim \mathbb{P}}\big[f(z)\big] + \mathbb{E}_{z \sim \mathbb{P}}\big[g_\theta^\epsilon(z)\big] \implies R_{\delta, \mathbb{P}^N}(f) \geqslant \mathbb{E}_{z \sim \mathbb{P}}\big[f(z)\big] \quad \text{for all } \theta \in \Theta,$$

By the Theorem 9, we have:

$$\alpha = 48\Big(1 + \|\mathcal{F}^\epsilon\|_\infty + \frac{1}{\lambda_0}\Big)\Big(I_{\mathcal{F}^\epsilon} + \frac{2\|\mathcal{F}^\epsilon\|_\infty}{\lambda_0}\sqrt{2\log\frac{4}{\sigma}}\Big), \quad \beta = \frac{96 I_{\mathcal{F}^\epsilon}}{\lambda_0} + 48\frac{\|\mathcal{F}^\epsilon\|_\infty}{\lambda_0}\sqrt{2\log\frac{4}{\sigma}}.$$

Now by applying Lemma 9 and Lemma 8, we can write $\mathcal{I}_{\mathcal{F}^\epsilon} \leqslant \sqrt{\pi}\, D\max(\frac{1}{p_0}, \frac{1}{p_1})\frac{2qM}{c\epsilon}\sqrt{K}$. It is easy to check that $\|\mathcal{F}^\epsilon\|_\infty = 1$. So by setting $\epsilon \leqslant \delta\max(\frac{1}{p_0}, \frac{1}{p_1})^{\frac{-1}{q}}$, we can write

$$\alpha = 48\Big(2 + \frac{1}{\lambda_0}\Big)\Big(I_{\mathcal{F}^\epsilon} + \frac{2}{\lambda_0}\sqrt{2\log\frac{4}{\sigma}}\Big), \quad \beta = \frac{96 I_{\mathcal{F}^\epsilon}}{\lambda_0} + 48\frac{1}{\lambda_0}\sqrt{2\log\frac{4}{\sigma}}.$$

So by the Theorem 9 Le & Malick (2024), for $N > \frac{16(\alpha+\beta)^2}{\rho_0^2}$ and $\delta > \frac{\alpha}{\sqrt{N}}$ we can write

$$R_{\delta,\mathbb{P}^N}(f) \geqslant \mathbb{E}_{x\sim\mathbb{P}}\big[f(x)\big] \quad \text{for all } \theta \in \Theta,$$

But we need to tie up conditions, so we re-derive the relation between the radius parameter $\delta$ and the sample size $N$ from the five hypotheses.

$$A := 48\left(2 + \frac{1}{\lambda_0}\right), \quad B := \frac{2}{\lambda_0}\sqrt{2\ln\frac{4}{\sigma}}, \quad C := \frac{96}{\lambda_0}, \quad S := \frac{48}{\lambda_0}\sqrt{2\ln\frac{4}{\sigma}}, \quad M := AB + S,$$

$$\kappa := \frac{2\sqrt{\pi}DqM}{c}\max\left(\frac{1}{p_0}, \frac{1}{p_1}\right)\sqrt{K}, \quad \eta := \max\left(\frac{1}{p_0}, \frac{1}{p_1}\right)^{-1/q}, \quad E := \kappa/\eta, \quad L := (A+C)E.$$

Thus $\alpha = AI_{\mathcal{F}^\varepsilon} + AB$, and $\beta = CI_{\mathcal{F}^\varepsilon} + S$. The complexity term satisfies $I_{\mathcal{F}^\varepsilon} \leqslant \frac{\kappa}{\varepsilon}$ and for the value of $\epsilon$ gives $\epsilon \leqslant \delta\eta$. Choosing $\varepsilon = \delta\eta$ (the worst admissible value) yields $I_{\mathcal{F}^\varepsilon} \leqslant \frac{E}{\delta}$. So by choosing these coefficients, we have below upper bound for $\alpha$ and $\beta$

$$\alpha \leqslant \frac{AE}{\delta} + AB, \qquad \beta \leqslant \frac{CE}{\delta} + S \implies \alpha + \beta \leqslant \frac{L}{\delta} + M.$$

$\square$

**Proof of Proposition 5** The result follows by a direct application of Proposition 12 (from Proposition Le & Malick (2024)) to the function $f_\theta^\epsilon$. Indeed, Proposition 12 guarantees that, whenever

$$n > \frac{16\alpha^2}{\rho_{\text{crit}}^2} \quad \text{and} \quad \rho \leqslant \frac{\rho_{\text{crit}}}{4} - \frac{\alpha}{\sqrt{n}},$$

Then, with probability at least $1 - \sigma$, we have

$$R_{\delta,\mathbb{P}^N}(f_\theta^\epsilon) \leqslant R_{\rho+\alpha/\sqrt{n},\mathbb{P}}(f_\theta^\epsilon) \qquad \text{for all } f_\theta^\epsilon \in \mathcal{F}^\epsilon.$$

Moreover, from the proof of Theorem 4, we know that by setting

$$\epsilon \leqslant \delta\max\left(\frac{1}{p_0}, \frac{1}{p_1}\right)^{-1/q},$$

and invoking Lemma 4, one obtains $f_{\theta,\lambda}^\epsilon = f_\lambda$. Hence, under the same sample-size and margin-parameter conditions,

$$R_{\delta,\mathbb{P}^N}(f) \leqslant R_{\rho+\alpha/\sqrt{n},\mathbb{P}}(f) \qquad \text{for all } \theta \in \Theta.$$

which completes the proof. $\square$

**Proof of Proposition 6.** By proposition 10 if $\delta < \delta_{\mathcal{S}}$ then $\lambda^* > 0$. We assert that if $\lambda^* > 0$, then it implies that $(p_0\lambda^*)^{-1/q} \geqslant s_0^+$ or $(p_1\lambda^*)^{-1/q} \geqslant s_1^-$. Assume contrary if the $(p_0\lambda^*)^{-1/q} < s_0^+$ and $(p_1\lambda^*)^{-1/q} < s_1^-$ then it implies that $\mathbb{P}_0(\mathcal{R}_0^+) = 0$ and $\mathbb{P}_1(\mathcal{R}_1^-) = 0$ then by part (ii) of Theorem 7 for optimal coupling $\pi^*$ we have

$$\delta^q = \mathbb{E}_{(z,z')\sim\pi^*}\big[d^q(z,z')\big]$$

but $(p_0\lambda^*)^{-1/q} < s_0^+$ and $(p_1\lambda^*)^{-1/q} < s_1^-$ implies that $\mathbb{E}_{(z,z')\sim\pi^*}\big[d^q(z,z')\big] = 0$, therefore by contradiction we have $\lambda^{*-1/q} > \min(s_0^+\,p_0^{-1/q}, s_1^-\,p_1^{-1/q})$.

By assumption (iii) we have $\mathbb{P}(\mathcal{L}_\theta) = 0$ then it implies $\mathbb{P}_0(\partial_0^-) = \mathbb{P}_1(\partial_1^+) = 0$. Then by Theorem 7 we have:

$$\begin{aligned}
\delta^q = &\,\mathbb{P}_0(\mathcal{X}^-)\int_0^{(p_0\lambda^*)^{-1/q}} p_0 s^q\,\mathrm{d}G_0^-(s) + \mathbb{P}_1(\mathcal{X}^+)\int_0^{(p_1\lambda^*)^{-1/q}} p_1 s^q\,\mathrm{d}G_1^+(s) = \\
&\,\mathbb{P}_0(\mathcal{X}^-)\int_{s_0^+}^{(p_0\lambda^*)^{-1/q}} p_0 s^q\,\mathrm{d}G_0^-(s) + \mathbb{P}_1(\mathcal{X}^+)\int_{s_1^-}^{(p_1\lambda^*)^{-1/q}} p_1 s^q\,\mathrm{d}G_1^+(s),
\end{aligned} \tag{A}$$

By Theorem 2 it can be written:

$$
\begin{aligned}
\mathcal{S}_{\delta,q}(\mathbb{P},\theta) =& \mathbb{P}_0(\mathcal{X}^-)\int_0^{(p_0\lambda^*)^{-1/q}} 1 \ \mathrm{d}G_0^-(s) + \mathbb{P}_1(\mathcal{X}^+)\int_0^{(p_1\lambda^*)^{-1/q}} 1 \ \mathrm{d}G_1^+(s) = \\
& \mathbb{P}_0(\mathcal{X}^-)\int_{s_0^+}^{(p_0\lambda^*)^{-1/q}} 1 \ \mathrm{d}G_0^-(s) + \mathbb{P}_1(\mathcal{X}^+)\int_{s_1^-}^{(p_1\lambda^*)^{-1/q}} 1 \ \mathrm{d}G_1^+(s) = \\
& \mathbb{P}_0(\mathcal{X}^-)(G_0^-(p_0\lambda^*) - G_0^-(s_0^+)) + \mathbb{P}_1(\mathcal{X}^+)(G_1^+((p_1\lambda^*)^{-1/q}) - G_1^+(s_1^-))
\end{aligned}
\tag{B}
$$

With combining (A) and (B), it follows that:

$$
\min(p_0 s_0^{+q}, p_1 s_1^{-q}) \Big( \mathbb{P}_0(\mathcal{X}^-)(G_0^-(p_0\lambda^*) - G_0^-(s_0^+)) + \mathbb{P}_1(\mathcal{X}^+)(G_1^+((p_1\lambda^*)^{-1/q}) - G_1^+(s_1^-)) \Big) \leqslant \delta^q =
$$

$$
\mathbb{P}_0(\mathcal{X}^-)\int_{s_0^+}^{(p_0\lambda^*)^{-1/q}} p_0 s^q \ \mathrm{d}G_0^-(s) + \mathbb{P}_1(\mathcal{X}^+)\int_{s_1^-}^{(p_1\lambda^*)^{-1/q}} p_1 s^q \ \mathrm{d}G_1^+(s) \leqslant
$$

$$
\lambda^{*-1}\Big( \mathbb{P}_0(\mathcal{X}^-)(G_0^-(p_0\lambda^*) - G_0^-(s_0^+)) + \mathbb{P}_1(\mathcal{X}^+)(G_1^+((p_1\lambda^*)^{-1/q}) - G_1^+(s_1^-)) \Big),
$$

which implies that

$$
\lambda^*\delta^q \leqslant \mathcal{S}_{\delta,q}(\mathbb{P},\theta) \leqslant \frac{\delta^q}{\min(p_0 s_0^{+q}, p_1 s_1^{-q})}
$$

The last equation completes the proof. $\qquad\square$

**Proof of Theorem 5.** By Theorem 7 and Assumption $v$ we can write:

$$
\delta^q = \mathbb{P}_0(\mathcal{X}^-)\int_{s_0^+}^{(p_0\lambda^*)^{-1/q}} p_0 s^q \ \mathrm{d}G_0^-(s) + \mathbb{P}_1(\mathcal{X}^+)\int_{s_1^-}^{(p_1\lambda^*)^{-1/q}} p_1 s^q \ \mathrm{d}G_1^+(s) \geqslant
\tag{A}
$$

$$
p_0\mathbb{P}_0(\mathcal{X}^-)\int_{s_0^+}^{(p_0\lambda^*)^{-1/q}} s_0^{+q} g_0^-(s) \ \mathrm{d}s = p_0\mathbb{P}_0(\mathcal{X}^-)s_0^{+q}\int_0^{\eta_0} g_0^-(s_0^+ + s) \ \mathrm{d}s \leqslant
$$

$$
p_0\mathbb{P}_0(\mathcal{X}^-)s_0^{+q}\int_{s_0^+}^{\eta_0} (g_0^-(s_0^+) - L_0 s) \ \mathrm{d}s = p_0\mathbb{P}_0(\mathcal{X}^-)s_0^{+q}\left[ (g_0^-(s_0^+)\eta_0 - \frac{1}{2}L_0\eta_0^2) \right] \implies
\tag{B}
$$

$$
\frac{1}{2}p_0\mathbb{P}_0(\mathcal{X}^-)L_0\eta_0^2 - g_0^-(s_0^+)p_0\mathbb{P}_0(\mathcal{X}^-)\eta_0 + \delta^q s_0^{+-q} \geqslant 0
\tag{C}
$$

The Eq. A is obtained by Lipschitz property of $g_0^-$. Similarly by considering the second term in Eq. B we have below inequality such as Eq. C:

$$
\frac{1}{2}p_1\mathbb{P}_1(\mathcal{X}^+)L_1\eta_1^2 - g_1^+(s_1^-)p_1\mathbb{P}_1(\mathcal{X}^+)\eta_1 + \delta^q s_1^{--q} \geqslant 0
$$

where $\eta_0 = (p_0\lambda^*)^{-1/q} - s_0^+$ and $\eta_1 = (p_1\lambda^*)^{-1/q} - s_1^+$. When

$$
\delta \leqslant \left( \frac{1}{2L_0}g_0^-(s_0^+)^2 p_0\mathbb{P}_0(\mathcal{X}^-)s_0^+ \right)^{\frac{1}{q}}
\tag{D}
$$

The inequality of is equivalent to either

$$
\eta_0 \geqslant \frac{g_0^-(s_0^+)p_0\mathbb{P}_0(\mathcal{X}^-) + \sqrt{(g_0^-(s_0^+)p_0\mathbb{P}_0(\mathcal{X}^-))^2 - 2L_0 p_0\mathbb{P}_0(\mathcal{X}^-)s_0^{+-q}\delta^q}}{L_0 p_0\mathbb{P}_0(\mathcal{X}^-)},
\tag{E}
$$

$$
\eta_0 \leqslant \frac{g_0^-(s_0^+)p_0\mathbb{P}_0(\mathcal{X}^-) - \sqrt{(g_0^-(s_0^+)p_0\mathbb{P}_0(\mathcal{X}^-))^2 - 2L_0 p_0\mathbb{P}_0(\mathcal{X}^-)s_0^{+-q}\delta^q}}{L_0 p_0\mathbb{P}_0(\mathcal{X}^-)}.
\tag{F}
$$

If the condition E satisfies then $\eta_0 \geqslant g_0^-(s_0^+)L_0^{-1}$, So we have:

$$
\delta^q \geqslant p_0\mathbb{P}_0(\mathcal{X}^-)\int_{s_0^+}^{s_0^+ + \eta_0} s^q \ \mathrm{d}G_0^-(s)
$$

$$
\geqslant p_0\mathbb{P}_0(\mathcal{X}^-)\int_{s_0^+}^{s_0^+ + g_0^-(s_0^+)L_0^{-1}} s_0^{+q} \ \mathrm{d}G_0^-(s) \geqslant p_0\mathbb{P}_0(\mathcal{X}^-)s_0^{+q}G_0^-\left( s_0^+ + g_0^-(s_0^+)L_0^{-1} \right).
$$

Now by setting

$$\delta \leqslant \left(p_0 \mathbb{P}_0(\mathcal{X}^-) s_0^{+q} G_0^- \left(s_0^+ + g_0^-(s_0^+) L_0^{-1}\right)\right)^{\frac{1}{q}}, \tag{G}$$

the inequality E does not satisfy. Therefore for estimation $\lambda^*$ we consider the inequality F:

$$\eta_0 \leqslant \frac{g_0^-(s_0^+) p_0 \mathbb{P}_0(\mathcal{X}^-) - \sqrt{(g_0^-(s_0^+) p_0 \mathbb{P}_0(\mathcal{X}^-))^2 - 2 L_0 p_0 \mathbb{P}_0(\mathcal{X}^-) s_0^{+-q} \delta^q}}{L_0 p_0 \mathbb{P}_0(\mathcal{X}^-)} =$$

$$\frac{2 s_0^{+-q} \delta^q}{g_0^-(s_0^+) p_0 \mathbb{P}_0(\mathcal{X}^-) + \sqrt{(g_0^-(s_0^+) p_0 \mathbb{P}_0(\mathcal{X}^-))^2 - 2 L_0 p_0 \mathbb{P}_0(\mathcal{X}^-) s_0^{+-q} \delta^q}} \leqslant \frac{2 s_0^{+-q} \delta^q}{g_0^-(s_0^+) p_0 \mathbb{P}_0(\mathcal{X}^-)},$$

By proposition 6 we have $\mathcal{S}_{\delta,q}(\mathbb{P}, \theta) \geqslant \lambda^* \delta^q = \frac{1}{p_0}(s_0^+ + \eta_0)^{-q} \delta^q$. By using inequality

$$(1+x)^{-q} \geqslant 1 - qx$$

for $x \geqslant 0$ and $p \geqslant 1$, it follows that

$$(s_0^+ + \eta_0)^{-q} \delta^q \geqslant \delta^q \left(s_0^+ + \frac{2 s_0^{+-q} \delta^q}{g_0^-(s_0^+) p_0 \mathbb{P}_0(\mathcal{X}^-)}\right)^{-q} = \delta^q s_0^{+-q} \left(1 + \frac{2 s_0^{+-p-1} \delta^q}{g_0^-(s_0^+) p_0 \mathbb{P}_0(\mathcal{X}^-)}\right)^{-q}$$

$$\geqslant \delta^q s_0^{+-q} \left(1 - p \frac{2 s_0^{+-p-1} \delta^q}{g_0^-(s_0^+) p_0 \mathbb{P}_0(\mathcal{X}^-)}\right) = \delta^q s_0^{+-q} - 2q \left(g_0^-(s_0^+) p_0 \mathbb{P}_0(\mathcal{X}^-)\right)^{-1} s_0^{+-2q-1} \delta^{2q}$$

The last equality has a simple form:

$$\mathcal{S}_{\delta,q}(\mathbb{P}, \theta) \geqslant \frac{1}{p_0} \left(\delta^q s_0^{+-q} - 2q \left(g_0^-(s_0^+) p_0 \mathbb{P}_0(\mathcal{X}^-)\right)^{-1} s_0^{+-2q-1} \delta^{2q}\right) \tag{H}$$

By similar reasoning for $\delta^q > \mathbb{P}_1(\mathcal{X}^+) \int_{s_1^-}^{(p_1 \lambda^*)^{-1/q}} p_1 s^q \, dG_1^+(s)$ we have:

$$\mathcal{S}_{\delta,q}(\mathbb{P}, \theta) \geqslant \frac{1}{p_1} \left(\delta^q (s_1^-)^{-q} - 2q \left(g_1^+(s_1^-) p_1 \mathbb{P}_1(\mathcal{X}^+)\right)^{-1} (s_1^-)^{-2q-1} \delta^{2q}\right) \tag{I}$$

By combining the both equations H and I we have:

$$\mathcal{S}_{\delta,q}(\mathbb{P}, \theta) \geqslant \frac{\delta^q}{\min(p_0 s_0^{+q}, p_1 s_1^{-q})} - \frac{2q \delta^{2q}}{\min(p_0 s_0^{+2q+1} g_0^-(s_0^+) \mathbb{P}_0(\mathcal{X}^-), p_1(s_1^-)^{2q+1} g_1^+(s_1^-) \mathbb{P}_1(\mathcal{X}^+))}.$$

By setting $K = 2q \min\left(p_0 s_0^{+2q+1} g_0^-(s_0^+) \mathbb{P}(\mathcal{X}^-), p_1(s_1^-)^{2q+1} g_1^+(s_1^-) \mathbb{P}(\mathcal{X}^+)\right)^{-1}$ we have

$$\mathcal{S}_{\delta,q}(\mathbb{P}, \theta) \geqslant \frac{\delta^q}{\min(p_0 s_0^{+q}, p_1 s_1^{-q})} - K \delta^{2q}$$

Where $K$ depend only to the $\mathbb{P}$ and $q$. By combining the bounds in the equations D and G, to ensure that the above inequality is correct, we need that $\delta$ should be less than

$$\delta_0 = \min\left( \left(\frac{1}{2} L_0^{-1} g_0^-(s_0^+)^2 p_0 \mathbb{P}_0(\mathcal{X}^-) s_0^+\right)^{\frac{1}{q}}, \left(\frac{1}{2} L_1^{-1} g_1^+(s_1^-)^2 p_1 \mathbb{P}_1(\mathcal{X}^+) s_1^-\right)^{\frac{1}{q}}, \right.$$

$$\left. \left(p_0 \mathbb{P}_0(\mathcal{X}^-) s_0^{+q} G_0^- \left(s_0^+ + g_0^-(s_0^+) L_0^{-1}\right)\right)^{\frac{1}{q}}, \left(p_1 \mathbb{P}_1(\mathcal{X}^+) s_1^{+p} G_1^+ \left(s_1^- + g_1^+(s_1^-) L_1^{-1}\right)\right)^{\frac{1}{q}}\right).$$

The value of $\delta_0$ only depends on the $\mathbb{P}$ and $q$, and it completes the proof. $\qquad \square$

**Proof of Theorem 6.** Since the most interesting part of claim of Theorem 5 happens when $g_0^+(0) = g_1^-(0) \neq 0$, without loss of generality to have sharper upper bound, we suppose $g_0^+(0), g_1^-(0) > 0$, under Assumption $v$, there exist constants $0 < \delta_1 < \delta$ and $0 < C_1 \leqslant C_2 < \infty$ such that

$$0 < C_1 \leqslant g_0^+(s), g_1^-(s) \leqslant C_2 < \infty, \quad \forall s \in [0, \delta_1]$$

Hence, $g_0^+(s) \geqslant C_1$ on $[0, \delta_1]$. Let $\delta \leqslant \left(\frac{C_1}{q+1} \min\left(p_0 \mathbb{P}_0(\mathcal{X}^-), p_1 \mathbb{P}_0(\mathcal{X}^+)\right)\right)^{\frac{1}{q}} \delta_1^{\frac{q+1}{q}}$. We claim that $\lambda_*^{-1/q} \leqslant \min(p_0, p_1)^{\frac{1}{q}} \delta_1$. Suppose on the contrary that $\lambda_*^{-1/q} > \min(p_0, p_1)^{\frac{1}{q}} \delta_1$. Then

without loss generality if $p_0 = \min(p_0, p_1)$, then we have $(p_0\lambda^*)^{-1/q} < \delta_1$, so we can write

$$\delta^q = \mathbb{P}_0(\mathcal{X}^-)\int_0^{(p_0\lambda^*)^{-1/q}} p_0 s^q \, \mathrm{d}G_0^-(s) + \mathbb{P}_1(\mathcal{X}^+)\int_0^{(p_1\lambda^*)^{-1/q}} p_1 s^q \, \mathrm{d}G_1^+(s)$$

$$> \mathbb{P}_0(\mathcal{X}^-)\int_0^{(p_0\lambda^*)^{-1/q}} p_0 s^q \, \mathrm{d}G_0^-(s) > p_0\mathbb{P}_0(\mathcal{X}^-)\int_0^{\delta_1} s^q \, \mathrm{d}G_0^-(s) > p_0\mathbb{P}_0(\mathcal{X}^-)C_1\int_0^{\delta_1} s^q \, \mathrm{d}s$$

$$= \frac{C_1}{q+1}\left(p_0\mathbb{P}_0(\mathcal{X}^-)\right)\delta_1^{q+1} \implies \delta > \left(\frac{C_1}{q+1}\left(p_0\mathbb{P}_0(\mathcal{X}^-)\right)\right)^{\frac{1}{q}}\delta_1^{\frac{q+1}{q}}$$

The last equation contradicts by assumption about $\delta$, therefore $\lambda_*^{-1/q} \leqslant \min(p_0,p_1)^{\frac{1}{q}}\delta_1$. Let us define two functions.

$$F(\lambda) := \mathbb{P}_0(\mathcal{X}^-)\int_0^{(p_0\lambda)^{-1/q}} p_0 s^q \, \mathrm{d}G_0^-(s) + \mathbb{P}_1(\mathcal{X}^+)\int_0^{(p_1\lambda)^{-1/q}} p_1 s^q \, \mathrm{d}G_1^+(s)$$

$$G(\lambda) := p_0\mathbb{P}_0(\mathcal{X}^-)\int_0^{(p_0\lambda)^{-1/q}} s^q(g_0^-(0) - L_0 s) \, \mathrm{d}s + p_1\mathbb{P}_1(\mathcal{X}^+)\int_0^{(p_1\lambda)^{-1/q}} s^q(g_1^+(0) - L_1 s) \, \mathrm{d}s$$

$$= \frac{1}{q+1}\left(p_0^{-\frac{1}{q}}\mathbb{P}_0(\mathcal{X}^-)g_0^-(0) + p_1^{-\frac{1}{q}}\mathbb{P}_1(\mathcal{X}^+)g_1^+(0)\right)\lambda^{-\frac{q+1}{q}}$$

$$- \frac{1}{q+2}\left(p_0^{-\frac{2}{q}}\mathbb{P}_0(\mathcal{X}^-)L_0 + p_1^{-\frac{2}{q}}\mathbb{P}_1(\mathcal{X}^+)L_1\right)\lambda^{-\frac{q+2}{q}}$$

Both function $F(\lambda)$ and $G(\lambda)$ are strictly decreasing in the interval $(\delta_1^{-q}, +\infty)$ and we have $F(\lambda) > G(\lambda)$ by assumption $v$. Therefore we have $F(\lambda^*) > G(\lambda^*)$. Define $\tilde{\lambda}$ such that:

$$\tilde{\lambda} = \frac{1}{2}\left(p_0^{-\frac{1}{q}}\mathbb{P}_0(\mathcal{X}^-)g_0^-(0) + p_1^{-\frac{1}{q}}\mathbb{P}_1(\mathcal{X}^+)g_1^+(0)\right)^{\frac{q}{q+1}}(q+1)^{-\frac{q}{q+1}}\delta^{-\frac{p^2}{q+1}}$$

We want to ensure that $\tilde{\lambda} > \delta_1^{-q}$. To do that, it is sufficient to have the following condition:

$$\delta < 2^{-\frac{q+1}{p^2}}\left(p_0^{-\frac{1}{q}}\mathbb{P}_0(\mathcal{X}^-)g_0^-(0) + p_1^{-\frac{1}{q}}\mathbb{P}_1(\mathcal{X}^+)g_1^+(0)\right)^{\frac{1}{q}}(q+1)^{-\frac{1}{q}}\delta_1^{\frac{q+1}{q}} \tag{A}$$

We put $\tilde{\lambda}$ in the function $G$ so we have:

$$G(\tilde{\lambda}) = \quad 2^{\frac{q+1}{q}}\delta^q - \frac{1}{q+2}\left(p_0^{-\frac{2}{q}}\mathbb{P}_0(\mathcal{X}^-)L_0 + p_1^{-\frac{2}{q}}\mathbb{P}_1(\mathcal{X}^+)L_1\right)$$

$$\times \left(\frac{1}{2}\left(p_0^{-\frac{1}{q}}\mathbb{P}_0(\mathcal{X}^-)g_0^-(0) + p_1^{-\frac{1}{q}}\mathbb{P}_1(\mathcal{X}^+)g_1^+(0)\right)^{\frac{q}{q+1}}(q+1)^{-\frac{q}{q+1}}\delta^{-\frac{p^2}{q+1}}\right)^{-\frac{q+2}{q}}$$

$$= 2^{\frac{q+1}{q}}\delta^q - \frac{2^{\frac{q+2}{q}}(q+1)^{\frac{q+2}{q}}}{q+2}\left(p_0^{-\frac{2}{q}}\mathbb{P}_0(\mathcal{X}^-)L_0 + p_1^{-\frac{2}{q}}\mathbb{P}_1(\mathcal{X}^+)L_1\right)$$

$$\times \left(p_0^{-\frac{1}{q}}\mathbb{P}_0(\mathcal{X}^-)g_0^-(0) + p_1^{-\frac{1}{q}}\mathbb{P}_1(\mathcal{X}^+)g_1^+(0)\right)^{-\frac{q+2}{q+1}}\delta^{\frac{p(q+2)}{q+1}}$$

If we restrict the value of $\delta$ to:

$$\delta < (2^{\frac{(q+1)(q+2)}{p^2}} - 2^{-\frac{q+2}{q}})^{\frac{q+1}{q}}\frac{(q+1)^{-\frac{(q+1)(q+2)}{p^2}}}{(q+2)^{-\frac{q+1}{q}}}\left(p_0^{-\frac{2}{q}}\mathbb{P}_0(\mathcal{X}^-)L_0 + p_1^{-\frac{2}{q}}\mathbb{P}_1(\mathcal{X}^+)L_1\right)^{-\frac{q+1}{q}} \tag{B}$$

$$\times \left(p_0^{-\frac{1}{q}}\mathbb{P}_0(\mathcal{X}^-)g_0^-(0) + p_1^{-\frac{1}{q}}\mathbb{P}_1(\mathcal{X}^+)g_1^+(0)\right)^{\frac{q+2}{q}}$$

It results that $G(\tilde{\lambda}) > \delta^q$. Since $F(\lambda)$ is strictly decreasing on $(\delta_1^{-q}, +\infty)$, it results $\lambda_* > \tilde{\lambda}$. By this fact, we can write

$$
\mathcal{S}_{\delta,q}(\mathbb{P}, \theta) = \inf_{\mu > 0} \left\{ \mu^{-q} \delta^q + \mathbb{P}_0(\mathcal{X}^-) \int_0^{p_0^{-\frac{1}{q}}\mu} (1 - p_0 \mu^{-q} s^q) \, \mathrm{d}G_0^+(s) \right.
$$

$$
\left. + \mathbb{P}_1(\mathcal{X}^+) \int_0^{p_1^{-\frac{1}{q}}\mu} (1 - p_1 \mu^{-q} s^q) \, \mathrm{d}G_1^-(s) \right\}
$$

$$
= \inf_{0 < \mu < \tilde{\lambda}^{-\frac{1}{q}}} \left\{ \mu^{-q} \delta^q + \mathbb{P}_0(\mathcal{X}^-) \int_0^{p_0^{-\frac{1}{q}}\mu} (1 - p_0 \mu^{-q} s^q) \, \mathrm{d}G_0^+(s) \right.
$$

$$
\left. + \mathbb{P}_1(\mathcal{X}^+) \int_0^{p_1^{-\frac{1}{q}}\mu} (1 - p_1 \mu^{-q} s^q) \, \mathrm{d}G_1^-(s) \right\}
$$

$$
> \inf_{0 < \mu < \tilde{\lambda}^{-\frac{1}{q}}} \left\{ \mu^{-q} \delta^q + \mathbb{P}_0(\mathcal{X}^-) \int_0^{p_0^{-\frac{1}{q}}\mu} (1 - p_0 \mu^{-q} s^q)[g_0^+(0) - L_0 s] \, \mathrm{d}s \right.
$$

$$
\left. + \mathbb{P}_1(\mathcal{X}^+) \int_0^{p_1^{-\frac{1}{q}}\mu} (1 - p_1 \mu^{-q} s^q)[g_1^-(0) - L_1 s] \, \mathrm{d}s \right\}
$$

$$
= \inf_{0 < \mu < \tilde{\lambda}^{-\frac{1}{q}}} \left\{ \mu^{-q} \delta^q + \frac{q}{q+1} \left( g_0^+(0) \mathbb{P}_0(\mathcal{X}^-) p_0^{-\frac{1}{q}} + g_1^-(0) \mathbb{P}_1(\mathcal{X}^+) p_1^{-\frac{1}{q}} \right) \mu \right.
$$

$$
\left. - \frac{q}{2(q+2)} \left( L_0 \mathbb{P}_0(\mathcal{X}^-) p_0^{-\frac{2}{q}} + L_0 \mathbb{P}_1(\mathcal{X}^+) p_1^{-\frac{2}{q}} \right) \mu^2 \right\}
$$

$$
\geqslant \inf_{0 < \mu < \tilde{\lambda}^{-\frac{1}{q}}} \left\{ \mu^{-q} \delta^q + \frac{q}{q+1} \left( g_0^+(0) \mathbb{P}_0(\mathcal{X}^-) p_0^{-\frac{1}{q}} + g_1^-(0) \mathbb{P}_1(\mathcal{X}^+) p_1^{-\frac{1}{q}} \right) \mu \right\}
$$

$$
- \frac{q}{2(q+2)} \left( L_0 \mathbb{P}_0(\mathcal{X}^-) p_0^{-\frac{2}{q}} + L_0 \mathbb{P}_1(\mathcal{X}^+) p_1^{-\frac{2}{q}} \right) \tilde{\lambda}^2
$$

$$
= (q+1)^{\frac{1}{q+1}} \left( \mathbb{P}_0(\mathcal{X}^-) g_0^+(0) p_0^{-\frac{1}{q}} + \mathbb{P}_1(\mathcal{X}^+) g_1^-(0) p_1^{-\frac{1}{q}} \right)^{\frac{q}{q+1}} \delta^{\frac{q}{q+1}} -
$$

$$
\frac{2^{\frac{2-p}{q}} p}{(q+2)} (q+1)^{\frac{2}{q+1}} \left( \mathbb{P}_1(\mathcal{X}^+) L_0 p_0^{-\frac{2}{q}} + \mathbb{P}_0(\mathcal{X}^-) L_1 p_1^{-\frac{2}{q}} \right)
$$

$$
\times \left( \mathbb{P}_0(\mathcal{X}^-) g_0^+(0) p_0^{-\frac{1}{q}} + \mathbb{P}_1(\mathcal{X}^+) g_1^-(0) p_1^{-\frac{1}{q}} \right)^{\frac{-2}{q+1}} \delta^{\frac{2q}{q+1}} \tag{C}
$$

The result is valid when $\delta$ satisfies in two inequalities, A and B. By result of equations C we can write

$$
\mathcal{S}_{\delta,q}(\mathbb{P}, \theta) \geqslant (q+1)^{\frac{1}{q+1}} \left( \mathbb{P}_0(\mathcal{X}^-) g_0^+(0) p_0^{-\frac{1}{q}} + \mathbb{P}_1(\mathcal{X}^+) g_1^-(0) p_1^{-\frac{1}{q}} \right)^{\frac{q}{q+1}} \delta^{\frac{q}{q+1}} - C \delta^{\frac{2q}{q+1}}
$$

where $C = \zeta \left( \mathbb{P}_1(\mathcal{X}^+) L_0 p_0^{-\frac{2}{q}} + \mathbb{P}_0(\mathcal{X}^-) L_1 p_1^{-\frac{2}{q}} \right) \left( \mathbb{P}_0(\mathcal{X}^-) g_0^+(0) p_0^{-\frac{1}{q}} + \mathbb{P}_1(\mathcal{X}^+) g_1^-(0) p_1^{-\frac{1}{q}} \right)^{\frac{-2}{q+1}}$
and $\zeta = 2^{\frac{2-q}{q}} \frac{q}{(q+2)} (q+1)^{\frac{2}{q+1}}$. The above inequality is satisfied when

$$
\delta < \delta_0 = \min(p_0, p_1)^{-\frac{q+1}{p^2}} \rho^{\frac{q+1}{q}} (q+1)^{\frac{1}{q}} \left( \mathbb{P}_0(\mathcal{X}^-) g_0^+(0) p_0^{-\frac{1}{q}} + \mathbb{P}_1(\mathcal{X}^+) g_1^-(0) p_1^{-\frac{1}{q}} \right)^{\frac{-1}{q}}
$$

and it completes the proof. $\qquad\square$

**Proof of Proposition 7.** To find the maximum of the expectation of $\psi(x, a, y)$ over the ambiguity set $\mathcal{B}_\delta(\mathbb{P})$, we use strong duality Mohajerin Esfahani & Kuhn (2018b); Blanchet & Murthy (2019), which was explained before in Eq. 4.

With assumption (iv), we have

$$
c\big((x, a, y), (x', a', y')\big) = d(x, x') + \infty \cdot \mathbb{I}(a \neq a') + \infty \cdot \mathbb{I}(y \neq y'),
$$

so in the case $q \in [1, \infty)$ the conjugate function is obtained by
$$\psi_\lambda(x, a, y) = \sup_{x' \in \mathcal{X}} \left\{ \psi(x', a, y) - \lambda d^q(x, x') \right\}$$

Therefore, by the strong duality theorem, we can write
$$\sup_{\mathbb{Q} \in \mathcal{B}_\delta(\mathbb{P})} \mathbb{E}_\mathbb{Q}[\psi(x, a, y)] = \inf_{\lambda \geqslant 0} \left\{ \lambda \delta^q + \mathbb{E}_\mathbb{P} \left[ \sup_{x' \in \mathcal{X}} \left( \psi(x', a, y) - \lambda d^q(x, x') \right) \right] \right\}.$$

similarly for $q = \infty$ we can have:
$$\sup_{z' : c(z, z') \leqslant \delta} f(x', a', y') = \sup_{x' : d(x, x') \leqslant \delta} f(x', a, y)$$

By substituting the above equation into the strong duality theorem, the proof is completed.
$\square$

**Proof of proposition 8.** The proposition is a straightforward consequence of Lemma EC.6 Yang & Gao, 2022 once we impose the cost-function restriction set out in Assumption (iv) and use the strong duality theorem that is described in Proposition 7. $\square$

**Proof of Theorem 7.** To prove we use the Proposition 8. The formula of $\psi$ function is
$$\psi(z) = h_\theta(x) \left( p_0^{-1} \mathbb{1}_{S_0}(a, y) - p_1^{-1} \mathbb{1}_{S_1}(a, y) \right).$$

**(i)**

By Proposition 8, for $q = \infty$, there is a $\mathbb{P}$-measurable map $T^* : \mathcal{Z} \to \mathcal{Z}$ such that :
$$T^*(z) \in \left\{ (\tilde{x}, a, y) : \tilde{x} \in \arg\max_{x' \in \mathcal{X}} \{ \psi(x', a, y) : d(x', x) \leqslant \delta \} \right\}, \quad \mathbb{P} - \text{a.e.}$$

as in the proof of Theorem 2, by replacing the argument of $\psi(z)$, $T^*$ is obtained by solving for each $(a, y)$:
$$T_1^*(z) \in \arg\max_{x' \in \mathcal{X}} \{ h_\theta(x) \left( p_0^{-1} \mathbb{1}_{S_0}(a, y) - p_1^{-1} \mathbb{1}_{S_1}(a, y) \right) : d(x', x) \leqslant \delta \} \implies$$

$$T_1^*(z) \in \begin{cases} \mathcal{X}^+ & (x, a, y) \in \mathcal{X}^- \times S_0 \wedge d_+(x) < \delta, \\ \mathcal{X}^- & (x, a, y) \in \mathcal{X}^+ \times S_1 \wedge d_-(x) < \delta \end{cases} \implies \begin{cases} \mathbb{P}^*(\mathcal{X}^- \mid S_0) = \mathbb{P}(\mathcal{X}^- \backslash \mathcal{R}_0^- \mid S_0); \\ \mathbb{P}^*(\mathcal{X}^+ \mid S_1) = \mathbb{P}(\mathcal{X}^+ \backslash \mathcal{R}_1^+ \mid S_1) \end{cases}$$

where $T_1^*(z)$ is the value of first coordinate of $x$. For $q \in [1, \infty)$ and $\lambda^* = 0$, there is a $\mathbb{P}$-measurable map $T^*$ satisfying:
$$T^*(z) \in \arg\min_{z' \in \mathcal{Z}} \left\{ c(z, z') : z' \in \arg\max_{z \in \mathcal{Z}} \psi(z') \right\}, \quad \mathbb{P}\text{-a.e.} \implies T_1^*(z) \in \begin{cases} \mathcal{X}^+ & z \in \mathcal{X}^- \times S_0, \\ \mathcal{X}^- & z \in \mathcal{X}^+ \times S_1 \end{cases}$$

By the definition of $\mathcal{R}_a^+$, when $\lambda^* = 0$ then $\mathcal{R}_0^- = \mathcal{X}^-$ and similarly $\mathcal{R}_1^+ = \mathcal{X}^+$ so we have:
$$\begin{cases} \mathbb{P}^*(\mathcal{X}^- \mid S_0) = \mathbb{P}(\mathcal{X}^- \backslash \mathcal{R}_0^- \mid S_0) = 0; \\ \mathbb{P}^*(\mathcal{X}^+ \mid S_1) = \mathbb{P}(\mathcal{X}^+ \backslash \mathcal{R}_1^+ \mid S_1) = 0 \end{cases}$$

**(ii)** For $q \in [1, \infty)$ and $\lambda^* > 0$, there are $\mathbb{P}$-measurable maps $T^*$ and $T^-$ such that
$$T^*(z) \in \arg\max_{z' \in \mathcal{Z}} \left\{ c(z, z') : z' \in \arg\max_{\tilde{z} \in \mathcal{Z}} \psi(\tilde{z}) - \lambda^* c(z, \tilde{z})^q \right\} \implies \tag{A}$$

$$T_1^*(z) \in \begin{cases} x & z \in \mathcal{X} \backslash \mathcal{R}_0^- \times S_0, \\ \arg\min_{x' \in \mathcal{X}^+} d(x, x'), & z \in \mathcal{R}_0^- \times S_0 \end{cases}, \quad T_1^*(z) \in \begin{cases} x & z \in \mathcal{X} \backslash \mathcal{R}_1^+ \times S_1, \\ \arg\min_{x' \in \mathcal{X}^+} d(x, x'), & z \in \mathcal{R}_1^- \times S_1 \end{cases}$$

$$T^-(z) \in \arg\min_{z' \in \mathcal{Z}} \left\{ c(z, z') : z' \in \arg\max_{\tilde{z} \in \mathcal{Z}} \psi(\tilde{z}) - \lambda^* c(z, \tilde{z})^q \right\} \implies \tag{B}$$

$$T_1^-(z) \in \begin{cases} x & z \in \mathcal{X} \backslash \mathcal{R}_0^- \times S_0, \\ \arg\min_{x' \in \mathcal{X}^+} d(x, x'), & z \in \mathcal{R}_0^- \backslash \partial_0^- \times S_0, \\ x, & x \in \partial_0^- \times S_0 \end{cases}, \quad T_1^-(z) \in \begin{cases} x & z \in \mathcal{X} \backslash \mathcal{R}_1^+ \times S_1, \\ \arg\min_{x' \in \mathcal{X}^+} d(x, x'), & z \in \mathcal{R}_1^+ \backslash \partial_1^+ \times S_1, \\ x, & z \in \partial_1^+ \times S_1 \end{cases}$$

Define $t^*$ as the largest number in $[0, 1]$ such that:
$$\delta^q = t^* \mathop{\mathbb{E}}_{z \sim \mathbb{P}}[d^q(T^*(z), z)] + (1 - t^*) \mathop{\mathbb{E}}_{z \sim \mathbb{P}}[d^q(T^-(z), z)].$$

Then, $\mathbb{P}^* := t^* T_\#^* \mathbb{P} + (1 - t^*) T_\#^- \mathbb{P}$ is a worst-case distribution. Moreover if define $\mathcal{Z}^* = \mathcal{R}_0^+ \times S_0 \bigcup \mathcal{R}_1^- \times S_1$, then it can be easily to check for optimal coupling $\pi^*$ we have:
$$\{(z, \mathcal{T}^-(z)) : z \in \mathcal{Z}^*\} \subseteq \mathrm{supp}(\pi^*) \subseteq \{(z, \mathcal{T}^*(z)) : z \in \mathcal{Z}^*\}.$$
By using equations A and B it is easily to find that:
$$\mathbb{P}^*(\mathcal{X}^- \mid S_0) = \mathbb{P}(\mathcal{X}^- \backslash \mathcal{R}_0^- \mid S_0) + (1 - t^*)\mathbb{P}(\partial_0^- \mid S_0)$$
$$\mathbb{P}^*(\mathcal{X}^+ \mid S_1) = \mathbb{P}(\mathcal{X}^+ \backslash \mathcal{R}_1^+ \mid S_1) + (1 - t^*)\mathbb{P}(\partial_1^+ \mid S_1)$$
The last equation completes the proof. $\qquad\square$

**Proof of Proposition 9.** Let $\mathbb{P}^*$ is the worst-case distribution for finding the $\mathcal{S}_{\delta,q}(\mathbb{P}, \theta)$. By applying it on the formulation of fairness score 10, and Theorem 7 we have:
$$\mathcal{S}_{\delta,q}(\mathbb{P}, \theta) = \mathbb{E}_{\mathbb{P}^*}[f(z)] = \mathbb{P}^*(\mathcal{X}^- \mid S_0) + \mathbb{P}^*(\mathcal{X}^+ \mid S_1)$$
$$= \mathbb{P}(\mathcal{X}^- \backslash \mathcal{R}_0^- \mid S_0) + (1 - t^*)\mathbb{P}(\partial_0^- \mid S_0) + \mathbb{P}(\mathcal{X}^+ \backslash \mathcal{R}_1^+ \mid S_1) + (1 - t^*)\mathbb{P}(\partial_1^+ \mid S_1)$$
Similarly, by swapping the indices of 0 to 1, we can obtain
$$\mathcal{I}_{\delta,q}(\mathbb{P}, \theta) = \mathbb{P}_1(\mathcal{R}_1^- \backslash \partial_1^-) + (1 - t^*)\mathbb{P}_1(\partial_1^-) + \mathbb{P}_0(\mathcal{R}_0^+ \backslash \partial_0^+) + (1 - t^*)\mathbb{P}_0(\partial_0^+)$$
$$\square$$

**Proof of Proposition 10.** Let $\mathbb{P}^*$ be a worst-case distribution. If $\delta \geqslant \delta_{\mathcal{S}}$,
$$\mathcal{S}_{\delta,q}(\mathbb{P}, \theta) \tag{A}$$
$$= \inf_{\lambda \geqslant 0}\left\{\lambda\delta^q + \mathbb{E}_{x \sim \mathbb{P}_0}\left[\mathbb{1}_{\mathcal{X}^-}(x)(1 - p_0\lambda d_+^q(x))^+\right] + \mathbb{E}_{x \sim \mathbb{P}_1}\left[\mathbb{1}_{\mathcal{X}^+}(x)(1 - p_1\lambda d_-^q(x))^+\right]\right\}$$
$$= \inf_{\lambda \geqslant 0}\left\{\lambda\delta^q + \mathbb{E}_{x \sim \mathbb{P}_0}\left[(1 - h_\theta(x))(1 - p_0\lambda d_+^q(x))^+\right] + \mathbb{E}_{x \sim \mathbb{P}_1}\left[h_\theta(x)(1 - p_1\lambda d_-^q(x))^+\right]\right\}$$
$$= \inf_{\lambda \geqslant 0}\left\{\lambda\delta^q + \mathbb{E}_{x \sim \mathbb{P}_0}\left[(1 - h_\theta(x))(1 - \min(1, p_0\lambda d_+^q(x)))\right]\right.$$
$$\left. + \mathbb{E}_{x \sim \mathbb{P}_1}\left[h_\theta(x)(1 - \min(1, p_1\lambda d_-^q(x)))\right]\right\}$$
$$= \inf_{\lambda \geqslant 0}\left\{\lambda\delta^q - \mathbb{E}_{x \sim \mathbb{P}_0}\left[(1 - h_\theta(x))\min(1, p_0\lambda d_+^q(x))\right] - \mathbb{E}_{x \sim \mathbb{P}_1}\left[h_\theta(x)\min(1, p_1\lambda d_-^q(x))\right]\right\}$$
$$+ \mathbb{E}_{\mathbb{P}}\left[p_0^{-1}(1 - h_\theta(x)) + p_1^{-1}h_\theta(x)\right]$$
By definition of $\delta_{\mathcal{S}}$:
$$\mathbb{E}_{x \sim \mathbb{P}_0}\left[(1 - h_\theta(x))\min(1, p_0\lambda d_+^q(x))\right] + \mathbb{E}_{x \sim \mathbb{P}_1}\left[h_\theta(x)\min(1, p_1\lambda d_-^q(x))\right]$$
$$\leqslant \mathbb{E}_{x \sim \mathbb{P}_0}\left[(1 - h_\theta(x))p_0\lambda d_+^q(x)\right] + \mathbb{E}_{x \sim \mathbb{P}_1}\left[h_\theta(x)p_1\lambda d_-^q(x)\right] = \lambda\delta_{\mathcal{S}}^q \tag{B}$$
Let $\delta \geqslant \delta_{\mathcal{S}}$. By applying Eq. B in Eq. A, we have:
$$\lambda\delta^q - \mathbb{E}_{x \sim \mathbb{P}_0}\left[(1 - h_\theta(x))\min(1, p_0\lambda d_+^q(x))\right] - \mathbb{E}_{x \sim \mathbb{P}_1}\left[h_\theta(x)\min(1, p_1\lambda d_-^q(x))\right] \geqslant \lambda(\delta^q - \delta_{\mathcal{S}}^q) \geqslant 0$$
so the infimum happens when $\lambda^* = 0$.

Now consider the case $\delta < \delta_{\mathcal{S}}$. By proof by contradiction, suppose $\lambda^* = 0$, so by previous part, we have:
$$\inf_{\lambda \geqslant 0}\left\{\lambda\delta^q - \mathbb{E}_{x \sim \mathbb{P}_0}\left[(1 - h_\theta(x))\min(1, p_0\lambda d_+^q(x))\right] - \mathbb{E}_{x \sim \mathbb{P}_1}\left[h_\theta(x)\min(1, p_1\lambda d_-^q(x))\right]\right\} = 0 \tag{C}$$
Let $\epsilon := \delta_{\mathcal{S}}^q - \delta^q$, so by assumption we have $\epsilon > 0$. By the definition of $\delta_{\mathcal{S}}$,
$$\delta_{\mathcal{S}} = \left(p_0\mathbb{E}_{\mathbb{P}_0}\left[(1 - h_\theta(x))d_+^q(x)\right] + p_1\mathbb{E}_{\mathbb{P}_1}\left[h_\theta(x)d_-^q(x)\right]\right)^{\frac{1}{q}} < \infty,$$
By Billingsley (2013) Applying Dominated Convergence Theorem, we can find the constant $M$, such that
$$p_0\mathbb{E}_{\mathbb{P}_0}\left[(1 - h_\theta(x))d_+^q(x)\mathbb{I}(d_+^q(x) > M)\right] + p_1\mathbb{E}_{\mathbb{P}_1}\left[h_\theta(x)d_-^q(x)\mathbb{I}(d_-^q(x) > M)\right] < \frac{\epsilon}{2}$$

So if we put $\lambda < 1/M$, so for $\lambda$ we have:

$$\lambda\delta^q - \mathop{\mathbb{E}}_{x\sim\mathbb{P}_0}\left[(1-h_\theta(x))\min(1,p_0\lambda d_+^q(x))\right] - \mathop{\mathbb{E}}_{x\sim\mathbb{P}_1}\left[h_\theta(x)\min(1,p_1\lambda d_-^q(x))\right]$$

$$= \lambda\delta^q - \lambda\delta_\mathcal{S}^q + \mathop{\mathbb{E}}_{x\sim\mathbb{P}_0}\left[(1-h_\theta(x))\max(1,p_0\lambda d_+^q(x))\right] + \mathop{\mathbb{E}}_{x\sim\mathbb{P}_1}\left[h_\theta(x)\max(1,p_1\lambda d_-^q(x))\right]$$

$$\leqslant -\lambda\epsilon + \lambda\left(p_0\mathbb{E}_{\mathbb{P}_0}\left[(1-h_\theta(x))d_+^q(x)\mathbb{I}(d_+^q(x) > M)\right] + p_1\mathbb{E}_{\mathbb{P}_1}\left[h_\theta(x)d_-^q(x)\mathbb{I}(d_-^q(x) > M)\right]\right)$$

$$< -\lambda\frac{\epsilon}{2}.$$

Therefore, we can find $\lambda$ such that the inf of Eq. C is less than zero, so by contradiction, we can prove that $\lambda^* > 0$. $\qquad\square$

**Proof of Theorem 8.** First of all, it is easy to check that:

$$\mathcal{F}(\mathbb{P},\theta) = \mathop{\mathbb{E}}_{z\sim\mathbb{P}}\left[h_\theta(x)\left(\frac{\mathbb{1}_{S_0}(a,y)}{\mathbb{E}_\mathbb{P}[\mathbb{1}_{S_0}]} - \frac{\mathbb{1}_{S_1}(a,y)}{\mathbb{E}_\mathbb{P}[\mathbb{1}_{S_1}]}\right)\right] = \mathop{\mathbb{E}}_{x\sim\mathbb{P}_0}[h_\theta(x)] - \mathop{\mathbb{E}}_{x\sim\mathbb{P}_1}[h_\theta(x)]$$

$$= 1 - \mathop{\mathbb{E}}_{x\sim\mathbb{P}_0}\left[(1-h_\theta)(x)\right] - \mathop{\mathbb{E}}_{x\sim\mathbb{P}_1}[h_\theta(x)]$$

So by substituting the above equation beside equations from the proof of Theorem 2, We can write:

$$\sup_{\mathbb{Q}\in\mathcal{B}_\delta(\mathbb{P})}\mathbb{E}_\mathbb{Q}[\psi(z)] = \mathcal{S}_{\delta,q}(\mathbb{P},\theta) + \mathcal{F}(\mathbb{P},\theta) \leqslant \varepsilon \iff 1 - \mathop{\mathbb{E}}_{x\sim\mathbb{P}_0}\left[(1-h_\theta)(x)\right] - \mathop{\mathbb{E}}_{x\sim\mathbb{P}_1}[h_\theta(x)] +$$

$$\inf_{\lambda\geqslant 0}\left\{\lambda\delta^q + \mathop{\mathbb{E}}_{z\sim\mathbb{P}}\left[\mathbb{1}_{\mathcal{X}^-\times S_0}(z)\left(p_0^{-1} - \lambda d_+^q(x)\right)^+ + \mathbb{1}_{\mathcal{X}^+\times S_1}(z)\left(p_1^{-1} - \lambda d_-^q(x)\right)^+\right]\right\} \leqslant \varepsilon$$

First, we show the direct implication. For each $\varepsilon' > \varepsilon$ there exist $\lambda > 0$ such that

$$1 - \mathop{\mathbb{E}}_{x\sim\mathbb{P}_0}\left[(1-h_\theta)(x)\right] - \mathop{\mathbb{E}}_{x\sim\mathbb{P}_1}[h_\theta(x)] +$$

$$\lambda\delta^q + \mathop{\mathbb{E}}_{z\sim\mathbb{P}}\left[\mathbb{1}_{\mathcal{X}^-\times S_0}(z)\left(p_0^{-1} - \lambda d_+^q(x)\right)^+ + \mathbb{1}_{\mathcal{X}^+\times S_1}(z)\left(p_1^{-1} - \lambda d_-^q(x)\right)^+\right] \leqslant \varepsilon' \implies$$

$$1 - \mathop{\mathbb{E}}_{x\sim\mathbb{P}_0}\left[(1-h_\theta)(x)\right] - \mathop{\mathbb{E}}_{x\sim\mathbb{P}_1}[h_\theta(x)] +$$

$$\lambda\delta^q + \mathop{\mathbb{E}}_{x\sim\mathbb{P}_0}\left[(1-h_\theta)(x)\left(1-\lambda p_0 d_+^q(x)\right)^+\right] + \mathop{\mathbb{E}}_{x\sim\mathbb{P}_1}\left[h_\theta(x)\left(1-\lambda p_1 d_-^q(x)\right)^+\right] \leqslant \varepsilon' \implies$$

$$\lambda\delta^q - \mathop{\mathbb{E}}_{x\sim\mathbb{P}_0}\left[(1-h_\theta)(x)\min(\lambda p_0 d_+^q(x),1)\right] - \mathop{\mathbb{E}}_{x\sim\mathbb{P}_1}\left[h_\theta(x)\min(\lambda p_1 d_-^q(x),1)\right] \leqslant \varepsilon' - 1 \implies$$

$$\lambda\delta^q \leqslant \mathop{\mathbb{E}}_{x\sim\mathbb{P}_0}\left[(1-h_\theta)(x)\min(\lambda p_0 d_+^q(x),1)\right] + \mathop{\mathbb{E}}_{x\sim\mathbb{P}_1}\left[h_\theta(x)\min(\lambda p_1 d_-^q(x),1)\right] - (1-\varepsilon') \implies$$

$$\delta^q \leqslant \mathop{\mathbb{E}}_{x\sim\mathbb{P}_0}\left[(1-h_\theta)(x)\min(p_0 d_+^q(x),t)\right] + \mathop{\mathbb{E}}_{x\sim\mathbb{P}_1}\left[h_\theta(x)\min(p_1 d_-^q(x),t)\right] - t(1-\varepsilon') \implies$$

$$\delta^q \leqslant \sup_{t\in\mathbb{R}^+}\left\{\mathop{\mathbb{E}}_{x\sim\mathbb{P}_0}\left[(1-h_\theta)(x)\min(p_0 d_+^q(x),t)\right] + \mathop{\mathbb{E}}_{x\sim\mathbb{P}_1}\left[h_\theta(x)\min(p_1 d_-^q(x),t)\right] - t(1-\varepsilon')\right\} \implies$$

$$\delta^q \leqslant \inf_{t\in\mathbb{R}^+}\left\{t(1-\varepsilon') - \mathop{\mathbb{E}}_{x\sim\mathbb{P}_0}\left[(1-h_\theta)(x)\min(p_0 d_+^q(x),t)\right] - \mathop{\mathbb{E}}_{x\sim\mathbb{P}_1}\left[h_\theta(x)\min(p_1 d_-^q(x),t)\right]\right\}$$

$$\implies \delta^q \leqslant \inf_{t\in\mathbb{R}^+}\left\{(1-\varepsilon')t - \Psi_\mathcal{S}(t)\right\}$$

In the above, dividing both sides by $\lambda$ and replacing $t = \frac{1}{\lambda}$ and by definition of $\Psi_\mathcal{S}(t)$, the last equation is obtained. The concave conjugate of a function $\Psi_\mathcal{S}(t)$ is defined as $\Psi_\mathcal{S}^*(s) = \inf_t\{ts - \phi(t)\}$. By a similar reasoning, of theorem implies:

$$\sup_{\mathbb{Q}\in\mathcal{B}_\delta(\mathbb{P})}\mathbb{E}_\mathbb{Q}[\psi(z)] \leqslant \varepsilon \implies \Psi_\mathcal{S}^*(1-\varepsilon) \geqslant \delta^q$$

Now we prove the reverse by contradiction assumption that $\sup_{\mathbb{Q}\in\mathcal{B}_\delta(\mathbb{P})}\mathbb{E}_\mathbb{Q}[\psi(z)] > \varepsilon$, then it implies there exist $\varepsilon' > \varepsilon$ such that $\sup_{\mathbb{Q}\in\mathcal{B}_\delta(\mathbb{P})}\mathbb{E}_\mathbb{Q}[\psi(z)] \geqslant \varepsilon'$. We set $\kappa = \varepsilon' - \varepsilon > 0$. By

strong duality theorem, for all $\lambda > 0$ we have:

$$\lambda\delta^q > \mathop{\mathbb{E}}_{x\sim\mathbb{P}_0}\left[(1-h_\theta)(x)\min(\lambda p_0 d_+^q(x),1)\right] + \mathop{\mathbb{E}}_{x\sim\mathbb{P}_1}\left[h_\theta(x)\min(\lambda p_1 d_-^q(x),1)\right] - (1-\varepsilon') \implies$$

$$\lambda\delta^q - \kappa > \mathop{\mathbb{E}}_{x\sim\mathbb{P}_0}\left[(1-h_\theta)(x)\min(\lambda p_0 d_+^q(x),1)\right] + \mathop{\mathbb{E}}_{x\sim\mathbb{P}_1}\left[h_\theta(x)\min(\lambda p_1 d_-^q(x),1)\right] - (1-\varepsilon) \implies$$

$$\delta^q - \kappa t \geqslant \mathop{\mathbb{E}}_{x\sim\mathbb{P}_0}\left[(1-h_\theta)(x)\min(p_0 d_+^q(x),t)\right] + \mathop{\mathbb{E}}_{x\sim\mathbb{P}_1}\left[h_\theta(x)\min(p_1 d_-^q(x),t)\right] - (1-\varepsilon)t \implies$$

$$\delta^q - \kappa t \geqslant \Psi_\mathcal{S}(t) - (1-\varepsilon)t \implies \sup_t\{\Psi_\mathcal{S}(t) - (1-\varepsilon)t\} < \delta^q \implies \Psi_\mathcal{S}^*(1-\varepsilon) < \delta^q.$$

The last equation happens because the $\lambda^* > 0$, so $t^*$ the solution of optimization problem $\sup_t\{(1-\varepsilon)t - \Psi_\mathcal{S}(t)\}$ is greater than zero. By the above contradiction, the reverse proof is complete. The proof of the second part is totally similar to the first one. $\qquad\square$

**Proof of Proposition 11.** Let $\mathcal{Z} = \mathcal{X} \times \mathcal{A} \times \mathcal{Y}$ and recall that the cost $c\big((x,a,y),(x',a',y')\big) = d(x,x') + \infty\mathbb{I}(a \neq a') + \infty\mathbb{I}(y \neq y')$. Because a transport plan with finite cost must match the *labels* $(a,y)$ exactly, the $q$-Wasserstein metric induced by $d$ factorizes over the *finitely many* label pairs by proposition 3:

$$W_q\big(\mathbb{P},\mathbb{P}^N\big)^q = \sum_{(a,y)\in\mathcal{A}\times\mathcal{Y}} \mathbb{P}_{\mathbf{A},\mathbf{Y}}(a,y)W_q\big(\mathbb{P}_{a,y},\mathbb{P}_{a,y}^N\big)^q,$$

where $\mathbb{P}_{a,y}$ is the conditional law of $X$ given $(\mathbf{A},\mathbf{Y}) = (a,y)$ and $\mathbb{P}_{a,y}^N$ its empirical counterpart.

Assumption (iv) gives a finite $q$-moment on $\mathcal{X}$ and compact support, so each $\mathbb{P}_{a,y}$ lives in a $d$-dimensional compact metric space. The sharp non-asymptotic bound of Fournier–Guillin (Theorem 2 in Fournier & Guillin (2015)) implies that for some constants $C_{a,y}, c_{a,y} > 0$

$$\mathbb{P}^{\otimes}\big\{W_q\big(\mathbb{P}_{a,y},\mathbb{P}_{a,y}^N\big) > t\big\} \leqslant C_{a,y}\exp\Big[-c_{a,y}Nt^{\max\{d,2q\}}\Big], \qquad t > 0.$$

Let $K := |\mathcal{A}\times\mathcal{Y}| < \infty$. By a union bound and $W_q(\mathbb{P},\mathbb{P}^N) \leqslant K^{1/q}\max_{a,y}W_p(\mathbb{P}_{a,y},\mathbb{P}_{a,y}^N)$,

$$\mathbb{P}^{\otimes}\Big\{W_q(\mathbb{P},\mathbb{P}^N) > \delta\Big\} \leqslant KC_{\max}\exp\Big[-c_{\min}N(\delta/K^{1/q})^{\max\{d,2q\}}\Big],$$

where $C_{\max} := \max_{a,y}C_{a,y}$ and $c_{\min} := \min_{a,y}c_{a,y}$. Choose

$$\delta(N,\varepsilon) = \Big(\frac{K^{1/q}}{c_{\min}N}\ln\big(C_{\max}K\varepsilon^{-1}\big)\Big)^{1/\max\{d,2q\}}.$$

Then the exponential tail above is at most $\varepsilon$, yielding $\mathbb{P}^{\otimes}\big(\mathbb{P} \in \mathcal{B}_q(\mathbb{P}^N,\delta(N,\varepsilon))\big) \geqslant 1 - \varepsilon$. Absorbing the (fixed) label and constant factors into a single $C = C(\mathbb{P},d)$ gives exactly the upper-bound scale $\delta \lesssim \big(N\ln(C\varepsilon^{-1})\big)^{-1/\max\{d,2q\}}$, proving Proposition 11. $\qquad\square$

# C   NUMERICAL STUDIES SUPPLEMENTARY

## A. DATASETS

To demonstrate the fragility of group-fairness notions, we apply Scenarios 1 and 2 across a wide range of models—including Gradient Boosting and AdaBoost. However, when evaluating our DRUNE algorithm, we restrict our experiments to logistic regression, linear and non-linear SVMs, and MLPs. We evaluate our distributionally robust fairness approach on several real-world datasets. Table 1 provides a comprehensive overview of the datasets used in our study.

## B. MODEL SPECIFICATIONS

We evaluate four classification models:

## C. EXPERIMENTAL SETUP

Table 1: Overview of datasets used in the study

| Dataset | Protected Attribute | Label |
|---|---|---|
| Adult Census | Gender (Male=1, Female=0) | Income >50K (1) vs ⩽50K (0) |
| ACS Income | SEX (Male=1, Female=0) | PINCP > median (1) else (0) |
| HELOC | Age (above median=1, below=0) | RiskPerformance (Good=0, Bad=1) |
| Bank Marketing | Age (⩾25=1, <25=0) | Term deposit (yes=1, no=0) |
| CelebA | Male (1) vs Female (0) | Smiling (1) vs Not Smiling (0) |
| Heritage Health | Sex (M=1, F=0) | DaysInHospital_Y2 > median (1) else (0) |
| Law School | Race (white=1, non-white=0) | Pass bar exam (1=passed, 0=failed) |
| MEPS | SEX (1=male, 2=female) | TOTEXP16 > median (1) else (0) |

Table 2: Model specifications and parameters

| Model | Parameters |
|---|---|
| Logistic Regression | `max_iter=1000`, L2 regularization |
| Linear SVM | `max_iter=1000`, linear kernel |
| Non-linear SVM | `kernel='rbf'`, `gamma=0.5` |
| Gradient Boosting | `n_estimators=100`, `learning_rate=0.1`, `max_depth=3` |
| AdaBoost | `n_estimators=100`, `learning_rate=1.0` |
| MLP | `max_iter=1000`, `solver='lbfgs'`, `tol=1e-4`, hidden layers (10,10) |

Table 3: Experimental parameters and settings

| Parameter | Value/Description |
|---|---|
| Data Splitting | 80/20 train/test split (random state=42) |
| Sample Size | 1000 instances per experiment |
| Sampling Strategy | Balanced between privileged/unprivileged groups |
| Robustness Parameter ($\delta$) | 0.001 |
| Distance Norm ($q$) | 2 (Euclidean) |
| Convergence Parameters | $\epsilon_y = 10^{-6}$, $\epsilon_g = 10^{-6}$ |
| Maximum Iterations ($K_{max}$) | 100 |
| Number of Experiments | 1000 independent runs |
| Performance Metrics | Accuracy, demographic parity, equalized odds, DRUNE regularizer |
| Statistical Analysis | Mean and standard deviation of metrics, confidence intervals, comparative analysis |

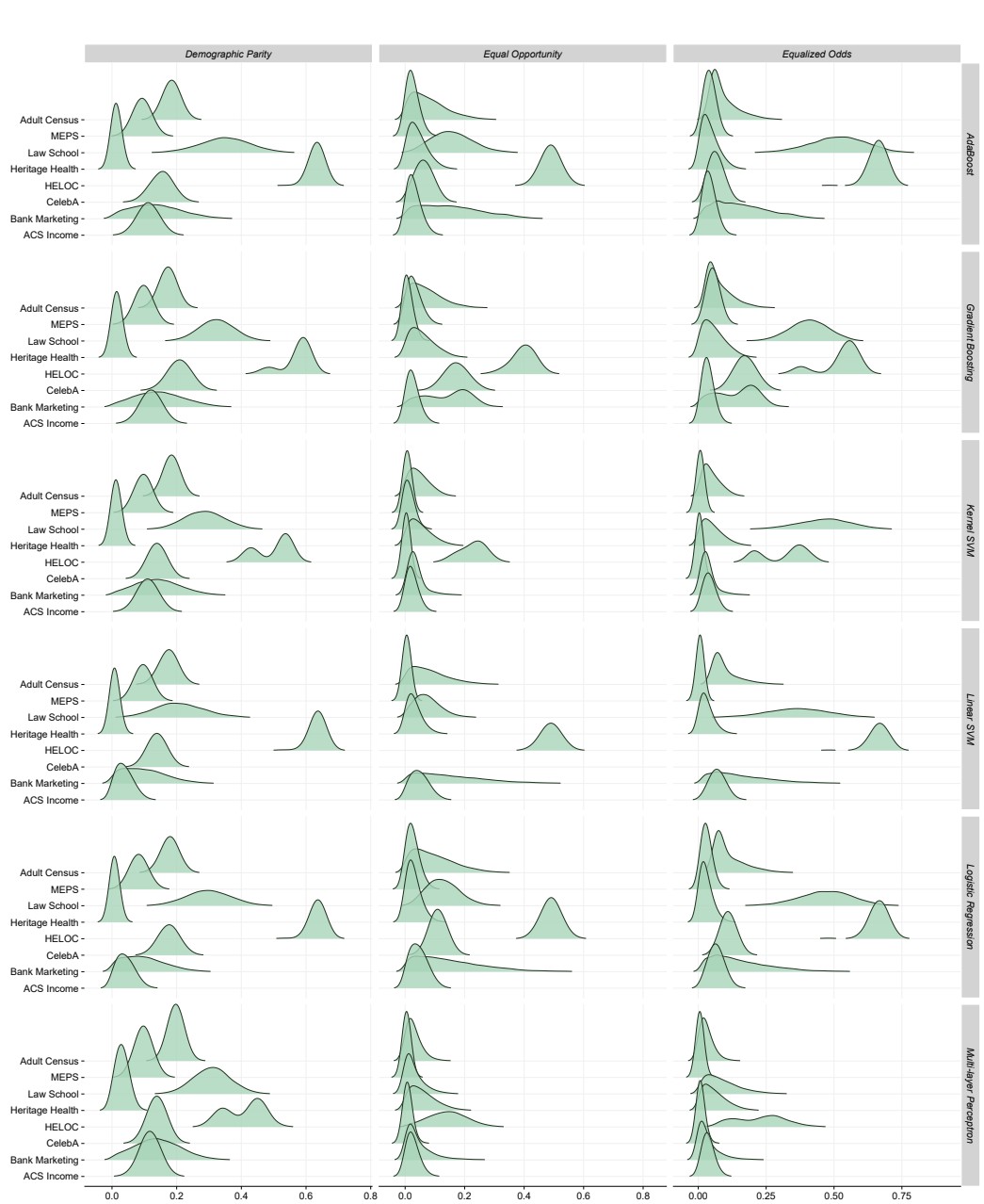

**Figure 4:** Variability of fairness metrics under Scenario 1. The green shaded bands depict the range of Demographic Parity, Equal Opportunity, and Equalized Odds across 10,000 trials, each of which trains a fresh classifier on a new random subsample of 1000 points. The substantial width of these bands illustrates the pronounced fragility of group-fairness measures to sampling variation.

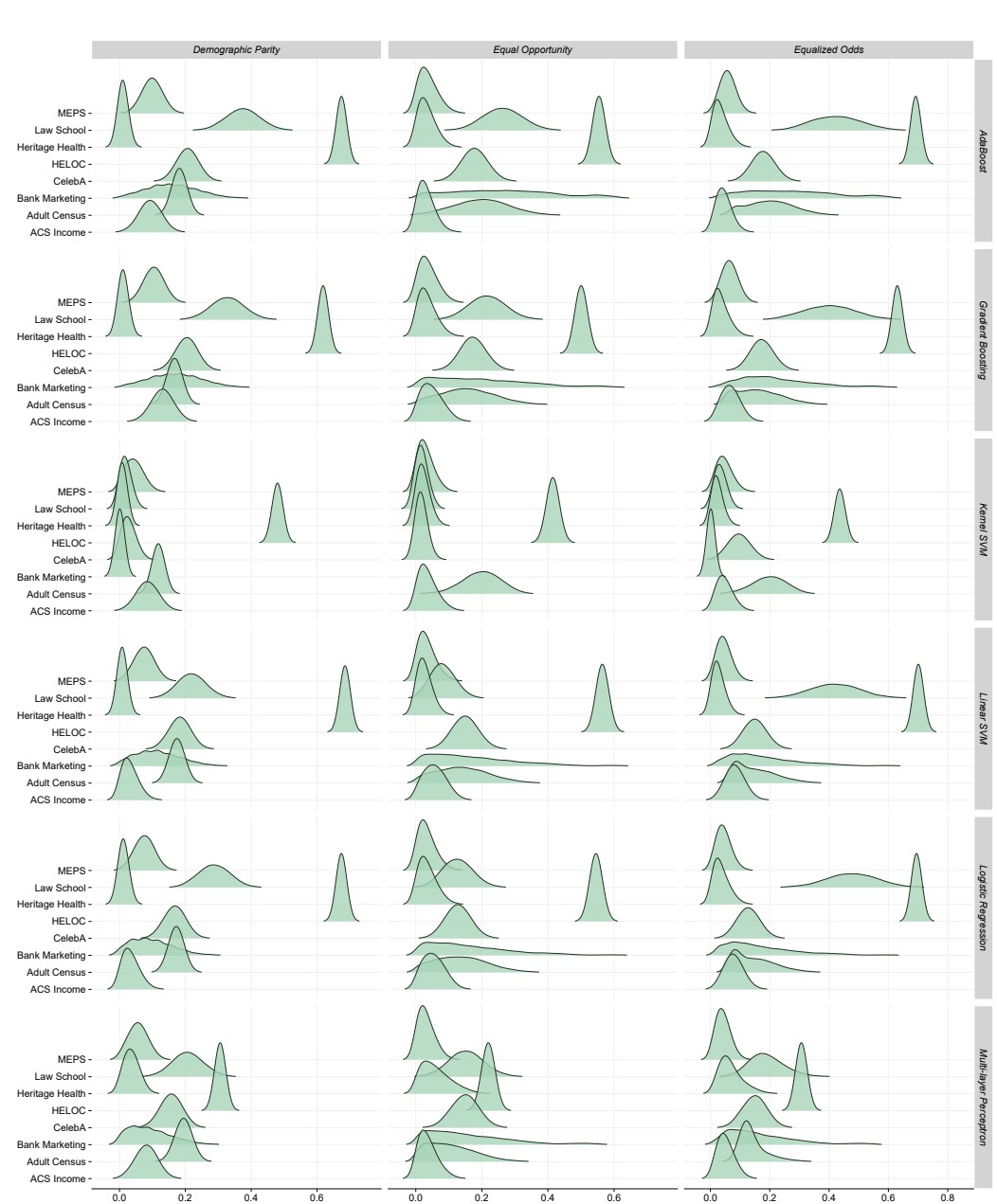

**Figure 5:** Variability of fairness metrics when recomputing on repeated subsamples. A single classifier is trained once on 1000 randomly drawn data points, and then Demographic Parity, Equal Opportunity, and Equalized Odds are recalculated over 10,000 different subsamples of size 1000. The shaded green bands reveal the extent to which fairness assessments fluctuate purely due to sampling variation.

