# OpenReview forum: "From Fragile to Certified: Wasserstein Audits of Group Fairness Under Distribution Shift"
_ICLR.cc/2026/Conference — Submitted to ICLR 2026_

### Official Review · Reviewer_jdqg · 2025-10-24

**Soundness:** 3
**Presentation:** 2
**Contribution:** 2
**Rating:** 2
**Confidence:** 4

**Summary:**

The paper studies the instability of existing group fairness functionals under distributional shifts. It proposes a distributionally robust framework that assesses fairness in the worst case over all test distributions within a Wasserstein ball centered at the empirical data. The key concept, $\varepsilon$-Wasserstein Distributional Fairness (WDF), provides a formal criterion for certifying fairness robustness. The authors present theoretical results ensuring feasibility and consistency, showing that the proposed framework offers stable and reliable fairness evaluation beyond a single observed dataset.

**Strengths:**

The main contribution lies in formulating fairness auditing as a distributionally robust optimization (DRO) problem under Wasserstein uncertainty. This approach generalizes group fairness notions such as demographic parity and equalized odds within a unified framework. The paper derives a tractable reformulation via strong duality and establishes finite-sample guarantees connecting empirical and population-level fairness. The method provides formal robustness certification, addressing a key limitation of prior fairness approaches that lack distributional reliability. Overall, the work offers a clear theoretical foundation and a systematic formulation for fairness auditing under distribution shift.

**Weaknesses:**

While the paper presents a clear and well-motivated formulation, its degree of theoretical novelty appears somewhat limited. The proposed framework essentially extends existing Wasserstein DRO theory to a fairness functional, without introducing new theoretical or algorithmic components. Although the application of DRO to fairness auditing is conceptually meaningful, many of the theoretical results—such as dual reformulation and finite-sample guarantees—are direct consequences of established DRO literature rather than novel contributions.

Beyond the issue of limited novelty, several other practical aspects deserve further consideration:

1. The claim of improved stability under distributional shifts is not fully substantiated. Figure 2 demonstrates that WDF serves as an upper bound on fairness violation; however, this alone does not establish empirical stability (in fact, the variability between the empirical estimates and the WDF estimates appears to be quite similar). The authors motivate the introduction of WDF by showing in Figure 1 that existing fairness functionals are fragile under distributional shifts. However, this qualitative observation does not quantitatively confirm that WDF indeed yields more stable fairness estimates. Analyzing the variability of fairness functionals under resampling or perturbations would provide stronger evidence for the robustness claims beyond merely showing an upper bound. Such an analysis would offer a more comprehensive validation of the proposed framework’s robustness.

2. The framework is theoretically sound but its scalability to high-dimensional or large-scale fairness auditing tasks remains unclear. Solving Wasserstein-based optimization problems typically requires large-scale linear programming or optimal transport computation, which may be computationally demanding in practice. Moreover, the proposed formulation may not readily extend to general fairness functionals, as many fairness notions are non-differentiable or non-convex, making their integration into the Wasserstein framework non-trivial. A discussion of approximate or scalable alternatives would enhance the practical impact of the work.

**Questions:**

The choice of the Wasserstein radius $\delta$ is fixed (e.g., 0.01) without explicit justification or sensitivity analysis. Since $\delta$ controls the size of the uncertainty set, it plays a crucial role in determining the robustness–accuracy balance. Providing a principled selection rule or an empirical calibration strategy for $\delta$ would strengthen the practical validity and reproducibility of the results.

---

> ### Author Response · Authors · 2025-11-13
> **Response to Reviewer  jdqg**
>
> Thank you, Reviewer, for taking the time to carefully read our paper and provide detailed feedback.
>
> ---
>
> ### 1. On Theoretical Novelty
>
> We respectfully disagree with the assessment that our theoretical results are "direct consequences of established DRO literature." Our work introduces several novel elements:
>
> 1. Tractable reformulation for non-smooth objectives: Standard DRO typically assumes smooth or Lipschitz losses. Our core contribution (Thm. 1, 2) is a novel tractable reformulation specifically for discontinuous indicator-based fairness metrics. This required new techniques to handle the non-smoothness, a specific challenge not addressed by generic DRO.
>
> 2. Fairness-specific bounds with margin analysis: Thms. 5-6 and Prop. 6 provide new quantitative bounds specific to the fairness auditing context. They reveal how decision boundary geometry impacts robust certification, a connection absent in general DRO literature.
>
> 3. Fairness-specific certification and calibration: We prove a finite-sample lower bound (Thm. 4) and a practical calibration rule $\delta_{N} = \delta + \alpha/\sqrt{N}$ (Prop. 5). Crucially, these are stated for the fairness score f(Z) itself, not a generic smooth loss, making them a fairness-specific contribution.
>
> 4. Efficient DRUNE Estimator (Alg. 1): A novel, efficient algorithm combining Newton-KKT projections with a greedy knapsack solution.
>
>
> ### 2. Empirical Validation (W1)
>
> We thank the reviewer for this point and believe there is a misunderstanding of the purpose of Fig. 2.
>
> - Fig.1 (Motivation): Shows that empirical point estimates are fragile and unreliable (wide red/blue bands).
>
> - WDF (Our Solution): $\delta$-WDF is not another point estimate; it is a certified upper bound. Its value is reliability, not low variance. It provides one number that provably bounds the fairness violation for an entire ball of distributions.
>
> - Fig. 2 (Verification): This figure verifies our theory. The reviewer notes the variability "appears to be quite similar," but this misses the key point.
>
>   - The red cloud (Empirical) is centered on the $y=x$ line. As Fig. 2(c) shows, the (Empirical - True) difference is centered at 0, meaning it's an unreliable estimate that underestimates the true unfairness ~50% of the time.
>
>   - The blue cloud ($\delta$-WDF) is provably an upper bound. Fig. 2(c) confirms this: the (WDF - True) distribution is almost positive by significance level. It is a reliable certificate.
>
>
> This is the stability we claim: WDF provides a trustworthy upper bound that guards against the unreliability demonstrated in Fig. 1, exactly as our theory (Thm. 4, Prop. 5) guarantees.
>
>
> ### 3. Scalability Concerns (W2)
>
> The complexity of DRUNE (Thm. 3) is $O(Nd^3 + N \log N)$, linear in $N$. The $O(d^3)$ per-point cost is for the exact KKT projection. We address scalability in two ways:
>
> 1. Large N: For the distance computation, alternatives like the Fast Sweeping method (Alg. 2) achieve $O(N_{grid})$ complexity, which is practical.
>
> 2. Large d: For high dimensions where the exact projection is costly, our theoretical bounds (Thm. 5, 6; Prop. 6) provide a practical alternative for estimation, as discussed in Section 5.
>
>
> For more details, see the reviewer Deok Rebattul.
>
>
> ### 4. Non-Differentiable/Non-Convex (W2)
>
> The reviewer worries our formulation "may not readily extend" to "non-differentiable or non-convex" functionals. We are happy to clarify that this is precisely the problem our paper solves.
>
> - Our entire framework is built on the non-continuous indicator $h_\theta(x) = {I}(g_\theta(x) \ge 0)$ (Assumption i).
>
> - Our regularizers (Thm 1, 2) are derived specifically for this non-differentiable case by leveraging distance to the boundary.
>
> ### 5. Choice of $\delta$ (Q1)
>
> This is an excellent question. The specific value of $\delta = 0.01$ mentioned is not a fixed constant but was chosen approximately only for the specific dataset and sample size used in the illustrative example.
>
> In the general case, the estimation of $\delta$ is detailed in the proof of Thm. 4 and Prop. 5. In both results, $\delta$ is determined by solving constraints that involve the sample size $N$, the probability ${P}$, the parameter space, and the diameter of the feature space. This process yields an empirical radius $\delta_N$ that naturally adapts to the sample size $N$.
>
> An interesting consequence is the resulting error rate: In the out-of-sample setting, we obtain $\delta_N = O(N^{-1/4})$, while in the more general Distributionally Robust Optimization (DRO) setting, we recover the usual curse of dimensionality, with $\delta_N = O(N^{-1/d})$, where $d$ is the dimension of the feature space. Further detailed discussion about error rates and their implications will be postponed to following works.
>
> ---
>
> We hope these clarifications address your concerns. If our responses are convincing, we would be very grateful if you could consider updating your overall evaluation accordingly.

---

> > ### Comment · Reviewer_jdqg · 2025-11-21
> >
> > Thank you for the clarification and additional explanations. Overall, I agree that the paper makes a meaningful contribution to fairness auditing under distribution shift.
> >
> > I still believe that, without explicitly considering variance, the resulting upper bound can become overly loose; however, I acknowledge that addressing this is likely beyond the scope of the current paper.
> >
> > I will continue to follow the remaining discussions with the other reviewers and will raise my score in the final update.

---

> > > ### Author Response · Authors · 2025-11-21
> > > **Thank You for the Re-evaluation**
> > >
> > > Thank you very much for your thoughtful reply and, especially, for your willingness to raise your score. I truly appreciate that you took the time to read the rebuttal carefully and that you now see the paper as making a meaningful contribution.
> > >
> > > I would just like to briefly add a note on the variance point: You are correct that variance affects the tightness of the bound. However, regarding safety, our certification is designed to be probabilistically valid. By choosing a small significance level $\delta$, the upper bound is guaranteed to hold with probability $1-\delta$. Thus, even without explicit variance modeling, the certificate remains a valid worst-case guarantee.

---

### Official Review · Reviewer_PPd1 · 2025-10-28

**Soundness:** 3
**Presentation:** 3
**Contribution:** 2
**Rating:** 6
**Confidence:** 4

**Summary:**

The paper introduces a Wasserstein Distributionally Robust Optimization (DRO) framework to certify group fairness under distributional shifts.

**Strengths:**

- The paper tackles an important problem, i.e.,  the fragility of fairness metrics under distributional changes. This is a critical issue in several real-world applications.
-  By unifying multiple fairness notions, the paper provides a flexible and comprehensive tool for fairness certifications.
-The introduction of DRUNE offers a practical method for estimating worst-case fairness, making the approach applicable to real-world scenarios.
- I did not read all the proofs in details but the results seem reasonable.

**Weaknesses:**

- The analysis assumes that the empirical distribution accurately represents the population, which may be unrealistic in practice, especially for rare subgroups or OOD samples. How robust are the certificates if this assumption fails?

- Although DRUNE is proposed for efficiency, the DRO optimization may remain computationally intensive for large-scale or high-dimensional datasets. Can the method scale in practice?

- Certification relies on finite-sample empirical distributions. For small datasets or complex models, the finite-sample bounds may be loose, which is related to the phenomenon of fairness overfitting (Laakom et al., Fairness Overfitting in Machine Learning: An Information-Theoretic Perspective, ICML 2025), potentially leading to over-optimistic fairness guarantees.  Are the finite-sample bounds reliable enough for your approach to certify fairness in practice, or could they overestimate true robustness?

**Questions:**

See section above.

---

> ### Author Response · Authors · 2025-11-13
> **Response to Reviewer PPd1**
>
> We thank the Reviewer for their time and constructive feedback. We are glad they recognized the importance of the problem, the flexibility of our unifying framework, and the practical value of the DRUNE estimator.
>
> ---
>
> ### 1. On the Assumption of $P^N$ vs. $P$ (W1)
>
> We respectfully clarify that our framework is built on the premise that the empirical distribution $P^N$ is not a reliable representation of the true population $P$. Our model is designed to explicitly address this OOD problem by creating an uncertainty set that accounts for shifts in the feature space ($P_X$).
>
> The limitation of our model is the assumption that the marginal probabilities of labels and sensitive attributes align with the true distribution ($P_{A,Y} = P^N_{A,Y}$). This means we assume the ratio of sensitive groups or labels is known (e.g., gender ratios), while their features are allowed to shift.
>
> We agree that this assumption is a limitation for very rare subgroups whose true distribution is hard to estimate from $P^N$. This is a common and standard assumption [1, 2, 3] to avoid overly complex formulations, especially (as in our work) when the number of labels and sensitive levels is small. As mentioned in our paper, we intend to relax this assumption in future work.
>
> ---
>
> ### 2. On Computational Scalability (W2)
>
> We appreciate the reviewer's valid point regarding the $O(d^3)$ per-point cost of Algorithm 1. However, we must respectfully point out that we address this limitation and provide solutions within the paper. This is a potential bottleneck, which is why we included two distinct, practical solutions to address scalability for both **large N** (dataset size) and **high d** (dimensionality):
>
> - **For High** $d$**:** Section 5 provides analytical, **closed-form upper bounds** for the regularizer (Prop. 6, Thm. 5, Thm. 6). These bounds enable a highly efficient, tractable robust constraint, serving as a proxy for the exact regularizer without requiring $O(d^3)$ computation.
>
> - **For Large** $N$**:** We propose the **Fast Sweeping method** (Algorithm 2, Appendix p. 18), introduced in Section 4 (Lines 310-313). This method has **linear complexity** ($\tilde{O}(N_{grid})$) and is our recommended scalable solution for large datasets, entirely avoiding the per-point $O(d^3)$ cost.
>
> These two solutions directly and complementarily address the scalability concern for practical use. We will clarify in the revision how Algorithm 2 and Section 5 bounds serve as integrated solutions to this computational challenge.
>
> ---
>
> ### 3. On Loose Bounds and Fairness Overfitting (W3)
>
> We thank the reviewer for this insightful comment and for pointing us to the related work of Laakom et al.. We will add this reference and a discussion to our related work section.
>
> The phenomenon the reviewer calls "fairness overfitting" aligns perfectly with what we term "sampling fragility," which we address as a core form of distributional uncertainty. While our motivational examples are similar, our approach is different. Laakom et al. focus on deriving upper bounds for the generalization error of nominal fairness. Our method provides a complementary, and in this case, more direct, solution: we provide finite-sample guarantees for the robust fairness audit itself under not only out-of-sample but also shifted distributions.
>
> Our finite-sample bounds (Thm. 4, Prop. 5) are explicitly designed to be conservative, precisely to counter the risk of "fairness overfitting." Formally, we show that with high probability, our empirical certificate (which we can compute) bounds the true population fairness, which in turn is bounded by a population-level robust guarantee:
>
> $$\left|\mathcal{F}({P}, \theta)\right| \leq \sup\_{Q \in \mathcal{B}\_{\delta}({P}^N)} \left|\mathcal{F}(Q, \theta)\right| \leq \sup\_{Q \in \mathcal{B}\_{\delta + \frac{\alpha}{\sqrt{N}}}({P})} \left|\mathcal{F}(Q, \theta)\right|$$
>
> So the finite-sample bounds are reliable enough, because they are designed to prevent over-optimism. The reviewer is correct that if the sample size $N$ is small (or the model class is complex, increasing $\alpha$), the inflation term $\alpha/\sqrt{N}$ grows. This automatically and correctly makes our certified bound wider (i.e., more conservative). Our method, therefore, does not overestimate robustness; it provides a certificate that is explicitly calibrated by the limitations of the finite sample.
>
> ---
>
> We hope our revisions and clarifications—made possible by your insightful feedback—have fully addressed your concerns, and we would be grateful if you would consider these improvements in your final evaluation of our work.
>
> ---
>
> # References
>
> [1] Taskesen, et al., A distributionally robust approach to fair classification (2020).
>
> [2] Wang, et al., Wasserstein robust classification with fairness constraints (2024).
>
> [3] Si, et al., Testing group fairness via optimal transport projections (2021).

---

> > ### Author Response · Authors · 2025-11-28
> > **Brief Reminder of Our Revisions and Clarifications**
> >
> > We sincerely thank the reviewer once again for their time and thoughtful feedback. As a brief reminder of our previous response, we (i) clarified the role and limitations of our assumption on label and sensitive-group marginals, which was already stated in the paper (W1), (ii) provided additional explanation on how our analytical bounds address scalability in the high-dimensional regimes (W2), and (iii) highlighted the advantages of our framework relative to Laakom et al., and incorporated this discussion into the revised draft. We would be very happy to address any remaining concerns the reviewer may have and look forward to hearing their further comments.

---

### Official Review · Reviewer_Deok · 2025-10-29

**Soundness:** 4
**Presentation:** 3
**Contribution:** 3
**Rating:** 6
**Confidence:** 4

**Summary:**

This paper addresses the fairness assessment under distribution shifts. The core contribution of this work is the proposal of a Wasserstein distributionally robust optimization framework to certify worst-case group fairness. Specifically, they define a $\epsilon$-Wasserstein Distributional Fairness ($\epsilon$-WDF) as a robust audit target that quantifies worse-case group fairness over a set of plausible test distributions, modeled as a Wasserstein ball centered at the empirical data distribution. Recognizing the computational challenges of optimizing over infinite-dimensional sets of distributions, the authors derive tractable reformulations for $\epsilon$-WDF and its associated DRO regularizers. To address the out-of-sample problem, where only empirical data is observed, the paper establishes finite-sample certification guarantees for auditing fairness. And the theoretical insight is that the worst-case fairness estimated from the finite samples could upper bound the true worse-case disparity under shifts within an $\epsilon$-Wasserstein ball.

**Strengths:**

**Originality**:
The paper's primary original contribution lies in the novel formulation of $\epsilon$-WDF. By proposing a Wasserstein distributionally robust framework, the authors introduces a new and more resilient approach to fairness assessment that directly addresses the fragility of existing metrics under distribution shifts. This re-framing of the fairness audit problem, moving towards worse-case certification over plausible distributions, is a significant contribution. While DRO and Wasserstein distances have been applied in various machine learning contexts (especially distribution shifts), their comprehensive integration specifically for certifying group fairness showcases the originality of this framework.

**Quality**:
The theoretical development is rigorous, starting from a clear problem statement and building towards a robust solution.

**Clarity**:
The paper is well-written and generally clear. The logical flow from defining the problem to proposing $\epsilon$-WDF, detailing its theoretical underpinnings, and outlining the estimation method is easy to follow. The mathematical notation is consistent.

**Significance**:
The significance of this work is substantial, with implications for both theoretical research and practical applications. By providing a framework to certify worse-case worse-case group fairness, the paper offers a more reliable basis for auditing algorithmic systems. The proposed DRUNE estimator has the potential to be adopted in practical auditing and training pipelines.

**Weaknesses:**

- **Computational Complexity**: the authors acknowledged that the stage 1 of Algorithm 1 has an $O(d^3)$ per-point time cost and stage 2 runs in $O(N\log N)$ time. While it might be efficient for a handful of closest-point computations, it could still pose a significant bottleneck for high-dimensional feature space (especially embedded with textual or visual embeddings).
- **Comparison to Related Works**: The related work section is underdeveloped and lacks discussions with a broad range of relevant studies. In particular, it fails to explicitly discuss or compare the proposed framework with (Chen et al., 2022), which also addresses fairness certification in the target distribution under small distributional shifts.

[1] Yatong Chen, Reilly Raab, Jialu Wang, and Yang Liu. 2022. Fairness transferability subject to bounded distribution shift. In Proceedings of the 36th International Conference on Neural Information Processing Systems (NIPS '22). Curran Associates Inc., Red Hook, NY, USA, Article 819, 11266–11278.

**Questions:**

1. The notations of $\mathcal{S}_{\delta, q}$ and $\mathcal{I}\_{\delta, q}$ are a bit repetitive. If $f$ was defined as the absolute value, i.e.,  $f=|h(x) \phi(\cdot, \cdot)|$, then the two quantities $\mathcal{S}\_{\delta, q}$ and $\mathcal{I}\_{\delta, q}$ could be consolidated into a single, unified notation $\mathcal{S}\_{\delta, q}$. In that way, Eq. (9), (10), (11) could be furthered reduced into only one regularizer. Similarly, you won't need to swap the 0 and 1 in the afterwards expressions.
2. The time complexity of the DRUNE estimator is $O(d^3)$ to compute $d_i$ for each data point and $O(N\log N)$ for solving the knapsack problem. However, if one wants to incorporate the estimator into a real fairness optimization problem, it may incur such time cost during each iteration (as the model parameters will change). This may not be effective for training a fair classifier.
3. How does the Wasserstein regularizer compare to the bounds in (Chen et al., 2022)? Basically, they also provided provable guarantee of group fairness at the worst case. If we only look at the data examples within the thin boundary, it seems that the regularizers in your work could be converted into the bounds in their work.

[1] Yatong Chen, Reilly Raab, Jialu Wang, and Yang Liu. 2022. Fairness transferability subject to bounded distribution shift. In Proceedings of the 36th International Conference on Neural Information Processing Systems (NIPS '22). Curran Associates Inc., Red Hook, NY, USA, Article 819, 11266–11278.

---

> ### Author Response · Authors · 2025-11-13
> **Response to Reviewer Deok**
>
> We sincerely thank the Reviewer for their positive evaluation, thorough summary, and insightful feedback, and will address each point, referencing strong evidence from our paper.
>
> ---
>
> ### 1. On Computational Complexity (W1 & Q2)
>
> The reviewer correctly notes that Algorithm 1 has an $O(d^3)$ per-point cost, which could be a bottleneck for high-dimensional data. We share this concern, and **our work provides two distinct solutions** for this exact problem, addressing both high-dimensionality ($d$) and large datasets ($N$).
>
> - **For High $d$**: we provide analytical bounds in Section 5 that can be used as an efficient proxy. Proposition 6, Theorem 5, and Theorem 6 provide closed-form upper and lower bounds for the regularizer. Instead of computing the exact regularizer (via Alg. 1 or 2), one can use its analytical upper bound as a more computationally tractable constraint to control worst-case fairness.
>
> - **For Large $N$**: We provide Algorithm 2. As stated in Section 4, "When $N$ is large, the Fast Sweeping method Wong & Leung (2016), which has linear complexity in the grid size $\tilde{O}(N_{grid})$, becomes more efficient." We detail this in Algorithm 2 (Appendix, p. 18). This method avoids the per-point $O(d^3)$ cost and is our proposed scalable solution for large datasets.
>
>
> We will **clarify in the main text** how the bounds in Section 5 work together to overcome these computational challenges.
>
> ---
>
> ### 2. On Notation and $\mathcal{S}/\mathcal{I}$ Regularizers (Q1)
>
> This is an insightful suggestion, but the **separation of $\mathcal{S}\_{\delta,q}$ and $\mathcal{I}\_{\delta,q}$ is fundamental** to our framework's ability to correctly model standard group fairness metrics.
>
> 1. **Fairness as a Signed Difference**: As defined in our paper (Eq. 3 and Eq. 10), standard fairness metrics are *signed differences* of probabilities (e.g., $TPR_A - TPR_B$). This value $\mathcal{F}(\mathbb{P}, \theta)$ can be positive or negative.
>
> 2. **Why Two Regularizers are Necessary**: As proven in Proposition 4 (Eq. 9), this single constraint $\sup ||\mathbb{E}\_Q[f]||_\infty \le \epsilon$ mathematically and *necessarily* decomposes into two distinct conditions:
>
>   - An upper bound on the positive deviation: $\mathcal{S}\_{\delta,q}(\mathbb{P},\theta)+\mathcal{F}(\mathbb{P},\theta)\le\epsilon$
>
>   - A bound on the negative deviation: $\mathcal{I}\_{\delta,q}(\mathbb{P},\theta)-\mathcal{F}(\mathbb{P},\theta)\le\epsilon$
>
>
> If we were to use $f = |h(x)\phi(\cdot, \cdot)|$ as suggested, we would be taking the absolute value *inside* the expectation. This would **change the definition of the fairness** metric itself (to a non-standard one, perhaps more aligned with individual fairness). **Therefore, the two regularizers are essential.**
>
> ---
>
> ### 3. On Comparison to Chen et al. (W2 & Q3)
>
> Thank you for pointing out this highly relevant work. We will add a detailed comparison to our related work section. **While both papers share a high-level goal, our regularizers do not simply convert into their bounds**. Our framework **is fundamentally different** and provides a more comprehensive and provably tighter certificate for several reasons:
>
> 1. **Flexibility of Our Method**: A key limitation of Chen et al. is its reliance on the shared support assumption. **This prevents it from handling the most practical audit scenario**: certifying fairness by transferring from a discrete empirical (source) distribution to a continuous population (target) distribution. Furthermore, their framework is designed for **specific, structured shifts** (e.g., covariate and label shifts) and employs divergence measures (such as $D_g(T\Vert S) := \mathrm{Var}_S[\omega_g(T,S,X)]$) that are explicitly *tailored* to these shift types. In contrast, our Wasserstein framework is a more general, non-parametric uncertainty model. It does not require shared support and makes no assumptions about the *type* of shift, allowing it to robustly model the gap between the empirical sample and the true population law.
>
> 2. **Exact DRO vs. Bounding Change**: Our framework is designed to compute the exact worst-case fairness value ($\epsilon$-WDF) over the entire Wasserstein uncertainty set. We provide tractable reformulations for this exact quantity (Thm. 1, Thm. 2) and an efficient estimator for it (DRUNE, Alg. 1). Our finite-sample guarantees (Thm. 4, Prop. 5) then certify this *value* against the true population distribution. In contrast, Chen et al. (2022) derive bounds on the *change* (or "transferability") of the fairness metric under specific shift models and divergence . They do not compute the exact worst-case fairness value itself, nor do they establish whether their bounds on this change are sharp.
>
>
> ---
>
> We sincerely appreciate your thorough review. We believe the points addressed above substantially strengthen the paper, and we are **fully open to addressing any remaining concerns or further questions** you may have to merit a higher score.

---

> > ### Author Response · Authors · 2025-11-28
> > **Brief Reminder of Our Revisions and Clarifications**
> >
> > We sincerely thank the reviewer once again for their time and thoughtful feedback. As a brief reminder of our previous response, we (i) provided additional explanation on how our analytical bounds address scalability in both high-dimensional (W1 & Q2), (ii) explained why standard signed group-fairness metrics mathematically require two separate Wasserstein regularizers rather than a single absolute-value term (Q1), and (iii) added a clearer comparison to Chen et al., emphasizing the greater generality of our Wasserstein framework and its ability to compute the exact worst-case fairness value with finite-sample guarantees (W2 & Q3). We would be very happy to address any remaining concerns the reviewer may have and look forward to their further comments.

---

### Author Response · Authors · 2025-11-19
**Request for Feedback on Rebuttal During Discussion Phase**

Dear Reviewers,

First of all, we would like to sincerely thank you for the time and effort you have already invested in reviewing our submission. We fully understand that you are very busy and that reviewing is a demanding service to the community.

However, as we are now already about a week into the rebuttal/discussion phase, we are concerned that we have not yet received any responses to our rebuttal. We spent about a year working on this paper, and we prepared our rebuttal very quickly and carefully precisely so that there would be enough time for an interactive discussion and to clarify any misunderstandings or concerns you might have.

We deeply appreciate your expertise and your time, and we know that reviewing is voluntary work. At the same time, for many authors, including us, the outcome of these reviews and the possibility to receive clarifications can have a significant impact on our careers. If possible, we would be very grateful if you could briefly engage with our rebuttal and indicate whether our clarifications address some of your concerns or if there are remaining issues we should focus on.

Thank you again for your time and service to the community.

---

### Author Response · Authors · 2025-12-03
**Official Comment to the Area Chairs**

Dear Area Chairs,

As the discussion phase concludes, we thank the reviewers for their time and constructive engagement and appreciate your efforts in leading the remaining review process. We are pleased to report that following our rebuttal, a consensus has emerged regarding the technical soundness and significance of our contributions. We provide a brief summary of the review process and the resolution of key questions below.

**1. Consensus on Significance and Rigor:** We are encouraged that the reviewers have recognized the core value of our proposed $\epsilon$-Wasserstein Distributional Fairness ($\epsilon$-WDF) framework:

- **Reviewer Deok** assessed the work as **"Soundness: Excellent"** and highlighted the **"substantial significance"** of moving fairness audits toward worst-case certification. They noted the theoretical development is "rigorous."

- **Reviewer PPd1** recognized the method as a "flexible and comprehensive tool" for addressing the critical real-world issue of fairness fragility.

- **Reviewer jdqg** (post-rebuttal) affirmed that the paper makes a **"meaningful contribution to fairness auditing under distribution shift."**


**2. Resolution of Concerns and Score Increase (Reviewer jdqg)** Reviewer `jdqg` initially raised concerns regarding theoretical novelty and empirical stability (giving a score of 2). In our detailed response, we clarified:

- **Novelty:** We demonstrated that our work is not merely an application of standard DRO, but requires novel tractable reformulations for **non-smooth** fairness indicators and derives fairness-specific margin bounds (Thm 5-6).

- **Stability:** We clarified that $\epsilon$-WDF is a *certified upper bound*, designed to provide safety guarantees rather than low-variance point estimation.


**Outcome:** Following this exchange, **Reviewer jdqg explicitly stated:** *"I agree that the paper makes a meaningful contribution... and I will raise my score in the final update."*

**3. Summary of Main Issues and Technical Resolutions** We have successfully addressed the common technical inquiries raised across the reviews:

- **Computational Scalability (Reviewers Deok, PPd1, jdqg):**

  - *Concern:* The $O(d^3)$ per-point complexity of the exact estimator (DRUNE).

  - *Resolution:* We highlighted our dual-pronged solution, ensuring scalability:

    1. **For High Dimension (**$d$**):** We provide analytical closed-form bounds (Section 5) that serve as efficient proxies, bypassing the optimization cost.

    2. **For Large Sample Size (**$N$**):** We utilize the Fast Sweeping method (Algorithm 2), reducing complexity to linear time $\tilde{O}(N_{grid})$.


    In the revised version (highlighted in blue), we have expanded the explanation of how our method addresses scalability.

- **Comparison to Related Work (Reviewers Deok, PPd1):**

  - *Concern:* Comparison with Chen et al. (2022) and Laakom et al. (2025).

  - *Resolution:* We differentiated our work by showing that:

    1. Unlike Chen et al., our framework does **not** assume shared support, allowing for general shifts from discrete empirical data to continuous populations, and provides exact worst-case values rather than transferability bounds.

    2. Regarding Laakom et al. ("Fairness Overfitting"), our method *is* the solution to this phenomenon. Our finite-sample certificates are explicitly constructed to prevent over-optimism by bounding the population fairness with high probability.


    In the revised version (highlighted in blue), we added explicit mentions and explanations of both papers in the related work section.

- **Theoretical Novelty (Raised by jdqg):**

  - *Concern:* Perceived as extensions of existing DRO without new components.

  - *Resolution:* We highlighted novel elements: Tractable reformulations for non-smooth indicator-based fairness (Thms. 1-2), fairness-specific bounds linking decision boundaries to robustness (Thms. 5-6, Prop. 6), calibration rules for the fairness score itself (Thm. 4, Prop. 5), and the efficient DRUNE algorithm (Alg. 1). jdqg found this convincing.

- **Assumptions on empirical vs. population distribution (Reviewer PPd1)**:

  - *Concern*: Reliance on empirical representation of population (esp. for rare subgroups/OOD)

  - *Resolution:* We explained our framework explicitly models shifts (features can shift via Wasserstein ball, while label/sensitive marginals are assumed known—a standard assumption (Taskesen (2020); Si (2021); Wang (2024)); As we clarified in the discussion section, we adopted this assumption to avoid overcomplicating the theoretical development in the present paper, and we plan to relax it in subsequent work.

We believe the rebuttal has resolved the technical concerns and confirmed the validity of our theoretical claims. We stand ready to provide any final clarifications the AC might require.

Sincerely,
The Authors

---

### Meta-Review · Area_Chair_RHK8 · 2026-01-08

**Summary:**

This submission proposes a fairness certification method that provides rigorous model fairness bounds under Wasserstein-constrained distribution shift. Besides the theoretical bounds, the finite-sample guarantee and optimized computing methods are provided to make the method practical.

Reviewers appreciated this new and more resilient fairness assessment notion and the accompanied certification approach that tackles an important problem. On the other hand, here are major concerns from reviewers:
1. (shared concern) The computational complexity is high. Whether the method can scale to larger model sizes or larger dataset sizes is questionable.
2. The related work section is relatively undeveloped.
3. The finite bound may not work for small datasets or complex models.
4. Novelty is limited - it just extends existing Wasserstein DRO theory to a fairness functional.

In addition, a criticial concern is about the page limit violation. The revised manuscript exceeds 10 pages. Aaccording to the Author Guide https://iclr.cc/Conferences/2026/AuthorGuide,
> At the time of submission, the main text should be 9 pages or fewer. During the discussion/rebuttal phase and for the camera ready, the page limit will be increased to 10 pages to allow for new results/discussions. This limit will be strictly enforced. Papers with main text beyond the page limit will be desk-rejected.
As a result, we recommend rejection.

**Reviewer Concerns:**

The authors provided detailed rebuttals and revised the manuscript. The major concerns are well addressed. Maybe the only remaining concern is the computational efficiency. Though the authors introduce acceleration techniques, whether it still provides a strong fairness certificate for large models or large datasets is still questionable. However, I understand that the authors have already made great progress to this end.

**Reviewer Scores:**

The only negative-score reviewer, Reviewer jdqg, replied and promised to increase their score. Hence, I would foresee that the submission will end up with overall positive review scores (or close to that) if reviewers had participated fully in the discussion.

However, due to the violation of the page limit, according to the conference rule, I recommend rejection.

---

### Decision · Program_Chairs · 2026-01-26

Reject